# A Kernel Perspective of Skip Connections in Convolutional Networks

**Daniel Barzilai, Amnon Geifman, Meirav Galun & Ronen Basri**
Weizmann Institute of Science
{daniel.barzilai,amnon.geifman,meirav.galun,ronen.basri}@weizmann.ac.il

## Abstract

Over-parameterized residual networks are amongst the most successful convolutional neural architectures for image processing. Here we study their properties through their Gaussian Process and Neural Tangent kernels. We derive explicit formulas for these kernels, analyze their spectra and provide bounds on their implied condition numbers. Our results indicate that (1) with ReLU activation, the eigenvalues of these residual kernels decay polynomially at a similar rate as the same kernels when skip connections are not used, thus maintaining a similar frequency bias; (2) however, residual kernels are more locally biased. Our analysis further shows that the matrices obtained by these residual kernels yield favorable condition numbers at finite depths than those obtained without the skip connections, enabling therefore faster convergence of training with gradient descent.

## 1 Introduction

In the past decade, deep convolutional neural network (CNN) architectures with hundreds and even thousands of layers have been utilized for various image processing tasks. Theoretical work has indicated that shallow networks may need exponentially more nodes than deep networks to achieve the same expressive power (Telgarsky, 2016; Poggio et al., 2017). A critical contribution to the utilization of deeper networks has been the introduction of Residual Networks (He et al., 2016).

To gain an understanding of these networks, we turn to a recent line of work that has made precise the connection between neural networks and kernel ridge regression (KRR) when the width of a network (the number of channels in a CNN) tends to infinity. In particular, for such a network $f(\mathbf{x}; \theta)$, KRR with respect to the corresponding Gaussian Process Kernel (GPK) $\mathcal{K}(\mathbf{x}, \mathbf{z}) = \mathbb{E}_\theta[f(\mathbf{x}; \theta) \cdot f(\mathbf{z}; \theta)]$ (also called Conjugate Kernel or NNGP Kernel) is equivalent to training the final layer while keeping the weights of the other layers at their initial values (Lee et al., 2017). Furthermore, KRR with respect to the neural tangent kernel $\Theta(\mathbf{x}, \mathbf{z}) = \mathbb{E}_\theta\left[\left\langle \frac{\partial f(\mathbf{x}; \theta)}{\partial \theta}, \frac{\partial f(\mathbf{z}; \theta)}{\partial \theta} \right\rangle\right]$ is equivalent to training the entire network (Jacot et al., 2018). Here $\mathbf{x}$ and $\mathbf{z}$ represent input data items, $\theta$ are the network parameters, and expectation is computed with respect to the distribution of the initialization of the network parameters.

We distinguish between four different models; Convolutional Gaussian Process Kernel (CGPK), Convolutional Neural Tangent Kernel (CNTK), and ResCGPK, ResCNTK for the same kernels with additional skip connections. Yang (2020); Yang & Littwin (2021) showed that for any architecture made up of convolutions, skip-connections, and ReLUs, in the infinite width limit the network converges almost surely to its NTK. This guarantees that sufficiently over-parameterized ResNets converge to their ResCNTK.

Lee et al. (2019; 2020) showed that these kernels are highly predictive of finite width networks as well. Therefore, by analyzing the spectrum and behavior of these kernels at various depths, we can better understand the role of skip connections. Thus the question of what we can learn about skip connections through the use of these kernels begs to be asked. In this work, we aim to do precisely that. By analyzing the relevant kernels, we expect to gain information that is applicable to finite width networks. Our contributions include:

1. A precise closed form recursive formula for the Gaussian Process and Neural Tangent Kernels of both equivariant and invariant convolutional ResNet architectures.

2. A spectral decomposition of these kernels with normalized input and ReLU activation, showing that the eigenvalues decay polynomially with the frequency of the eigenfunctions.

3. A comparison of eigenvalues with non-residual CNNs, showing that ResNets resemble a weighted ensemble of CNNs of different depths, and thus place a larger emphasis on nearby pixels than CNNs.

4. An analysis of the condition number associated with the kernels by relating them to the so called double-constant kernels. We use these tools to show that skip connections speed up the training of the GPK.

Derivations and proofs are given in the Appendix.

## 2  RELATED WORK

The equivalence between over-parameterized neural networks and positive definite kernels was made precise in (Lee et al., 2017; Jacot et al., 2018; Allen-Zhu et al., 2019; Lee et al., 2019; Chizat et al., 2019; Yang, 2020) amongst others. Arora et al. (2019a) derived NTK and GPK formulas for convolutional architectures and trained these kernels on CIFAR-10. Arora et al. (2019b) showed subsequently that CNTKs can outperform standard CNNs on small data tasks.

A number of studies analyzed NTK for fully connected (FC) architectures and their associated Reproducing Kernel Hilbert Spaces (RKHS). These works showed for training data drawn from a uniform distribution over the hypersphere that the eigenvalues of NTK and GPK are the spherical harmonics and with ReLU activation the eigenvalues decay polynomially with frequency (Bietti & Bach, 2020). Bietti & Mairal (2019) further derived explicit feature maps for these kernels. Geifman et al. (2020) and Chen & Xu (2020) showed that these kernels share the same functions in their RKHS with the Laplace Kernel, restricted to the hypersphere.

Recent works applied spectral analysis to kernels associated with standard convolutional architectures that include no skip connections. (Geifman et al., 2022) characterized the eigenfunctions and eigenvalues of CGPK and CNTK. Xiao (2022); Cagnetta et al. (2022) studied CNTK with non-overlapped filters, while Xiao (2022) focused on high dimensional inputs.

Formulas for NTK for residual, *fully connected* networks were derived and analyzed in Huang et al. (2020); Tirer et al. (2022). They further showed that, in contrast with FC-NTK and with a particular choice of balancing parameter relating the skip and the residual connections, ResNTK does not become degenerate as the depth tends to infinity. As we mention later in this manuscript, this result critically depends on the assumption that the last layer is *not* trained. Belfer et al. (2021) showed that the eigenvalues of ResNTK for fully connected architectures decay polynomially at the same rate as NTK for networks without skip connections, indicating that residual and conventional FC architectures are subject to the same frequency bias.

In related works, (Du et al., 2019) proved that training over-parametrized convolutional ResNets converges to a global minimum. (Balduzzi et al., 2017; Philipp et al., 2018; Orhan & Pitkow, 2017) showed that deep residual networks better address the problems of vanishing and exploding gradients compared to standard networks, as well as singularities that are present in these models. Veit et al. (2016) made the empirical observation that ResNets behave like an ensemble of networks. This result is echoed in our proofs, which indicate that the eigenvalues of ResCNTK are made of weighted sums of eigenvalues of CNTK for an ensemble of networks of different depths.

Below we derive explicit formulas and analyze kernels corresponding to *residual, convolutional* network architectures. We provide lower and upper bounds on the eigenvalues of ResCNTK and ResCGPK. Our results indicate that these residual kernels are subject to the same frequency bias as their standard convolutional counterparts. However, they further indicate that residual kernels are significantly more locally biased than non-residual kernels. Indeed, locality has recently been attributed as a main reason for the success of convolutional networks (Shalev-Shwartz et al., 2020; Favero et al., 2021). Moreover, we show that with the standard choice of constant balancing parameter used in practical residual networks, ResCGPK attains a better condition number than the standard CGPK, allowing it to train significantly more efficiently. This result is motivated by the work of Lee

et al. (2019); Xiao et al. (2020) and Chen et al. (2021), who related between the condition number of NTK and the trainability of corresponding finite width networks.

## 3 PRELIMINARIES

We consider mutli-channel 1-D input signals $\mathbf{x} \in \mathbb{R}^{C_0 \times d}$ of length $d$ with $C_0$ channels. We use 1-D input signals to simplify notations and note that all our results can naturally be extended to 2-D signals. Let $\mathbb{MS}(C_0, d) = \underbrace{\mathbb{S}^{C_0-1} \times \ldots \times \mathbb{S}^{C_0-1}}_{d \text{ times}} \subseteq \sqrt{d} \mathbb{S}^{dC_0-1}$ be the multi-sphere, so $\mathbf{x} = (\mathbf{x}_1, ..., \mathbf{x}_d) \in \mathbb{MS}(C_0, d)$ iff $\forall i \in [d], \|\mathbf{x}_i\| = 1$. For our analysis, we assume that the input signals are distributed uniformly on the multi-sphere.

The discrete convolution of a filter $\mathbf{w} \in \mathbb{R}^q$ with a vector $\mathbf{v} \in \mathbb{R}^d$ is defined as $[\mathbf{w} * \mathbf{v}]_i = \sum_{j=-\frac{q-1}{2}}^{\frac{q-1}{2}} [\mathbf{w}]_{j+\frac{q+1}{2}} [\mathbf{v}]_{i+j}$, where $1 \leq i \leq d$. We use circular padding, so indices $[\mathbf{v}]_j$ with $j \leq 0$ and $j > d$ are well defined.

We use multi-index notation denoted by bold letters, i.e., $\mathbf{n}, \mathbf{k} \in \mathbb{N}^d$, where $N$ is the set of natural numbers including zero. $b_\mathbf{n}, \lambda_\mathbf{k} \in \mathbb{R}$ are scalars that depend on $\mathbf{n}, \mathbf{k}$, and for $\mathbf{t} \in \mathbb{R}^d$ we let $\mathbf{t}^\mathbf{n} = t_1^{n_1} \cdot \ldots \cdot t_d^{n_d}$. As is convention, we say that $\mathbf{n} \geq \mathbf{k}$ iff $n_i \geq k_i$ for all $i \in [d]$. Thus, the power series $\sum_{\mathbf{n} \geq \mathbf{0}} b_\mathbf{n} \mathbf{t}^\mathbf{n}$ should read $\sum_{n_1 \geq 0, n_2 \geq 0, \ldots} b_{n_1, n_2, \ldots} t_1^{n_1} t_2^{n_2} \ldots$

We further use the following notation to denote sub-vectors and sub-matrices. $\forall i \in \mathbb{N}$, let $\mathcal{D}_i = (i+j)_{j=-\frac{q-1}{2}}^{\frac{q-1}{2}}$, so that $[\mathbf{v}]_{\mathcal{D}_i} = [\mathbf{v}]_{i-\frac{q-1}{2}:i+\frac{q-1}{2}}$. Additionally, $\forall i, i' \in \mathbb{N}$, let $\mathcal{D}_{i,i'} = \mathcal{D}_i \times D_{i'}$, so that for a matrix $M$ we can write: $M_{\mathcal{D}_{i,i'}} = M_{i-\frac{q-1}{2}:i+\frac{q-1}{2}, i'-\frac{q-1}{2}:i'-\frac{q-1}{2}}$. We use $(\mathbf{s}_i \mathbf{v})_j = v_{j+i}$ to denote the cyclic shift of $\mathbf{v}$ to the left by $i$ pixels.

Finally, for every kernel $\mathcal{K} : \mathbb{R}^d \times \mathbb{R}^d \to \mathbb{R}$ we define the normalized kernel to be $\overline{\mathcal{K}}(\mathbf{x}, \mathbf{z}) = \frac{\mathcal{K}(\mathbf{x}, \mathbf{z})}{\sqrt{\mathcal{K}(\mathbf{x}, \mathbf{x}) \mathcal{K}(\mathbf{z}, \mathbf{z})}}$. Note that $\overline{\mathcal{K}}(\mathbf{x}, \mathbf{x}) = 1, \forall \mathbf{x} \in \mathbb{R}^d$, and $\overline{\mathcal{K}} \in [-1, 1]$.

### 3.1 CONVOLUTIONAL RESNET

We consider a residual, convolutional neural network with $L$ hidden layer (often just called ResNet). Let $\mathbf{x} \in \mathbb{R}^{C_0 \times d}$ and $q$ be the filter size. We define the hidden layers of the Network as:

$$f_i^{(0)}(\mathbf{x}) = \frac{1}{\sqrt{C_0}} \sum_{j=1}^{C_0} \mathbf{V}_{1,j,i}^{(0)} * \mathbf{x}_j, \qquad g_i^{(1)}(\mathbf{x}) = \frac{1}{\sqrt{C_0}} \sum_{j=1}^{C_0} \mathbf{W}_{1,j,i}^{(1)} * \mathbf{x}_j \tag{1}$$

$$f_i^{(l)}(\mathbf{x}) = f_i^{(l-1)}(\mathbf{x}) + \alpha \sqrt{\frac{c_v}{qC_l}} \sum_{j=1}^{C_l} \mathbf{V}_{:,j,i}^{(l)} * \sigma\left(g_j^{(l)}(\mathbf{x})\right), \quad l = 1, ..., L, \ i = 1, \ldots, C_l \tag{2}$$

$$g_i^{(l)}(\mathbf{x}) = \sqrt{\frac{c_w}{qC_{l-1}}} \sum_{j=1}^{C_{l-1}} \mathbf{W}_{:,j,i}^{(l)} * f_j^{(l-1)}(\mathbf{x}), \quad l = 2, ..., L, \ i = 1, \ldots, C_l, \tag{3}$$

where $C_l \in \mathbb{N}$ is the number of channels in the $l$'th layer; $\sigma$ is a nonlinear activation function, which in our analysis below is the ReLU function; $\mathbf{W}^{(l)} \in \mathbb{R}^{q \times C_{l-1} \times C_l}, \mathbf{V}^{(l)} \in \mathbb{R}^{q \times C_l \times C_l}, \mathbf{W}^{(1)}, \mathbf{V}^{(0)} \in \mathbb{R}^{1 \times C_0 \times C_1}$ are the network parameters, where $\mathbf{W}^{(1)}, \mathbf{V}^{(0)}$ are convolution filters of size 1, and $\mathbf{V}^{(0)}$ is fixed throughout training; $c_v, c_w \in \mathbb{R}$ are normalizing factors set commonly as $c_v = 1/\mathbb{E}_{u \sim \mathcal{N}(0,1)}[\sigma(u)]$ (for ReLU $c_v = 2$) and $c_w = 1$; $\alpha$ is a balancing factor typically set in applications to $\alpha = 1$, however previous analyses of non-covolutional kernels also considered $\alpha = L^{-\gamma}$, with $0 \leq \gamma \leq 1$. We will occasionally omit explicit reference to $c_v$ and $c_w$ and assume in such cases that $c_v = 2$ and $c_w = 1$.

As in Geifman et al. (2022), we consider three options for the final layer of the network:

$$f^{\text{Eq}}(\mathbf{x}; \theta) := \frac{1}{\sqrt{C_L}} \mathbf{W}^{\text{Eq}} f_{:,1}^{(L)}(\mathbf{x})$$

$$f^{\mathrm{Tr}}\left(\mathbf{x};\theta\right) := \frac{1}{\sqrt{d}\sqrt{C_L}}\left\langle \mathbf{W}^{\mathrm{Tr}}, f^{(L)}\left(\mathbf{x}\right)\right\rangle$$

$$f^{\mathrm{GAP}}\left(\mathbf{x};\theta\right) := \frac{1}{d\sqrt{C_L}}\mathbf{W}^{\mathrm{GAP}}f^{(L)}\left(\mathbf{x}\right)\vec{\mathbf{1}},$$

where $\mathbf{W}^{\mathrm{Eq}}, \mathbf{W}^{\mathrm{GAP}} \in \mathbb{R}^{1\times C_L}, \mathbf{W}^{\mathrm{Tr}} \in \mathbb{R}^{C_L\times d}$ and $\vec{\mathbf{1}} = (1,\dots,1)^T \in \mathbb{R}^d$.

$f^{\mathrm{Eq}}$ is fully convolutional. Therefore, applying it to all shifted versions of the input results in a network that is shift-equivariant. $f^{\mathrm{Tr}}$ implements a linear layer in the last layer and $f^{\mathrm{GAP}}$ implements a global average pooling (GAP) layer, resulting in a shift invariant network. The three heads allow us to analyze kernels corresponding to (1) shift equivariant networks (e.g., image segmentation networks), (2) a convolutional network followed by a fully connected head, akin to AlexNet (Krizhevsky et al., 2017) (but with additional skip connections), and (3) a convnet followed by global average pooling, akin to ResNet (He et al., 2016).

Note that $f^{(l)}\left(\mathbf{x}\right) \in \mathbb{R}^{C_l\times d}$ and $g^{(l)}\left(\mathbf{x}\right) \in \mathbb{R}^{C_l\times d}$. $\theta$ denote all the network parameters, which we initialize from a standard Gaussian distribution as in (Jacot et al., 2018).

## 3.2 Multi-dot product Kernels

Following (Geifman et al., 2022), we call a kernel $\mathcal{K} : \mathbb{MS}\left(C_0, d\right)\times\mathbb{MS}\left(C_0, d\right) \to \mathbb{R}$ *multi-dot product* if $\mathcal{K}\left(\mathbf{x}, \mathbf{z}\right) = \mathcal{K}\left(\mathbf{t}\right)$ where $\mathbf{t} = \left(\left(\mathbf{x}^T\mathbf{z}\right)_{1,1}, \left(\mathbf{x}^T\mathbf{z}\right)_{2,2}, \dots, \left(\mathbf{x}^T\mathbf{z}\right)_{d,d}\right) \in [-1,1]^d$ (note the overload of notation which should be clear by context.) Under our uniform distribution assumption on the multi-sphere, multi-dot product kernels can be decomposed as $\mathcal{K}\left(\mathbf{x}, \mathbf{z}\right) = \sum_{\mathbf{k},\mathbf{j}} \lambda_{\mathbf{k}} Y_{\mathbf{k},\mathbf{j}}\left(\mathbf{x}\right) Y_{\mathbf{k},\mathbf{j}}\left(\mathbf{z}\right)$, where $\mathbf{k}, \mathbf{j} \in \mathbb{N}^d$. $Y_{\mathbf{k},\mathbf{j}}\left(\mathbf{x}\right)$ (the eigenfunctions of $\mathcal{K}$) are products of spherical harmonics in $\mathbb{S}^{C_0-1}$, $Y_{\mathbf{k},\mathbf{j}}\left(\mathbf{x}\right) = \prod_{i=1}^{d} Y_{k_i,j_i}\left(\mathbf{x}_i\right)$ with $k_i \geq 0$, $j_i \in [N(C_0, k_i)]$, where $N(C_0, k_i)$ denotes the number of harmonics of frequency $k_i$ in $\mathbb{S}^{C_0-1}$. For $C_0 = 2$ these are products of Fourier series in a $d$-dimensional torus. Note that the eigenvalues $\lambda_{\mathbf{k}}$ are non-negative and do not depend on $\mathbf{j}$.

Using Mercer's Representation of RKHSs (Kanagawa et al., 2018), we have that the RKHS $\mathcal{H}_{\mathcal{K}}$ of $\mathcal{K}$ is

$$\mathcal{H}_{\mathcal{K}} := \left\{ f = \sum_{\mathbf{k},\mathbf{j}} \alpha_{\mathbf{k},\mathbf{j}} Y_{\mathbf{k},\mathbf{j}} \;\middle|\; \|f\|_{\mathcal{H}_{\mathcal{K}}}^2 = \sum_{\mathbf{k},\mathbf{j}} \frac{\alpha_{\mathbf{k},\mathbf{j}}^2}{\lambda_{\mathbf{k}}} < \infty \right\}.$$

For multi-dot product kernels the normalized kernel simplifies to $\overline{\mathcal{K}}\left(\mathbf{t}\right) = \frac{\mathcal{K}(\mathbf{t})}{\mathcal{K}(\vec{\mathbf{1}})}$, where $\vec{\mathbf{1}} = (1,\dots,1)^T \in \mathbb{R}^d$. $\mathcal{K}$ and $\overline{\mathcal{K}}$ thus differ by a constant, and so they share the same eigenfunctions and their eigenvalues differ by a multiplicative constant.

## 4 Kernel Derivations

We next provide explicit formulas for ResCGPK and ResCNTK.

### 4.1 ResCGPK

Given a network $f(\mathbf{x};\theta)$, the corresponding Gaussian process kernel is defined as $\mathbb{E}_\theta\left[f(\mathbf{x};\theta)f(\mathbf{z};\theta)\right]$. Below we consider the network in Sec. 3.1, which can have either one of three heads, the equivariant head, trace or GAP. We denote the corresponding ResCGPK by $\mathcal{K}_{\mathrm{Eq}}^{(L)}$, $\mathcal{K}_{\mathrm{Tr}}^{(L)}$ and $\mathcal{K}_{\mathrm{GAP}}^{(L)}$, where $L$ denotes the number of layers. We proceed with the following definition.

**Definition 4.1.** *Let $\mathbf{x}, \mathbf{z} \in \mathbb{R}^{C_0\times d}$ and $f$ be a residual network with $L$ layers. For every $1 \leq l \leq L$ (and $i$ is an arbitrary choice of channel) denote by*

$$\Sigma_{j,j'}^{(l)}\left(\mathbf{x}, \mathbf{z}\right) := \mathbb{E}_\theta\left[g_{ij}^{(l)}\left(\mathbf{x}\right) g_{ij'}^{(l)}\left(\mathbf{z}\right)\right], \tag{4}$$

$$\Lambda_{j,j'}^{(l)}\left(\mathbf{x}, \mathbf{z}\right) = \begin{pmatrix} \Sigma_{j,j}^{(l)}\left(\mathbf{x}, \mathbf{x}\right) & \Sigma_{j,j'}^{(l)}\left(\mathbf{x}, \mathbf{z}\right) \\ \Sigma_{j',j}^{(l)}\left(\mathbf{z}, \mathbf{x}\right) & \Sigma_{j',j'}^{(l)}\left(\mathbf{z}, \mathbf{z}\right) \end{pmatrix} \tag{5}$$

$$K_{j,j'}^{(l)}(\mathbf{x}, \mathbf{z}) = c_v c_w \underset{(u,v)\sim\mathcal{N}\left(0,\Lambda_{j,j'}^{(l)}(\mathbf{x},\mathbf{z})\right)}{\mathbb{E}} [\sigma(u)\sigma(v)] \tag{6}$$

$$\dot{K}_{j,j'}^{(l)}(\mathbf{x}, \mathbf{z}) = c_v c_w \underset{(u,v)\sim\mathcal{N}\left(0,\Lambda_{j,j'}^{(l)}(\mathbf{x},\mathbf{z})\right)}{\mathbb{E}} [\dot{\sigma}(u)\dot{\sigma}(v)], \tag{7}$$

*where $\dot{\sigma}$ is the derivative of the ReLU function expressed by the indicator $\mathbf{1}_{x\geq 0}$.*

Our first contribution is to give an exact formula for the ResCGPK. We refer the reader to the appendix for the precise derivation and note here some of the key ideas. We give precise formulas for $\Sigma, K$ and $\dot{K}$ and prove that

$$\mathcal{K}_{\text{Eq}}^{(L)}(\mathbf{x},\mathbf{z}) = \Sigma_{1,1}^{(1)}(\mathbf{x},\mathbf{z}) + \frac{\alpha^2}{qc_w}\sum_{l=1}^{L}\text{tr}\left(K_{\mathcal{D}_{1,1}}^{(l)}(\mathbf{x},\mathbf{z})\right).$$

This gives us the equivariant kernel, and by showing that $\mathcal{K}_{\text{Tr}}^{(L)}(\mathbf{x},\mathbf{z}) = \frac{1}{d}\sum_{j=1}^{d}\mathcal{K}_{\text{Eq}}^{(L)}(\mathbf{s}_j\mathbf{x},\mathbf{s}_j\mathbf{z})$ and $\mathcal{K}_{\text{GAP}}^{(L)}(\mathbf{x},\mathbf{z}) = \frac{1}{d^2}\sum_{j,j'=1}^{d}\mathcal{K}_{\text{Eq}}^{(L)}(\mathbf{s}_j\mathbf{x},\mathbf{s}_{j'}\mathbf{z})$ we obtain precise formulas for the trace and GAP kernels.

For clarity, we give here the case of the normalized ResCGPK with multi-sphere inputs, which we prove to simplify significantly. The full derivation for arbitrary inputs is given in Appendix A.2.

**Theorem 4.1.** *[Multi-Sphere Case] For any $\mathbf{x}, \mathbf{z} \in \mathbb{MS}(C_0, d)$ let $\mathbf{t} = \left((\mathbf{x}^T\mathbf{z})_{1,1}, \ldots, (\mathbf{x}^T\mathbf{z})_{d,d}\right)$*

$\in [-1,1]^d$. *Fixing $c_v = 2, c_w = 1$ and let $\kappa_1(u) = \frac{\sqrt{1-\rho^2}+(\pi-\cos^{-1}(u))u}{\pi}$. Then,*

$$\overline{\mathcal{K}}_{Eq}^{(1)}(\mathbf{t}) = \frac{1}{1+\alpha^2}\left(t_1 + \frac{\alpha^2}{q}\sum_{k=-\frac{q-1}{2}}^{\frac{q-1}{2}}\kappa_1(t_{1+k})\right)$$

$$\overline{\mathcal{K}}_{Eq}^{(L)}(\mathbf{t}) = \frac{1}{1+\alpha^2}\left(\overline{\mathcal{K}}_{Eq}^{(L-1)}(\mathbf{t}) + \frac{\alpha^2}{q^2}\sum_{k=-\frac{q-1}{2}}^{\frac{q-1}{2}}\sum_{k'=-\frac{q-1}{2}}^{\frac{q-1}{2}}\kappa_1\left(\overline{\mathcal{K}}_{Eq}^{(L-1)}(\mathbf{s}_{k+k'}\mathbf{t})\right)\right).$$

### 4.2 ResCNTK

For $\mathbf{x}, \mathbf{z} \in \mathbb{R}^{C_0\times d}$ and $f(\mathbf{x}; \theta)$ an $L$ layer ResNet, ResCNTK is defined as $\mathbb{E}\left[\left\langle\frac{\partial f(\mathbf{x};\theta)}{\partial\theta}, \frac{\partial f(\mathbf{z};\theta)}{\partial\theta}\right\rangle\right]$. Considering the three heads in Sec. 3.1, we denote the corresponding kernels by $\Theta_{\text{Eq}}^{(L)}, \Theta_{\text{Tr}}^{(L)}$ and $\Theta_{\text{GAP}}^{(L)}$, depending on the choice of last layer. Our second contribution is providing a formula for the ResCNTK for arbitrary inputs.

**Theorem 4.2.** *Let $\mathbf{x}, \mathbf{z} \in \mathbb{R}^{C_0\times d}$ and $f$ be a residual network with $L$ layers. Then, the ResCNTK for $f$ has the form*

$$\Theta_{Eq}^{(L)}(\mathbf{x},\mathbf{z}) = \mathcal{K}_{Eq}^{(L)}(\mathbf{x},\mathbf{z}) + \frac{\alpha^2}{q}\sum_{l=1}^{L}\sum_{p=1}^{d}P_p^{(l)}(\mathbf{x},\mathbf{z})\left(tr\left(\dot{K}_{\mathcal{D}_{p,p}}^{(l)}(\mathbf{x},\mathbf{z})\odot\Sigma_{\mathcal{D}_{p,p}}^{(l)}(\mathbf{x},\mathbf{z}) + K_{\mathcal{D}_{p,p}}^{(l)}(\mathbf{x},\mathbf{z})\right)\right),$$

*where $\forall 1 \leq j \leq d, P_j^{(L)}(\mathbf{x},\mathbf{z}) = \frac{\mathbf{1}_{j=1}}{c_w}$ and for $1 \leq l \leq L-1$,*

$$P_j^{(l)}(\mathbf{x},\mathbf{z}) = P_j^{(l+1)}(\mathbf{x},\mathbf{z}) + \frac{\alpha^2}{q^2}\sum_{k=-\frac{q-1}{2}}^{\frac{q-1}{2}}\dot{K}_{j+k,j+k}^{(l+1)}(\mathbf{x},\mathbf{z})\sum_{k'=\frac{q-1}{2}}P_{j+k+k'}^{(l+1)}(\mathbf{x},\mathbf{z}).$$

*The Tr and GAP kernels are given by $\Theta_{Tr}^{(L)}(\mathbf{x},\mathbf{z}) = \frac{1}{d}\sum_{j=1}^{d}\Theta_{Eq}^{(L)}(\mathbf{s}_j\mathbf{x},\mathbf{s}_j\mathbf{z})$ and $\Theta_{GAP}^{(L)}(\mathbf{x},\mathbf{z}) = \frac{1}{d^2}\sum_{j,j'=1}^{d^2}\Theta_{Eq}^{(L)}(\mathbf{s}_j\mathbf{x},\mathbf{s}_{j'}\mathbf{z})$.*

## 5 SPECTRAL DECOMPOSITION

$\mathcal{K}_{Eq}^{(L)}$ and $\Theta_{Eq}^{(L)}$ are multi-dot product kernels, and therefore their eigenfunctions consist of spherical harmonic products. Next, we derive bounds on their eigenvalues. We subsequently use a result due to (Geifman et al., 2022) to extend these to their trace and GAP versions.

### 5.1 ASYMPTOTIC BOUNDS

The next theorem provides upper and lower bounds on the eigenvalues of $\mathcal{K}_{Eq}^{(L)}$ or $\Theta_{Eq}^{(L)}$.

**Theorem 5.1.** *Let* $\mathbf{x}, \mathbf{z} \in \mathbb{MS}(C_0, d)$, *where* $d$ *denotes the number of pixels and* $C_0$ *denotes the number of input channels for each pixel. The eigenvalues* $\lambda_{\mathbf{k}}$ *of either* $\mathcal{K}_{Eq}^{(L)}$ *or* $\Theta_{Eq}^{(L)}$ *satisfy*

$$c_1 \prod_{i \in \mathcal{R}, k_i > 0} k_i^{-(C_0 + 2\nu_a - 3)} \leq \lambda_{\mathbf{k}} \leq c_2 \prod_{i \in \mathcal{R}, k_i > 0} k_i^{-(C_0 + 2\nu_b - 3)}$$

*where* $\nu_a = 2.5$ *and* $\nu_b$ *is* $1 + \frac{3}{2d}$ *for ResCGPK and* $1 + \frac{1}{2d}$ *for ResCNTK.* $c_1, c_2 > 0$ *are constants that depend on* $L$. *The set* $\mathcal{R}$ *denotes the receptive field, defined as the set of indices of input pixels that affect the kernel output.*

We note that these bounds are identical, up to constants, to those obtained with convolutional networks that do not include skip connections (Geifman et al., 2022), although the proof for the case of ResNet is more complex. Overall, the theorem shows that over-parameterized ResNets are biased toward low-frequency functions. In particular, with input distributed uniformly on the multi-sphere, the time required to train such a network to fit an eigenfunction with gradient descent (GD) is inversely proportional to the respective eigenvalue (Basri et al., 2020). Consequently, training a network to fit a high frequency function is polynomially slower than training the network to fit a low frequency function. Note, however, that the rate of decay of the eigenvalues depends on the number of pixels over which the target function has high frequencies. Training a target function whose high frequencies are concentrated in a few pixels is much faster than if the same frequencies are spread over many pixels. This can be seen in Figure 1, which shows for a target function of frequency $k$ in $m$ pixels, that the exponent (depicted by the slope of the lines) grows with $m$. The same behaviour is seen when the skip connections are removed. This is different from fully connected architectures, in which the decay rate of the eigenvalues depends on the dimension of the input and is invariant to the pixel spread of frequencies, see (Geifman et al., 2022) for a comparison of the eigenvalue decay for standard CNNs and FC architectures.

### 5.2 LOCALITY BIAS AND ENSEMBLE BEHAVIOR

To better understand the difference between ResNets and vanilla CNNs we next turn to a fine-grained analysis of the decay. Consider an $l$-layer CNN that is identical to our ResNet but with the skip connections removed. Let $p_i^{(l)}$ be the number of paths from input pixel $i$ to the output in the corresponding CGPK, or equivalently, the number of paths in the same CNN but in which there is only one channel in each node..

**Theorem 5.2.** *For both* $\Theta_{Eq}^{(L)}$ *or* $\mathcal{K}_{Eq}^{(L)}$ *there exist scalars* $A > 1$ *and* $c_l > 0$ *s.t. letting* $c_{\mathbf{k},l} = c_l \prod_{i=1}^{d} A^{\min(p_i^{(l)}, k_i)}$ *for every* $1 \leq l \leq L$ *and* $c_{\mathbf{k}} = \sum_{l=1}^{L} c_{\mathbf{k},l}$, *it holds that*

$$\lambda_{\mathbf{k}} \geq c_{\mathbf{k}} \prod_{\substack{i=1 \\ k_i > 0}}^{d} k_i^{-C_0 - 2}.$$

The constant $c_{\mathbf{k}}$ differs significantly from that of CNTK (without skip connections) which takes the form $c_{\mathbf{k},L}$ (Geifman et al., 2022). In particular, notice that the constants in the ResCNTK are (up to scale factors) the sum of the constants of the CNTK at depths $1, \ldots, L$. Thus, a major contribution of the paper is providing theoretical justification for the following result, observed empirically in (Veit et al., 2016): *over-parameterized ResNets act like a weighted ensemble of CNNs of various depths.* In particular, information from smaller receptive fields is propagated through the skip connections, resulting in larger eigenvalues for frequencies that correspond to smaller receptive fields.

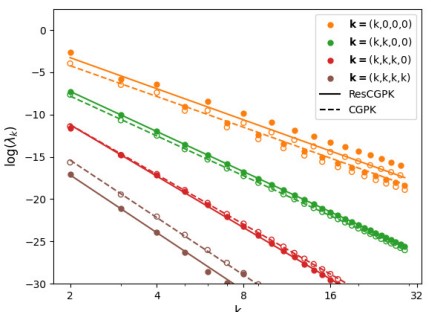
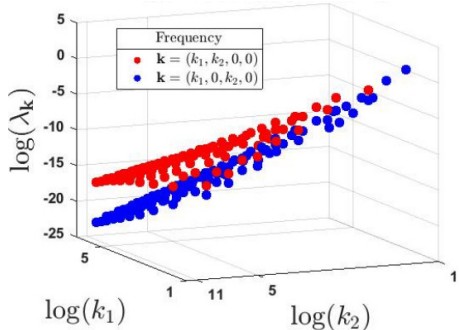

Figure 1: Left: The eigenvalues of ResCGPK (filled circles and solid lines) computed numerically for various eigenfunctions, compared to those of CGPK (empty circles and dashed lines). Here $L = 3$, $q = 2$, $d = 4$ and the output head is the equivariant one. The slopes (respectively, -5.25, -6.8, -8.5, -9.8 for CGPK and -5.3, -6.84, -8.77, -9.85 for ResCGPK) approximate the exponent for each pattern. Notice that the slope increases for eigenfunctions involving more pixels. Right: Additional eigenvalues of ResCGPK. Notice that the eigenvalues for eigenfunction involving nearby pixels (in red) are larger compared to one involving farther pixels (in blue).



| (a) ResCNTK | (b) CNTK | (c) ResNet | (d) CNN |

Figure 2: The effective receptive field of ResCNTK (left) compared to that of actual ResNet and to CNTK and CNN (i.e., no skip connections). We followed (Luo et al., 2016) in computing the ERF, where the networks are first trained on CIFAR-10. All values are re-scaled to the [0,1] interval. We used $L = 8$ in all cases.

Figure 1 shows the eigenvalues computed numerically for various frequencies, for both the CGPK and ResCGPK. Consistent with our results, eigenfunctions with high frequencies concentrated in a few pixels, e.g., $\mathbf{k} = (k, 0, 0, 0)$ have larger eigenvalues than those with frequencies spread over more pixels, e.g., $\mathbf{k} = (k, k, k, k)$. See appendix G for implementation details.

Figure 2 shows the effective receptive field (ERF) induced by ResCNTK compared to that of a network and to the kernel and network with the skip connections removed. The ERF is defined to be $\partial f^{\text{Eq}}(\mathbf{x}; \theta)/\partial \mathbf{x}$ for ResNet (Luo et al., 2016) and $\Theta^{\text{Eq}}(\mathbf{x}, \mathbf{x})/\partial \mathbf{x}$ for ResCNTK. A similar calculation is applied to CNN and CNTK. We see that residual networks and their kernels give rise to an increased locality bias (more weight at the center of the receptive field (for the equivariant architecture) or to nearby pixels (at the trace and GAP architectures).

## 5.3 EXTENSION TO $f^{\text{TR}}$ AND $f^{\text{GAP}}$

Using (Geifman et al., 2022)[Thm. 3.7], we can extend our analysis of equivariant kernels to trace and GAP kernels. In particular, for ResCNTK, the eigenfunctions of the trace kernel are a product of spherical harmonics. In addition, let $\lambda_{\mathbf{k}}$ denote the eigenvalues of $\Theta_{\text{Eq}}^{(L)}$, then the eigenvalues of $\Theta_{\text{Tr}}^{(L)}$ are $\lambda_{\mathbf{k}}^{\text{Tr}} = \frac{1}{d} \sum_{i=0}^{d-1} \lambda_{\mathbf{s}_i \mathbf{k}}$, i.e., average over all shifts of the frequency vector $\mathbf{k}$. This implies that for the trace kernel, the eigenvalues (but not the eigenfunctions) are invariant to shift. For the GAP kernel, the eigenfunctions are $\frac{1}{\sqrt{d}} \sum_{i=0}^{d-1} \mathbf{Y}_{\mathbf{s}_i \mathbf{k}, \mathbf{s}_i \mathbf{j}}$, i.e., scaled shifted sums of spherical harmonic products. These eigenfunctions are shift invariant and generally span all shift invariant functions. The eigenvalues of the GAP kernel are identical to those of the Trace kernel. The eigenfunctions and eigenvalues of the trace and GAP ResCGPK are determined in a similar way. Finally, we note that the eigenvalues for the trace and GAP kernels decay at the same rate as their equivariant counterparts,

and likewise they are biased in frequency and in locality. Moreover, while the equivariant kernel is biased to prefer functions that depend on the center of the receptive field (position biased), the trace and GAP kernels are biased to prefer functions that depend on nearby pixels.

# 6 STABILITY AT LARGE DEPTHS

## 6.1 DECAYING BALANCING PARAMETER $\alpha$

We next investigate the effects of skip connections in very deep networks. Here the setting of balancing parameter $\alpha$ between the skip and residual connection (2) plays a critical role. Previous work on residual, non-convolutional kernels Huang et al. (2020); Belfer et al. (2021) proposed to use a balancing parameter of the form $\alpha = L^{-\gamma}$ for $\gamma \in (0.5, 1]$, arguing that a decaying $\alpha$ contributes to the stability of the kernel for very deep architectures. However, below we prove that in this setting as the depth $L$ tends to infinity, ResCNTK converges to a simple dot-product, $\mathbf{k}(\mathbf{x}, \mathbf{z}) = \mathbf{x}^T \mathbf{z}$, corresponding to a 1-layer, linear neural network, which may be considered degenerate. We subsequently further elaborate on the connection between this result and previous work and provide a more comprehensive discussion in Appendix F.

**Theorem 6.1.** *Suppose $\alpha = L^{-\gamma}$ with $\gamma \in (0.5, 1]$. Then, for any $\mathbf{t} \in [-1, 1]^d$ it holds that $\overline{\Theta}_{Eq}^{(L)}(\mathbf{t}) \xrightarrow[L \to \infty]{} \overline{\Sigma}_{1,1}^{(1)}(\mathbf{t}) = t_1$ and likewise $\overline{\mathcal{K}}_{Eq}^{(L)}(\mathbf{t}) \xrightarrow[L \to \infty]{} \overline{\Sigma}_{1,1}^{(1)}(\mathbf{t}) = t_1$.*

Clearly, this limit kernel, which corresponds to a linear network with no hierarchical features if undesired. A comparison to the previous work of Huang et al. (2020); Belfer et al. (2021), which addressed residual kernels for fully connected architectures, is due here. This previous work proved that FC-ResNTK converges when $L$ tends to infinity to a two-layer FC-NTK. They however made the additional assumption that the top-most layer is not trained. This assumption turned out to be critical to their result – training the last layer yields a result analogous to ours, namely, that as $L$ tends to infinity FC-ResNTK converges to a simple dot product. Similarly, if we consider ResCNTK in which we do not train the last layer we will get that the limit kernel is the CNTK corresponding to a two-layer convolutional neural network. However, while a two-layer FC-NTK is universal, the set of functions produced by a two-layer CNTK is very limited; therefore, this limit kernel is also not desired. We conclude that the standard setting of $\alpha = 1$ is preferable for convolutional architectures.

## 6.2 THE CONDITION NUMBER OF THE RESCGPK MATRIX WITH $\alpha = 1$

Next we investigate the properties of ResCGPK when the balancing factor is set to $\alpha = 1$. For ResCGPK and CGPK and any training distribution, we use *double-constant matrices* (O'Neill, 2021) to bound the condition numbers of their kernel matrices. We further show that with any depth, the lower bound for ResCGPK matrices is lower than that of CGPK matrices (and show empirically that these bounds are close to the actual condition numbers). Lee et al. (2019); Xiao et al. (2020); Chen et al. (2021) argued that a smaller condition number of the NTK matrix implies that training the corresponding neural network with GD convergences faster. Our analysis therefore indicates that GD with ResCGPK should generally be faster than GD with CGPK. This phenomenon may partly explain the advantage of residual networks over standard convolutional architectures.

Recall that the condition number of a matrix $\boldsymbol{A} \succeq 0$ is defined as $\rho(\boldsymbol{A}) := \lambda_{\max}/\lambda_{\min}$. Consider an $n \times n$ *double-constant matrix* $\boldsymbol{B}_{\tilde{b}, b}$ that includes $\tilde{b}$ in the diagonal entries and $b$ in each off-diagonal entry. The eigenvalues of $\boldsymbol{B}_{\tilde{b}, b}$ are $\lambda_1 = \tilde{b} - b + nb$ and $\lambda_2 = ... = \lambda_n = \tilde{b} - b$. Suppose $\tilde{b} = 1, 0 < b \le 1$, then $\boldsymbol{B}_{1,b}$ is positive semi-definite and its condition number is $\rho(\boldsymbol{B}_{1,b}) = 1 + \frac{nb}{1-b}$. This condition number diverges when either $b = 1$ or $n$ tends to infinity. The following lemma relates the condition numbers of kernel matrices with that of double-constant matrices.

**Lemma 6.1.** *Let $\boldsymbol{A} \in \mathbb{R}^{n \times n}$ ($n \ge 2$) be a normalized kernel matrix with $\sum_{i \neq j} \boldsymbol{A}_{ij} \ge 0$. Let $\boldsymbol{B}(\boldsymbol{A}) = \boldsymbol{B}_{1,b}$ with $b = \frac{1}{n(n-1)} \sum_{i \neq j} \boldsymbol{A}_{ij}$ and $\epsilon = \sup_i \sum_{j \neq i} |\boldsymbol{A}_{ij} - \boldsymbol{B}(\boldsymbol{A})_{ij}|$. Then,*

1. *$\rho(\boldsymbol{B}(\boldsymbol{A})) \le \rho(\boldsymbol{A})$.*

2. *If $\epsilon < \lambda_{\min}(\boldsymbol{B}(\boldsymbol{A}))$ then $\rho(\boldsymbol{A}) \le \frac{\lambda_{\max}(\boldsymbol{B}(\boldsymbol{A})) + \epsilon}{\lambda_{\min}(\boldsymbol{B}(\boldsymbol{A})) - \epsilon}$,*

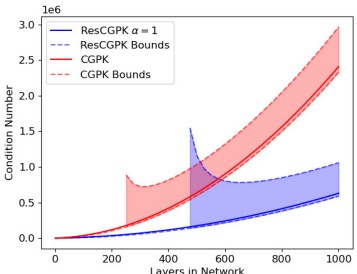

Figure 3: The condition number of ResCGPK-Tr (solid blue) as a function of depth compared to that of CGPK-Tr (solid red) and the corresponding lower and upper bounds (dashed lines) computed with $n = 100$.

*where $\lambda_{\max}$ and $\lambda_{\min}$ denote the maximal and minimal eigenvalues of $\boldsymbol{B}(\boldsymbol{A})$.*

The following theorem uses double-constant matrices to compare kernel matrices produced by ResCGPK and those produced by CGPK with no skip connections.

**Theorem 6.2.** *Let $\bar{K}^{(L)}_{ResCGPK}$ and $\bar{K}^{(L)}_{CGPK}$ respectively denote kernel matrices for the normalized trace kernels ResCGPK and CGPK of depth L. Let $\boldsymbol{B}(K)$ be a double-constant matrix defined for a matrix $K$ as in Lemma 6.1. Then,*

*1.* $\left\| \bar{K}^{(L)}_{ResCGPK} - \boldsymbol{B}\left(\bar{K}^{(L)}_{ResCGPK}\right) \right\|_1 \xrightarrow[L\to\infty]{} 0$ *and* $\left\| \bar{K}^{(L)}_{CGPK} - \boldsymbol{B}\left(\bar{K}^{(L)}_{CGPK}\right) \right\|_1 \xrightarrow[L\to\infty]{} 0.$

*2.* $\rho\left(\boldsymbol{B}\left(\bar{K}^{(L)}_{ResCGPK}\right)\right) \xrightarrow[L\to\infty]{} \infty$ *and* $\rho\left(\boldsymbol{B}\left(\bar{K}^{(L)}_{CGPK}\right)\right) \xrightarrow[L\to\infty]{} \infty.$

*3.* $\exists L_0 \in \mathbb{N}$ *s.t.* $\forall L \geq L_0$, $\rho\left(\boldsymbol{B}\left(\bar{K}^{(L)}_{ResCGPK}\right)\right) < \rho\left(\boldsymbol{B}\left(\bar{K}^{(L)}_{CGPK}\right)\right).$

The theorem establishes that, while the condition numbers of both $\boldsymbol{B}\left(\bar{K}^{(L)}_{ResCGPK}\right)$ and $\boldsymbol{B}\left(\bar{K}^{(L)}_{CGPK}\right)$ diverge as $L \to \infty$, the condition number of $\boldsymbol{B}\left(\bar{K}^{(L)}_{ResCGPK}\right)$ is smaller than that of $\boldsymbol{B}\left(\bar{K}^{(L)}_{CGPK}\right)$ for all $L > L_0$. ($L_0$ is the minimal $L$ s.t. the entries of the double constant matrices are non-negative. We notice in practice that $L_0 \approx 2$.) We can therefore use Lemma 6.1 to derive approximate bounds for the condition numbers obtained with ResCGPK and CGPK. Figure 3 indeed shows that the condition number of the CGPK matrix diverges faster than that of ResCGPK and is significantly larger at any finite depth $L$. The approximate bounds, particularly the lower bounds, closely match the actual condition numbers produced by the kernels. (We note that with training sampled from a uniform distribution on the multi-sphere, the upper bound can be somewhat improved. In this case, the constant vector is the eigenvector of maximal eigenvalue for both $\boldsymbol{A}$ and $\boldsymbol{B}(\boldsymbol{A})$, and thus the rows of $\boldsymbol{A}$ sum to the same value, yielding $\rho(\boldsymbol{A}) \leq \frac{\lambda_{\max}(\boldsymbol{B}(\boldsymbol{A}))}{\lambda_{\min}(\boldsymbol{B}(\boldsymbol{A}))-\epsilon}$ with $\epsilon = \frac{1}{n} \|\boldsymbol{A} - \boldsymbol{B}(\boldsymbol{A})\|_1$. We used this upper bound in our plot in Figure 3.)

To the best of our knowledge, this is the first paper that establishes a relationship between skip connections and the condition number of the kernel matrix.

## 7 CONCLUSION

We derived formulas for the Gaussian process and neural tangent kernels associated with convolutional residual networks, analyzed their spectra, and provided bounds on their implied condition numbers. Our results indicate that over-parameterized residual networks are subject to both frequency and locality bias, and that they can be trained faster than standard convolutional networks. In future work, we hope to gain further insight by tightening our bounds. We further intend to apply our analysis of the condition number of kernel matrices to characterize the speed of training in various other architectures.

## ACKNOWLEDGEMENT

This research was partially supported by the Israeli Council for Higher Education (CHE) via the Weizmann Data Science Research Center and by research grants from the Estate of Tully and Michele Plesser and the Anita James Rosen Foundation.

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

APPENDIX

Below we provide derivations and proofs for our paper.

## A  DERIVATION OF RESCGPK

In this section, we derive explicit formulas for ResCGPK. We begin with a few preliminaries. As in (Jacot et al., 2018), we assume the network parameters $\theta$ are initialized with a standard Gaussian distribution, $\theta \sim \mathcal{N}(0, I)$. Therefore, at initialization, for every pair of parameters, $\theta_i, \theta_j$,

$$\mathbb{E}[\theta_i \cdot \theta_j] = \delta_{ij}. \tag{8}$$

We note that Lee et al. (2019) proved the convergence of a network with this initialization to its NTK. For a vector $\mathbf{v}$, we use the notation $v_*$ to denote an entry of $\mathbf{v}$ with arbitrary index.

### A.1  A CLOSED FORMULA FOR $K$ AND $\dot{K}$

For $u \in [-1, 1]$, let $\kappa_0(u) = \frac{\pi - \cos^{-1}(u)}{\pi}$ and $\kappa_1(u) = \frac{\sqrt{1-u^2} + (\pi - \cos^{-1}(u))u}{\pi}$ be the arc-cosine kernels defined in Cho & Saul (2009). Daniely et al. (2016) showed that

$$K^{(l)}(\mathbf{x}, \mathbf{z}) = \frac{c_v c_w}{2} \sqrt{\Sigma^{(l)}(\mathbf{x}, \mathbf{x}) \Sigma^{(l)}(\mathbf{z}, \mathbf{z})} \kappa_1\left(\overline{\Sigma}^{(l)}(\mathbf{x}, \mathbf{z})\right) \tag{9}$$

and

$$\dot{K}^{(l)}(\mathbf{x}, \mathbf{z}) = \frac{c_v c_w}{2} \kappa_0\left(\overline{\Sigma}^{(l)}(\mathbf{x}, \mathbf{z})\right), \tag{10}$$

where $K^{(l)}$ and $\dot{K}^{(l)}$ are defined in (6) and (7) and $c_v, c_w$ are defined in Sec. 3.1.

### A.2  RESCGPK DERIVATION

**Theorem A.1.** *For an $L$-layer neural network $f$ and $\mathbf{x}, \mathbf{z} \in \mathbb{R}^{C_0 \times d}$,*

$$\Sigma_{j,j'}^{(1)}(\mathbf{x}, \mathbf{z}) = \frac{1}{C_0}\left(\mathbf{x}^T \mathbf{z}\right)_{j,j'}$$

$$\Sigma_{j,j'}^{(2)}(\mathbf{x}, \mathbf{z}) = \frac{c_w}{q} tr\left(\Sigma_{\mathcal{D}_{j,j'}}^{(1)}(\mathbf{x}, \mathbf{z})\right) + \frac{\alpha^2}{q^2} \sum_{k=-\frac{q-1}{2}}^{\frac{q-1}{2}} tr\left(K_{\mathcal{D}_{j+k,j'+k}}^{(1)}(\mathbf{x}, \mathbf{z})\right).$$

*For $3 \le l \le L$,*

$$\Sigma_{j,j'}^{(l)}(\mathbf{x}, \mathbf{z}) = \Sigma_{j,j'}^{(l-1)}(\mathbf{x}, \mathbf{z}) + \frac{\alpha^2}{q^2} \sum_{k=-\frac{q-1}{2}}^{\frac{q-1}{2}} tr\left(K_{\mathcal{D}_{j+k,j'+k}}^{(l-1)}(\mathbf{x}, \mathbf{z})\right).$$

*Finally, for the output layer*

$$\mathcal{K}_{Eq}^{(L)}(\mathbf{x}, \mathbf{z}) = \Sigma_{1,1}^{(1)}(\mathbf{x}, \mathbf{z}) + \frac{\alpha^2}{qc_w} \sum_{l=1}^{L} tr\left(K_{\mathcal{D}_{1,1}}^{(l)}(\mathbf{x}, \mathbf{z})\right)$$

$$\mathcal{K}_{Tr}^{(L)}(\mathbf{x}, \mathbf{z}) = \frac{1}{d} \sum_{j=1}^{d} \mathcal{K}_{Eq}^{(L)}(\mathbf{s}_j \mathbf{x}, \mathbf{s}_j \mathbf{z})$$

$$\mathcal{K}_{GAP}^{(L)}(\mathbf{x}, \mathbf{z}) = \frac{1}{d^2} \sum_{j,j'=1}^{d} \mathcal{K}_{Eq}^{(L)}\left(\mathbf{s}_j \mathbf{x}, \mathbf{s}_{j'} \mathbf{z}\right).$$

*Proof.* We begin by deriving a formula for $\Sigma^{(l)(\mathbf{x},\mathbf{z})}$. The case of $l = 1$ is shown in Lemma (A.1). For $2 < l \leq L$, the strategy is to express $\mathbb{E}\left[g_{ij}^{(l)}(\mathbf{x})\, g_{ij'}^{(l)}(\mathbf{z})\right]$ using $\mathbb{E}\left[f_{c,l}^{(l-1)}(\mathbf{x})\, f_{c,l'}^{(l-1)}(\mathbf{z})\right]$ and vice versa (which we can do using Lemma A.2). This way we derive an expression for $\mathbb{E}\left[f_{c,l}^{(l-1)}(\mathbf{x})\, f_{c,l'}^{(l-1)}(\mathbf{z})\right]$ in Lemma (A.3) and subsequently get:

$$\mathbb{E}\left[g_{ij}^{(l)}(\mathbf{x})\, g_{ij'}^{(l)}(\mathbf{z})\right] \underset{\text{Lemma A.2}}{=} \frac{c_w}{qC_{l-1}} \sum_{k=-\frac{q-1}{2}}^{\frac{q-1}{2}} \sum_{c=1}^{C_{l-1}} \mathbb{E}\left[f_{c,j+k}^{(l-1)}(\mathbf{x})\, f_{c,j'+k}^{(l-1)}(\mathbf{z})\right]$$

$$\underset{\text{Lemma A.3}}{=} \frac{c_w}{qC_{l-1}} \sum_{c=1}^{C_{l-1}} \underbrace{\left( \sum_{k=-\frac{q-1}{2}}^{\frac{q-1}{2}} \mathbb{E}\left[f_{c,j+k}^{(l-2)}(\mathbf{x})\, f_{c,j'+k}^{(l-2)}(\mathbf{z})\right] \right)}_{\text{Denote by } A} +$$

$$+ \alpha^2 \frac{c_v c_w}{q^2 C_{l-1}} \sum_{c=1}^{C_{l-1}} \left( \sum_{k=-\frac{q-1}{2}}^{\frac{q-1}{2}} \sum_{k'=-\frac{q-1}{2}}^{\frac{q-1}{2}} \mathbb{E}\left[ \sigma\left(g^{(l-1)}(\mathbf{x})\right)_{c,j+k+k'} \sigma\left(g^{(l-1)}(\mathbf{z})\right)_{c,j'+k+k'} \right] \right)$$

$$= A + \alpha^2 \frac{c_v c_w}{q^2} \sum_{k,k'=-\frac{q-1}{2}}^{\frac{q-1}{2}} \mathbb{E}\left[ \sigma\left(g^{(l-1)}(\mathbf{x})\right)_{*,j+k+k'} \sigma\left(g^{(l-1)}(\mathbf{z})\right)_{*,j'+k+k'} \right]$$

$$= A + \frac{\alpha^2}{q^2} \sum_{k=-\frac{q-1}{2}}^{\frac{q-1}{2}} \mathrm{tr}\left( K_{\mathcal{D}_{j+k,j'+k}}^{(l-1)}(\mathbf{x},\mathbf{z}) \right).$$

If $l > 2$ then using Lemma (A.2) we obtain $A = \mathbb{E}\left[g_{ij}^{(l-1)}(\mathbf{x})\, g_{ij'}^{(l-1)}(\mathbf{z})\right] = \Sigma_{j,j'}^{(l-1)}(\mathbf{x},\mathbf{z})$. Otherwise if $l = 2$ then using Lemma (A.1) we obtain $A = \frac{c_w}{q}\mathrm{tr}\left(\Sigma_{\mathcal{D}_{j,j'}}^{(1)}(\mathbf{x},\mathbf{z})\right)$.

We leave the three output layers to lemma A.4 □

**Lemma A.1.**
$$\mathbb{E}\left[f_{ij}^{(0)}(\mathbf{x})\, f_{ij'}^{(0)}(\mathbf{z})\right] = \mathbb{E}\left[g_{ij}^{(1)}(\mathbf{x})\, g_{ij'}^{(1)}(\mathbf{z})\right] = \frac{1}{C_0}\left(\mathbf{x}^T\mathbf{z}\right)_{j,j'}.$$

*Proof.* For $g^{(1)}$ we have:

$$\Sigma_{j,j'}^{(1)}(\mathbf{x},\mathbf{z}) = \mathbb{E}\left[g_{ij}^{(1)}(\mathbf{x})\, g_{ij'}^{(1)}(\mathbf{z})\right] = \frac{1}{C_0} \sum_{l,l'=1}^{C_0} \mathbb{E}\left[\mathbf{W}_{1,l,i}^{(1)}\mathbf{x}_{l,j}\mathbf{W}_{1,l',i}^{(1)}\mathbf{z}_{l',j'}\right]$$

$$= \frac{1}{C_0} \sum_{l,l'=1}^{C_0} \mathbb{E}\left[\mathbf{W}_{1,l,i}^{(1)}\mathbf{W}_{1,l',i}^{(1)}\right] \mathbb{E}\left[\mathbf{x}_{l,j}\mathbf{z}_{l'j'}\right] \underset{(8)}{=} \frac{1}{C_0} \sum_{l=1}^{C_0} \mathbf{x}_{l,j}\mathbf{z}_{l,j'} = \frac{1}{C_0}\left(\mathbf{x}^T\mathbf{z}\right)_{j,j'}.$$

For $f^{(1)}$ the proof is analogous, by simply replacing $\mathbf{W}^{(1)}$ with $\mathbf{V}^{(0)}$. □

**Lemma A.2.** $\forall 2 \leq l \leq L, 1 \leq i, i' \leq C_l, 1 \leq j, j' \leq d$, we have:

$$\mathbb{E}\left[g_{ij}^{(l)}(\mathbf{x})\, g_{i'j'}^{(l)}(\mathbf{z})\right] = \delta_{i,i'} \frac{c_w}{qC_{l-1}} \sum_{k=-\frac{q-1}{2}}^{\frac{q-1}{2}} \sum_{c=1}^{C_{l-1}} \mathbb{E}\left[f_{c,j+k}^{(l-1)}(\mathbf{x})\, f_{c,j'+k}^{(l-1)}(\mathbf{z})\right].$$

*Proof.*

$$\mathbb{E}\left[g_{ij}^{(l)}(\mathbf{x})\, g_{i'j'}^{(l)}(\mathbf{z})\right] = \frac{c_w}{qC_{l-1}} \mathbb{E}\left[ \left[\sum_{c=1}^{C_{l-1}} \mathbf{W}_{:,c,i}^{(l)} * f_c^{(l-1)}(\mathbf{x})\right]_j \left[\sum_{c'=1}^{C_{l-1}} \mathbf{W}_{:,c',i'}^{(l)} * f_{c'}^{(l-1)}(\mathbf{z})\right]_{j'} \right]$$

$$= \frac{c_w}{qC_{l-1}} \sum_{c,c'=1}^{C_l} \sum_{k,k'=-\frac{q-1}{2}}^{\frac{q-1}{2}} \mathbb{E}\left[\mathbf{W}^{(l)}_{k+\frac{q+1}{2},c,i} \mathbf{W}^{(l)}_{k'+\frac{q+1}{2},c',i'}\right] \mathbb{E}\left[f^{(l-1)}_{c,j+k}(\mathbf{x}) f^{(l-1)}_{c',j'+k'}(\mathbf{z})\right]$$

$$\underset{Equation 8}{=} \delta_{i,i'} \frac{c_w}{qC_{l-1}} \sum_{k=-\frac{q-1}{2}}^{\frac{q-1}{2}} \sum_{c=1}^{C_{l-1}} \mathbb{E}\left[f^{(l-1)}_{c,j+k}(\mathbf{x}) f^{(l-1)}_{c,j'+k}(\mathbf{z})\right] \tag{11}$$

$\square$

**Lemma A.3.** $\forall 1 \leq l \leq L, 1 \leq i, i' \leq C_l, 1 \leq j, j' \leq d$, we have:

$$\mathbb{E}\left[f^{(l)}_{ij}(\mathbf{x}) f^{(l)}_{ij'}(\mathbf{z})\right] = \mathbb{E}\left[f^{(l-1)}_{ij}(\mathbf{x}) f^{(l-1)}_{ij'}(\mathbf{z})\right] + \alpha^2 \frac{c_v}{q} \sum_{k=-\frac{q-1}{2}}^{\frac{q-1}{2}} \mathbb{E}\left[\sigma\left(g^{(l)}(\mathbf{x})\right)_{*,j+k} \sigma\left(g^{(l)}(\mathbf{z})\right)_{*,j'+k}\right].$$

*Proof.* Using the definition for $f^{(l)}_{ij}$ (Def. 2), we get the expression:

$$\mathbb{E}\left[f^{(l)}_{ij}(\mathbf{x}) f^{(l)}_{ij'}(\mathbf{z})\right] = \mathbb{E}\left[f^{(l-1)}_{ij}(\mathbf{x}) f^{(l-1)}_{ij'}(\mathbf{z})\right] +$$

$$\underbrace{\mathbb{E}\left[f^{(l-1)}_{ij}(\mathbf{x}) \left[\sum_{c=1}^{C_l} \mathbf{V}^{(l)}_{:,c,i} * \sigma\left(g^{(l)}_c(\mathbf{z})\right)\right]_{j'}\right]}_{\text{Denote by } B_1} + \underbrace{\mathbb{E}\left[f^{(l-1)}_{ij'}(\mathbf{z}) \left[\sum_{c=1}^{C_l} \mathbf{V}^{(l)}_{:,c,i} * \sigma\left(g^{(l)}_c(\mathbf{x})\right)\right]_{j}\right]}_{\text{Denote by } B_2} +$$

$$+ \alpha^2 \frac{c_v}{q} \frac{1}{C_l} \underbrace{\mathbb{E}\left[\left[\sum_{c=1}^{C_l} \mathbf{V}^{(l)}_{:,c,i} * \sigma\left(g^{(l)}_c(\mathbf{x})\right)\right]_{j} \left[\sum_{c'=1}^{C_l} \mathbf{V}^{(l)}_{:,c',i} * \sigma\left(g^{(l)}_{c'}(\mathbf{z})\right)\right]_{j'}\right]}_{\text{Denote by } A}.$$

We will deal with this expression in parts. First, for $B_1$ we get:

$$B_1 = \sum_{k=-\frac{q-1}{2}}^{\frac{q-1}{2}} \sum_{c=1}^{C_l} \mathbb{E}\left[f^{(l-1)}_{ij}(\mathbf{x}) \mathbf{V}^{(l)}_{k+\frac{q+1}{2},c,i} \sigma\left(g^{(l)}_c(\mathbf{z})\right)_{c,j'+k}\right] = 0,$$

where the rightmost equality follows from (8) and the fact that $\mathbb{E}[\mathbf{V}] = 0$ mean in expectation in every index. Analogously, we also get $B_2 = 0$. Opening $A$ and using Equation (8) we get:

$$A = \frac{1}{C_l} \sum_{k'=-\frac{q-1}{2}}^{\frac{q-1}{2}} \sum_{k=-\frac{q-1}{2}}^{\frac{q-1}{2}} \sum_{c=1}^{C_l} \sum_{c'=1}^{C_l} \mathbb{E}\left[\mathbf{V}_{k+\frac{q+1}{2},c,i} \sigma\left(g^{(l)}(\mathbf{x})\right)_{c,j+k} \mathbf{V}_{k'+\frac{q+1}{2},c',i} \sigma\left(g^{(l)}(\mathbf{z})\right)_{c',j'+k'}\right]$$

$$\underset{(8)}{=} \sum_{k=-\frac{q-1}{2}}^{\frac{q-1}{2}} \frac{1}{C_l} \sum_{c=1}^{C_l} \mathbb{E}\left[\sigma\left(g^{(l)}(\mathbf{x})\right)_{c,j+k} \sigma\left(g^{(l)}(\mathbf{z})\right)_{c,j'+k}\right]$$

$$= \sum_{k=-\frac{q-1}{2}}^{\frac{q-1}{2}} \mathbb{E}\left[\sigma\left(g^{(l)}(\mathbf{x})\right)_{*,j+k} \sigma\left(g^{(l)}(\mathbf{z})\right)_{*,j'+k}\right].$$

Overall, we obtain

$$\mathbb{E}\left[f^{(l)}_{ij}(\mathbf{x}) f^{(l)}_{ij'}(\mathbf{z})\right] = \mathbb{E}\left[f^{(l-1)}_{ij}(\mathbf{x}) f^{(l-1)}_{ij'}(\mathbf{z})\right] + \alpha^2 \frac{c_v}{q} \sum_{k=-\frac{q-1}{2}}^{\frac{q-1}{2}} \mathbb{E}\left[\sigma\left(g^{(l)}(\mathbf{x})\right)_{*,j+k} \sigma\left(g^{(l)}(\mathbf{z})\right)_{*,j'+k}\right].$$

$\square$

**Lemma A.4.**

$$\mathcal{K}_{Eq}^{(L)}\left(\mathbf{x},\mathbf{z}\right) = \Sigma_{1,1}^{(0)}\left(\mathbf{x},\mathbf{z}\right) + \frac{\alpha^2}{qc_w}\sum_{l=1}^{L} tr\left(K_{\mathcal{D}_{1,1}}^{(l)}\left(\mathbf{x},\mathbf{z}\right)\right).$$

*Proof.* We start by proving the case for $\mathcal{K}_{Eq}^{(L)}$. Observe that:

$$\mathbb{E}_\theta\left[f^{Eq}\left(\mathbf{x};\theta\right)f^{Eq}\left(\mathbf{z};\theta\right)\right] = \frac{1}{C_L}\sum_{i,i'=1}^{C_L}\mathbb{E}\left[W_i^{Eq}f_{i,1}^{(L)}\left(\mathbf{x}\right)W_{i'}^{Eq}f_{i',1}^{(L)}\left(\mathbf{z}\right)\right]$$

$$= \frac{1}{C_L}\sum_{i,i'=1}^{C_L}\delta_{ii'}\mathbb{E}\left[f_{i,1}^{(L)}\left(\mathbf{x}\right)f_{i',1}^{(L)}\left(\mathbf{z}\right)\right] = \frac{1}{C_L}\sum_{i=1}^{C_L}\mathbb{E}\left[f_{i,1}^{(L)}\left(\mathbf{x}\right)f_{i,1}^{(L)}\left(\mathbf{z}\right)\right].$$

Applying Lemma A.3 recursively we obtain

$$\mathbb{E}_\theta\left[f^{Eq}\left(\mathbf{x};\theta\right)f^{Eq}\left(\mathbf{z};\theta\right)\right]$$

$$= \frac{1}{C_L}\sum_{i=1}^{C_L}\left(\mathbb{E}\left[f_{i,1}^{(L-1)}\left(\mathbf{x}\right)f_{i,1}^{(L-1)}\left(\mathbf{z}\right)\right] + \alpha^2\frac{c_v}{q}\sum_{k=-\frac{q-1}{2}}^{\frac{q-1}{2}}\mathbb{E}\left[\sigma\left(g^{(L)}\left(\mathbf{x}\right)\right)_{*,1+k}\sigma\left(g^{(L)}\left(\mathbf{z}\right)\right)_{*,1+k}\right]\right)$$

$$= \frac{1}{C_L}\sum_{i=1}^{C_L}\left(\mathbb{E}\left[f_{i,1}^{(L-1)}\left(\mathbf{x}\right)f_{i,1}^{(L-1)}\left(\mathbf{z}\right)\right] + \frac{\alpha^2}{qc_w}tr\left(K_{\mathcal{D}_{1,1}}^{(L)}\left(\mathbf{x},\mathbf{z}\right)\right)\right)$$

$$\underset{\text{Lemma A.3}}{=} \frac{1}{C_L}\sum_{i=1}^{C_L}\left(\mathbb{E}\left[f_{i,1}^{(L-2)}\left(\mathbf{x}\right)f_{i,1}^{(L-2)}\left(\mathbf{z}\right)\right] + \frac{\alpha^2}{qc_w}\sum_{l=L-1}^{L}tr\left(K_{\mathcal{D}_{1,1}}^{(l)}\left(\mathbf{x},\mathbf{z}\right)\right)\right).$$

Applying Lemma A.3 recursively for all layers we obtain

$$\mathbb{E}_\theta\left[f^{Eq}\left(\mathbf{x};\theta\right)f^{Eq}\left(\mathbf{z};\theta\right)\right] = \frac{1}{C_L}\sum_{i=1}^{C_L}\left(\mathbb{E}\left[f_{i,1}^{(0)}\left(\mathbf{x}\right)f_{i,1}^{(0)}\left(\mathbf{z}\right)\right] + \frac{\alpha^2}{qc_w}\sum_{l=1}^{L}tr\left(K_{\mathcal{D}_{1,1'}}^{(l)}\left(\mathbf{x},\mathbf{z}\right)\right)\right)$$

$$= \Sigma_{1,1}^{(1)}\left(\mathbf{x},\mathbf{z}\right) + \frac{\alpha^2}{qc_w}\sum_{l=1}^{L}tr\left(K_{\mathcal{D}_{1,1}}^{(l)}\left(\mathbf{x},\mathbf{z}\right)\right).$$

$\square$

**Lemma A.5.**

$$\mathcal{K}_{Tr}^{(L)}\left(\mathbf{x},\mathbf{z}\right) = \frac{1}{d}\sum_{j=1}^{d}\mathcal{K}_{Eq}^{(L)}\left(\mathbf{s}_j\mathbf{x},\mathbf{s}_j\mathbf{z}\right)$$

$$\mathcal{K}_{GAP}^{(L)}\left(\mathbf{x},\mathbf{z}\right) = \frac{1}{d^2}\sum_{j,j'=1}^{d}\mathcal{K}_{Eq}^{(L)}\left(\mathbf{s}_j\mathbf{x},\mathbf{s}_{j'}\mathbf{z}\right).$$

*Proof.* For $f^{Tr}$ we have:

$$\mathcal{K}_{Tr}^{(L)}\left(\mathbf{x},\mathbf{z}\right) = \mathbb{E}_\theta\left[f^{Tr}\left(\mathbf{x};\theta\right)f^{Tr}\left(\mathbf{z};\theta\right)\right] = \frac{1}{C_L d}\sum_{j,j'=1}^{d}\sum_{i,i'=1}^{C_L}\mathbb{E}\left[W_{i,j}^{Tr}f_{i,j}^{(L)}\left(\mathbf{x}\right)W_{i',j'}^{Tr}f_{i',j'}^{(L)}\left(\mathbf{z}\right)\right]$$

$$= \frac{1}{d}\sum_{j=1}^{d}\left(\frac{1}{C_L}\sum_{i=1}^{C_L}\mathbb{E}\left[f_{i,j}^{(L)}\left(\mathbf{x}\right)f_{i,j}^{(L)}\left(\mathbf{z}\right)\right]\right) = \frac{1}{d}\sum_{j=1}^{d}\left(\frac{1}{C_L}\sum_{i=1}^{C_L}\mathbb{E}\left[f_{i,1}^{(L)}\left(\mathbf{s}_j\mathbf{x}\right)f_{i,1}^{(L)}\left(\mathbf{s}_j\mathbf{z}\right)\right]\right),$$

where the part inside the parentheses was shown in the previous Lemma A.4 to equal $\mathcal{K}_{Eq}^{(L)}\left(\mathbf{s}_j\mathbf{x},\mathbf{s}_j\mathbf{z}\right)$.

For $f^{\text{GAP}}$ we analogously obtain

$$\mathcal{K}_{\text{GAP}}^{(L)}(\mathbf{x},\mathbf{z}) = \mathbb{E}_\theta\left[f^{\text{GAP}}(\mathbf{x};\theta)f^{\text{GAP}}(\mathbf{z};\theta)\right] = \frac{1}{C_L\frac{d^2}{s^2}}\sum_{j,j'=1}^{d}\sum_{i,i'=1}^{C_L}\mathbb{E}\left[W_i^{\text{GAP}}f_{i,j}^{(L)}(\mathbf{x})W_{i'}^{\text{GAP}}f_{i',j'}^{(L)}(\mathbf{z})\right]$$

$$= \frac{1}{d^2}\sum_{j,j'=1}^{d}\left(\frac{1}{C_L}\sum_{i=1}^{C_L}\mathbb{E}\left[f_{i,j}^{(L)}(\mathbf{x})f_{i,j'}^{(L)}(\mathbf{z})\right]\right) = \frac{1}{d^2}\sum_{j,j'=1}^{d}\left(\frac{1}{C_L}\sum_{i=1}^{C_L}\mathbb{E}\left[f_{i,1}^{(L)}(\mathbf{s}_j\mathbf{x})f_{i,1}^{(L)}(\mathbf{s}_{j'}\mathbf{z})\right]\right),$$

from which the claim follows.

$\square$

### A.3 FORMULAS FOR MULTISPHERE INPUT: PROOF OF THEOREM 4.1

**Lemma A.6.** *For an $L$-layer ResNet $f$ and $\mathbf{x} \in \mathbb{MS}(C_0, d)$, and for every $1 \le l \le L, 1 \le j, j' \le d$*

$$\Sigma_{j,j'}^{(l)}(\mathbf{x},\mathbf{x}) = \begin{cases} \frac{1}{C_0} & l = 1 \\ \frac{\left(1+\alpha^2\frac{c_v c_w}{2}\right)^{l-2}\left(2c_w+\alpha^2 c_v c_w\right)}{2C_0} & l \ge 2. \end{cases}$$

*Proof.* We prove this by induction using the formula in Theorem (A.1). For $l = 1$, since by assumption $\|\mathbf{x}_i\| = 1$, for every $i$ we get that $\mathbf{x}^T\mathbf{x}$ is the $d \times d$ matrix with 1 in every entry. Therefore,

$$\Sigma_{j,j'}^{(1)}(\mathbf{x},\mathbf{x}) = \frac{1}{C_0}\left(\mathbf{x}^T\mathbf{x}\right)_{j,j'} = \frac{1}{C_0}.$$

Similarly, for $l = 2$:

$$\Sigma_{j,j'}^{(2)}(\mathbf{x},\mathbf{x}) = \frac{c_w}{q}\text{tr}\left(\Sigma_{\mathcal{D}_{j,j'}}^{(1)}(\mathbf{x},\mathbf{x})\right) + \frac{\alpha^2}{q^2}\sum_{k=-\frac{q-1}{2}}^{\frac{q-1}{2}}\text{tr}\left(K_{\mathcal{D}_{j+k,j'+k}}^{(1)}(\mathbf{x},\mathbf{x})\right).$$

We can plug in the induction hypothesis, and express $K$ as in (9), obtaining

$$\Sigma_{j,j'}^{(2)}(\mathbf{x},\mathbf{x}) = \frac{c_w}{C_0} + \frac{\alpha^2}{q^2}\sum_{k,k'=-\frac{q-1}{2}}^{\frac{q-1}{2}}\frac{c_v c_w}{2}\kappa_1(1)\sqrt{\Sigma_{j+k+k',j'+k+k'}^{(1)}(\mathbf{x},\mathbf{x})\Sigma_{j+k+k',j'+k+k'}^{(1)}(\mathbf{x},\mathbf{x})}$$

$$= \frac{c_w}{C_0} + \alpha^2\frac{c_v c_w}{2}N_1 = \frac{2c_w + \alpha^2 c_v c_w}{2C_0},$$

where we used the fact that $\kappa_1(1) = 1$. The proof for $l \ge 3$ is analogous:

$$\Sigma_{j,j}^{(l)}(\mathbf{x},\mathbf{x}) = \Sigma_{j,j}^{(l-1)}(\mathbf{x},\mathbf{x}) + \frac{\alpha^2}{q^2}\sum_{k=-\frac{q-1}{2}}^{\frac{q-1}{2}}\text{tr}\left(K_{\mathcal{D}_{j+k,j+k}}^{(l-1)}(\mathbf{x},\mathbf{x})\right)$$

$$= N_{L-1} + \frac{\alpha^2}{q^2}\sum_{k,k'=-\frac{q-1}{2}}^{\frac{q-1}{2}}\frac{c_v c_w}{2}N_{L-1}\kappa_1(1) = \left(1+\alpha^2\frac{c_v c_w}{2}\right)N_{L-1}$$

$$= \frac{\left(1+\alpha^2\frac{c_v c_w}{2}\right)^{l-2}\left(c_w + \alpha^2 c_v c_w\right)}{2C_0}.$$

$\square$

**Lemma A.7.** *For any $L \in \mathbb{N}$ let $N_L$ be the value of $\Sigma_{j,j}^{(l)}(\mathbf{x},\mathbf{x})$ from Lemma A.6. Let $\mathbf{x}, \mathbf{z} \in \mathbb{MS}(C_0, d)$, then*

$$\mathcal{K}_{Eq}^{(L)}(\mathbf{x},\mathbf{z}) = \Sigma_{1,1}^{(1)}(\mathbf{x},\mathbf{z}) + \frac{\alpha^2 c_v}{2q}\sum_{l=1}^{L}N_l\text{tr}\left(\kappa_1\left(\overline{\Sigma}_{\mathcal{D}_{1,1}}^{(L)}(\mathbf{x},\mathbf{z})\right)\right).$$

*Proof.* Let $L \in \mathbb{N}$. We know from Theorem A.1 that

$$\mathcal{K}_{\text{Eq}}^{(L)}(\mathbf{x}, \mathbf{z}) = \Sigma_{1,1}^{(1)}(\mathbf{x}, \mathbf{z}) + \frac{\alpha^2}{qc_w} \sum_{l=1}^{L} \text{tr}\left(K_{\mathcal{D}_{1,1}}^{(l)}(\mathbf{x}, \mathbf{z})\right).$$

By expressing $K$ as in (9) and using Lemma A.6 we get

$$\mathcal{K}_{\text{Eq}}^{(L)}(\mathbf{x}, \mathbf{z}) = \Sigma_{1,1}^{(1)}(\mathbf{x}, \mathbf{z}) + \frac{\alpha^2 c_v}{2q} \sum_{l=1}^{L} N_l \text{tr}\left(\kappa_1\left(\overline{\Sigma}_{\mathcal{D}_{1,1}}^{(L)}(\mathbf{x}, \mathbf{z})\right)\right).$$

$\square$

**Corollary A.1.** *Fix $c_v = 2, c_w = 1$ then*

$$\overline{\mathcal{K}}_{Eq}^{(L)}(\mathbf{x}, \mathbf{z}) = \frac{C_0}{(1+\alpha^2)^L} \mathcal{K}_{Eq}^{(L)}(\mathbf{x}, \mathbf{z}).$$

*Proof.* Using the previous lemma and the fact that $\kappa(1) = 1$,

$$\overline{\mathcal{K}}_{\text{Eq}}^{(L)}(\mathbf{x}, \mathbf{x}) = \Sigma_{1,1}^{(1)}(\mathbf{x}, \mathbf{x}) + \alpha^2 \sum_{l=1}^{L} N_l = \frac{1}{C_0}\left(1 + \alpha^2\left(1 + (1+\alpha^2)\sum_{l=0}^{L-2}(1+\alpha^2)^l\right)\right)$$

$$= \frac{1}{C_0}\left(1 + \alpha^2\left(1 + (1+\alpha^2)\frac{1 - (1+\alpha^2)^{L-1}}{1 - (1+\alpha^2)}\right)\right) = \frac{1}{C_0}\left(1 + \alpha^2\left(1 + \frac{(1+\alpha^2)^L - (1+\alpha^2)}{\alpha^2}\right)\right)$$

$$= \frac{(1+\alpha^2)^L}{C_0}$$

.
$\square$

**Proposition A.1.** *For any $L \in \mathbb{N}$, let $\mathbf{x}, \mathbf{z} \in \mathbb{MS}(C_0, d)$, and denote $\mathbf{t} = \left((\mathbf{x}^T\mathbf{z})_{1,1}, (\mathbf{x}^T\mathbf{z})_{2,2}, \ldots, (\mathbf{x}^T\mathbf{z})_{d,d}\right) \in [-1, 1]^d$. Suppose that $\alpha$ is fixed for all networks of different depths, then:*

$$\mathcal{K}_{Eq}^{(1)}(\mathbf{t}) = \frac{1}{C_0}\mathbf{t}_1 + \frac{\alpha^2}{qc_wC_0}\sum_{k=-\frac{q-1}{2}}^{\frac{q-1}{2}} \kappa_1(\mathbf{t}_{1+k})$$

*and*

$$\mathcal{K}_{Eq}^{(L)}(\mathbf{t}) = \mathcal{K}_{Eq}^{(L-1)}(\mathbf{t}) + \tilde{N}_L \sum_{k,k'=-\frac{q-1}{2}}^{\frac{q-1}{2}} \kappa_1\left(\frac{\mathcal{K}_{Eq}^{(L-1)}(\mathbf{s}_{k+k'}\mathbf{t})}{N_L}\right)$$

*where $N_L$ be the value of $\Sigma_{j,j}^{(l)}(\mathbf{x}, \mathbf{x})$ from Lemma A.6*

*Proof.* If $\alpha$ is fixed, then for all $1 \leq l \leq L$ the definition of $K^{(l)}(\mathbf{x}, \mathbf{z})$ does not depend on $L$ (just that $l < L$). Therefore, using Lemma A.7 we obtain

$$\mathcal{K}_{\text{Eq}}^{(L)}(\mathbf{x}, \mathbf{z}) = \mathcal{K}_{\text{Eq}}^{(L-1)}(\mathbf{x}, \mathbf{z}) + \frac{\alpha^2 c_v}{2q} N_L \text{tr}\left(\kappa_1\left(\overline{\Sigma}_{\mathcal{D}_{1,1}}^{(L)}(\mathbf{x}, \mathbf{z})\right)\right).$$

To simplify this further, observe first that a direct consequence of Theorem A.1 is that $\forall L \geq 2$

$$\Sigma_{1,1}^{(L)}(\mathbf{x}, \mathbf{z}) = \frac{c_w}{q} \sum_{k=-\frac{q-1}{2}}^{\frac{q-1}{2}} \mathcal{K}_{\text{Eq}}^{(L-1)}(\mathbf{s}_k\mathbf{x}, \mathbf{s}_k\mathbf{z}).$$

We therefore get

$$\mathcal{K}_{\text{Eq}}^{(L)}(\mathbf{x}, \mathbf{z}) = \mathcal{K}_{\text{Eq}}^{(L-1)}(\mathbf{x}, \mathbf{z}) + \frac{\alpha^2 c_v c_w}{2q^2} N_L \sum_{k,k'=-\frac{q-1}{2}}^{\frac{q-1}{2}} \kappa_1\left(\overline{\mathcal{K}}_{\text{Eq}}^{(L-1)}(\mathbf{s}_{k+k'}\mathbf{x}, \mathbf{s}_{k+k'}\mathbf{z})\right).$$

$\square$

**Corollary A.2.** *For any* $\mathbf{x}, \mathbf{z} \in \mathbb{MS}(C_0, d)$, *let* $\mathbf{t} = \left( \left( \mathbf{x}^T \mathbf{z} \right)_{1,1}, \left( \mathbf{x}^T \mathbf{z} \right)_{2,2}, \ldots, \left( \mathbf{x}^T \mathbf{z} \right)_{d,d} \right) \in [-1,1]^d$. *Fix* $c_v = 2, c_w = 1$ *and some* $\alpha$ *for all neural networks. Then,*

$$\overline{\mathcal{K}}_{Eq}^{(1)}(\mathbf{t}) = \frac{1}{1+\alpha^2} \left( \mathbf{t}_1 + \frac{\alpha^2}{q} \sum_{k=-\frac{q-1}{2}}^{\frac{q-1}{2}} \kappa_1 \left( \mathbf{t}_{1+k} \right) \right)$$

$$\overline{\mathcal{K}}_{Eq}^{(L)}(\mathbf{t}) = \frac{1}{1+\alpha^2} \left( \overline{\mathcal{K}}_{Eq}^{(L-1)}(\mathbf{t}) + \frac{\alpha^2}{q^2} \sum_{k,k'=-\frac{q-1}{2}}^{\frac{q-1}{2}} \kappa_1 \left( \overline{\mathcal{K}}_{Eq}^{(L-1)} \left( \mathbf{s}_{k+k'} \mathbf{t} \right) \right) \right).$$

# B   DERIVATION OF RESCNTK

## B.1   REWRITING THE NEURAL NETWORK

The convolution of $\mathbf{w} \in \mathbb{R}^q$ with a vector $\mathbf{v} \in \mathbb{R}^d$ can be rewritten as:

$$[\mathbf{w} * \mathbf{v}]_i = \sum_{j=1}^{q} [\mathbf{w}]_j [\mathbf{v}]_{i+j-\frac{q+1}{2}}.$$

Therefore, let $\varphi(\mathbf{v}) \in \mathbb{R}^{q \times d}$ be $[\varphi(\mathbf{v})]_{ij} := [\mathbf{v}]_{i+j-\frac{q+1}{2}}$. Then we can rewrite the above as:

$$\mathbf{w} * \mathbf{v} = \left( \mathbf{w}^T \varphi(\mathbf{v}) \right)^T = \varphi(\mathbf{v})^T \mathbf{w}.$$

Using this definition, if we instead have $\mathbf{w} \in \mathbb{R}^{q \times c}$ then:

$$\boldsymbol{A}_{ij} := [\mathbf{w}_{:,i} * \mathbf{v}]_j = \left[ \varphi(\mathbf{v})^T \mathbf{w}_{:,i} \right]_j = \left[ \varphi(\mathbf{v})^T \mathbf{w} \right]_{ji} \implies \boldsymbol{A} = \mathbf{w}^T \varphi(\mathbf{v}).$$

Lastly, if we instead have $\mathbf{w} \in \mathbb{R}^{q \times c \times c'}$ and $v \in \mathbb{R}^{c' \times d}$ then:

$$\boldsymbol{A}_{ij} := \sum_{k=1}^{c'} [\mathbf{w}_{:,k,i} * \mathbf{v}_k]_j = \sum_{k=1}^{c'} \mathbf{w}_{:,k,:}^T \varphi(\mathbf{v}_k).$$

We can now rewrite the network architecture as:

$$f^{(0)}(\mathbf{x}) = \frac{1}{\sqrt{C_0}} \left( \mathbf{V}_1^{(0)} \right)^T \mathbf{x} \tag{12}$$

$$g^{(1)}(\mathbf{x}) = \frac{1}{\sqrt{C_0}} \left( \mathbf{W}_1^{(1)} \right)^T \mathbf{x} \tag{13}$$

$$f^{(l)}(\mathbf{x}) = f^{(l-1)}(\mathbf{x}) + \alpha \sqrt{\frac{c_v}{qC_l}} \sum_{j=1}^{C_l} \left( \mathbf{V}_{:,j,:}^{(l)} \right)^T \varphi \left( \sigma \left( g_j^{(l)}(\mathbf{x}) \right) \right) \quad l = 1, \ldots, L \tag{14}$$

$$g^{(l)}(\mathbf{x}) = \sqrt{\frac{c_w}{qC_{l-1}}} \sum_{j=1}^{C_{l-1}} \left( \mathbf{W}_{:,j,:}^{(l)} \right)^T \varphi \left( f_j^{(l-1)}(\mathbf{x}) \right) \quad l = 2, \ldots, L, \tag{15}$$

and as before we have an output layer that corresponds to one of: $f^{\text{Eq}}, f^{\text{Tr}}$, or $f^{\text{GAP}}$.

## B.2   NOTATIONS

We use a numerator layout notation, i.e., for $y \in \mathbb{R}$, $\boldsymbol{A} \in \mathbb{R}^{m \times n}$ we denote:

$$\mathbb{R}^{n \times m} \ni \frac{\partial y}{\partial \boldsymbol{A}} = \begin{bmatrix} \frac{\partial y}{\partial \boldsymbol{A}_{11}} & \frac{\partial y}{\partial \boldsymbol{A}_{21}} & \cdots & \frac{\partial y}{\partial \boldsymbol{A}_{m1}} \\ \frac{\partial y}{\partial \boldsymbol{A}_{12}} & \frac{\partial y}{\partial \boldsymbol{A}_{22}} & \cdots & \frac{\partial y}{\partial \boldsymbol{A}_{m2}} \\ \vdots & \vdots & \ddots & \vdots \\ \frac{\partial y}{\partial \boldsymbol{A}_{1n}} & \frac{\partial y}{\partial \boldsymbol{A}_{2n}} & \cdots & \frac{\partial y}{\partial \boldsymbol{A}_{mn}} \end{bmatrix}.$$

Also, let $\boldsymbol{J}_{mn}^{ij}$ be the $m \times n$ matrix with 1 in coordinate $(i,j)$ and 0 elsewhere. we write $\boldsymbol{J}^{ij}$ when $m, n$ are clear by context. Also, let $\boldsymbol{J}^{\mathcal{D}_i} = \sum_m \delta_{m \in \mathcal{D}_i} \boldsymbol{J}_{d,q}^{m, m-i+\frac{q+1}{2}}$.

### B.3 CHAIN RULE REMINDER

Recall that by the chain rule we know that we can decompose the Jacobian of a composition of functions $h \circ \psi(\mathbf{v})$ as:

$$J_{h \circ \psi}(\mathbf{v}) = J_h(\psi(\mathbf{v})) J_\psi(\mathbf{v}).$$

When $h, \psi$ are scalar functions we can write

$$\frac{\partial h}{\partial \mathbf{v}} = \frac{\partial h}{\partial \psi} \frac{\partial \psi}{\partial \mathbf{v}}.$$

However, if $h$ is scalar valued and $\psi(\mathbf{v})$ is a matrix we have

$$\left[\frac{\partial h \circ \psi}{\partial \mathbf{v}}\right]_{ij} = \frac{\partial h \circ \psi}{\partial \mathbf{v}_{ji}} = \sum_{p,q} \frac{\partial h}{\partial \psi_{pq}} \frac{\partial \psi_{pq}}{\partial \mathbf{v}_{ji}} = \mathrm{tr}\left(\frac{\partial h}{\partial \psi} \frac{\partial \psi}{\partial \mathbf{v}_{ji}}\right).$$

As such, the following definitions will come in handy:

**Definition B.1.** $\forall 1 \leq l \leq L, 1 \leq j, j' \leq d$, let

$$b^{(l)}(\mathbf{x}) := \left(\frac{\partial f^{Eq}(\mathbf{x}; \theta)}{\partial f^{(l)}(\mathbf{x})}\right)^T, \quad \Pi_{j,j'}^{(l)}(\mathbf{x}, \mathbf{z}) := \frac{1}{c_w} \mathbb{E}\left[\left(b^{(l)^T}(\mathbf{x}) b^{(l)}(\mathbf{z})\right)_{j,j'}\right].$$

*Notice that $f^{(l)}(\mathbf{x}) \in \mathbb{R}^{C_l \times d} \implies b^{(l)}(\mathbf{x}) \in \mathbb{R}^{C_l \times d}$.*

**Remark.** *There is a slight abuse of notation in the definition of $b^{(l)}$. By Lemma B.4 $b^{(L+1)}$ only depends on the weights of the last layer. Therefore, by the recursive formula for $b^{(l)}$ in Lemma B.3 and plugging in Lemma B.5 we get that $b^{(l)}$ can be written using only $W^{(l)}, \ldots, W^{(L)}, W^{Eq}, V^{(l)}, \ldots, V^{(L)}$ and $\dot\sigma\left(g^{(l+1)}(\mathbf{x})\right), \ldots, \dot\sigma\left(g^{(L)}(\mathbf{x})\right)$, where the latter are indicator functions that are always multiplied by some of $W^{(l)}, \ldots, W^{(L)}, W^{Eq}, V^{(l)}, \ldots, V^{(L)}$. It is easy to see now that for any $l' \leq l$, $\mathbb{E}\left[b_{ij}^{(l)}(\mathbf{x}) f_{i'j'}^{(l')}(\mathbf{x})\right] = 0 = \mathbb{E}\left[b_{ij}^{(l)}(\mathbf{x})\right] \mathbb{E}\left[f_{i'j'}^{(l')}(\mathbf{x})\right]$ and as a result, $b^{(l)}$ is uncorrelated with $f^{(0)}, \ldots, f^{(l)}, g^{(1)}, \ldots, g^{(l)}, \mathbf{x}$ and $\mathbf{z}$*

### B.4 PROOF OF THEOREM 4.2 IN THE MAIN TEXT

We start with a lemma that relates the trace and GAP ResCNTK to the equivariant kernel.

**Lemma B.1.** *For an $L$ layer ResNet,*

$$\Theta_{Tr}^{(L)}(\mathbf{x}, \mathbf{z}) = \frac{1}{d} \sum_{j=1}^{d} \Theta_{Eq}^{(L)}(\mathbf{s}_j \mathbf{x}, \mathbf{s}_j \mathbf{z})$$

*and*

$$\Theta_{GAP}^{(L)}(\mathbf{x}, \mathbf{z}) = \frac{1}{d^2} \sum_{j,j'=1}^{d^2} \Theta_{Eq}^{(L)}\left(\mathbf{s}_j \mathbf{x}, \mathbf{s}_{j'} \mathbf{z}\right).$$

*Proof.*

$$\mathbb{E}\left[\left\langle \frac{\partial f^{\mathrm{Tr}}(\mathbf{x}; \theta)}{\partial \theta}, \frac{\partial f^{\mathrm{Tr}}(\mathbf{z}; \theta)}{\partial \theta}\right\rangle\right] = \frac{1}{C_L d} \sum_{i,i'=1}^{C_L} \sum_{j,j'=1}^{d} \mathbb{E}\left[\left\langle \frac{\partial \mathbf{W}_{ij}^{\mathrm{Tr}} f_{ij}^{(L)}(\mathbf{x})}{\partial \theta}, \frac{\partial \mathbf{W}_{i'j'}^{\mathrm{Tr}} f_{i'j'}^{(L)}(\mathbf{z})}{\partial \theta}\right\rangle\right]$$

$$\underset{(8)}{=} \frac{1}{d} \sum_{j=1}^{d} \mathbb{E}\left[\left\langle \frac{\partial f^{\mathrm{Eq}}(\mathbf{s}_{j-1} \mathbf{x}; \theta)}{\partial \theta}, \frac{\partial f^{\mathrm{Eq}}(\mathbf{s}_{j-1} \mathbf{z}; \theta)}{\partial \theta}\right\rangle\right].$$

Similarly for GAP:

$$\mathbb{E}\left[\left\langle \frac{\partial f^{\mathrm{GAP}}(\mathbf{x}; \theta)}{\partial \theta}, \frac{\partial f^{\mathrm{GAP}}(\mathbf{z}; \theta)}{\partial \theta}\right\rangle\right] = \frac{1}{C_L d^2} \sum_{i,i'=1}^{C_L} \sum_{j,j'=1}^{d} \mathbb{E}\left[\left\langle \frac{\partial \mathbf{W}_i^{\mathrm{GAP}} f_{ij}^{(L)}(\mathbf{x})}{\partial \theta}, \frac{\partial \mathbf{W}_{i'}^{\mathrm{GAP}} f_{i'j'}^{(L)}(\mathbf{z})}{\partial \theta}\right\rangle\right]$$

$$\underset{(8)}{=} \frac{1}{d^2} \sum_{j,j'=1}^{d} \mathbb{E}\left[\left\langle \frac{\partial f^{\mathrm{Eq}}(\mathbf{s}_{j-1}\mathbf{x};\theta)}{\partial \theta}, \frac{\partial f^{\mathrm{Eq}}(\mathbf{s}_{j-1}\mathbf{z};\theta)}{\partial \theta}\right\rangle\right]$$

. $\qquad\qquad\qquad\qquad\qquad\qquad\qquad\qquad\qquad\qquad\qquad\qquad\qquad$ $\square$

We now return to the main proof. Using the lemma above, it remains to prove the theorem for $f^{\mathrm{Eq}}$

**Proposition B.1.** *Theorem (4.2) holds for the case of $f = f^{Eq}$.*

*Proof.* By linearity of the derivative operation and expectation we can rewrite:

$$\Theta_{\mathrm{Eq}}^{(L)}(\mathbf{x}, \mathbf{z}) = \mathbb{E}\left[\left\langle \frac{\partial f(\mathbf{x};\theta)}{\partial \theta}, \frac{\partial f(\mathbf{z};\theta)}{\partial \theta}\right\rangle\right]$$

$$= \mathbb{E}\left[\left\langle \frac{\partial f(\mathbf{x};\theta)}{\partial \mathbf{W}^{(\mathrm{Eq})}}, \frac{\partial f(\mathbf{z};\theta)}{\partial \mathbf{W}^{(\mathrm{Eq})}}\right\rangle\right] + \sum_{l=1}^{L}\mathbb{E}\left[\left\langle \frac{\partial f(\mathbf{x};\theta)}{\partial \mathbf{W}^{(l)}}, \frac{\partial f(\mathbf{z};\theta)}{\partial \mathbf{W}^{(l)}}\right\rangle\right] + \mathbb{E}\left[\left\langle \frac{\partial f(\mathbf{x};\theta)}{\partial \mathbf{V}^{(l)}}, \frac{\partial f(\mathbf{z};\theta)}{\partial \mathbf{V}^{(l)}}\right\rangle\right].$$

We deal with each term separately, starting with the first term.

$$\mathbb{E}\left[\left\langle \frac{\partial f(\mathbf{x};\theta)}{\partial \mathbf{W}^{\mathrm{Eq}}}, \frac{\partial f(\mathbf{z};\theta)}{\partial \mathbf{W}^{\mathrm{Eq}}}\right\rangle\right] = \frac{1}{C_L}\mathbb{E}\left[\left\langle f_{:,1}^{(L)}(\mathbf{x}), f_{:,1}^{(L)}(\mathbf{z})\right\rangle\right] = \mathcal{K}_{\mathrm{Eq}}^{(L)}(\mathbf{x}, \mathbf{z}).$$

Next, to handle $\mathbb{E}\left[\left\langle \frac{\partial f(\mathbf{x};\theta)}{\partial \mathbf{V}^{(l)}}, \frac{\partial f(\mathbf{z};\theta)}{\partial \mathbf{V}^{(l)}}\right\rangle\right]$, observe that $\forall 1 \le l \le L, 1 \le i \le C_l$ we can express $\frac{\partial f(\mathbf{x};\theta)}{\partial \mathbf{V}_{:,i,:}^{(l)}}$ as follows:

$$\frac{\partial f(\mathbf{x};\theta)}{\partial \mathbf{V}_{:,i,:}^{(l)}} \underset{\mathrm{Lemma}\,B.2}{=} \alpha\sqrt{\frac{c_v}{qC_l}}\left(\frac{\partial f(\mathbf{x};\theta)}{\partial f^{(l)}(\mathbf{x})}\right)^{T}\varphi^{T}\left(\sigma\left(g_i^{(l)}(\mathbf{x})\right)\right)$$

$$= \alpha\sqrt{\frac{c_v}{qC_l}}b^{(l)}(\mathbf{x})\varphi^{T}\left(\sigma\left(g_i^{(l)}(\mathbf{x})\right)\right).$$

Notice that Lemma B.7 implies that the conditions of Lemma B.6 are satisfied. Therefore,

$$\mathbb{E}\left[\left\langle \frac{\partial f(\mathbf{x};\theta)}{\partial \mathbf{V}_{:,i,:}^{(l)}}, \frac{\partial f(\mathbf{z};\theta)}{\partial \mathbf{V}_{:,i:,}^{(l)}}\right\rangle\right] = \alpha^2\frac{c_v}{qC_l}\sum_{p=1}^{d}\Pi_{p,p}^{(l)}(\mathbf{x}, \mathbf{z})\,\mathbb{E}\left[\left(\varphi^{T}\left(\sigma\left(g_i^{(l)}(\mathbf{x})\right)\right)\varphi\left(\sigma\left(g_i^{(l)}(\mathbf{z})\right)\right)\right)_{pp}\right]$$

$$= \alpha^2\frac{c_v c_w}{qC_l}\sum_{p=1}^{d}\Pi_{p,p}^{(l)}(\mathbf{x}, \mathbf{z})\sum_{k=-\frac{q-1}{2}}^{\frac{q-1}{2}}\mathbb{E}\left[\left(\sigma\left(g_{i,p+k}^{(l)}(\mathbf{x})\right)\right)\sigma\left(g_{i,p+k}^{(l)}(\mathbf{z})\right)\right]$$

$$= \frac{\alpha^2}{qC_l}\sum_{p=1}^{d}\Pi_{p,p}^{(l)}(\mathbf{x}, \mathbf{z})\,\mathrm{tr}\left(K_{\mathcal{D}_{p,p}}^{(l)}\right).$$

Note that as this does not depend on $i$. We therefore obtain

$$\mathbb{E}\left[\left\langle \frac{\partial f(\mathbf{x};\theta)}{\partial \mathbf{V}^{(l)}}, \frac{\partial f(\mathbf{z};\theta)}{\partial \mathbf{V}^{(l)}}\right\rangle\right] = \sum_{i=1}^{C_l}\mathbb{E}\left[\left\langle \frac{\partial f(\mathbf{x};\theta)}{\partial \mathbf{V}_{:,i,:}^{(l)}}, \frac{\partial f(\mathbf{z};\theta)}{\partial \mathbf{V}_{:,i:,}^{(l)}}\right\rangle\right] = \frac{\alpha^2}{q}\sum_{p=1}^{d}\Pi_{p,p}^{(l)}(\mathbf{x}, \mathbf{z})\,\mathrm{tr}\left(K_{\mathcal{D}_{p,p}}^{(l)}\right).$$

The next term we deal with is $\mathbb{E}\left[\left\langle \frac{\partial f(\mathbf{x};\theta)}{\partial \mathbf{W}^{(l)}}, \frac{\partial f(\mathbf{z};\theta)}{\partial \mathbf{W}^{(l)}}\right\rangle\right]$. Using Lemma B.9 we have

$$\mathbb{E}\left[\left\langle \frac{\partial f(\mathbf{x};\theta)}{\partial \mathbf{W}^{(l)}}, \frac{\partial f(\mathbf{z};\theta)}{\partial \mathbf{W}^{(l)}}\right\rangle\right] = \frac{\alpha^2}{q}\sum_{p=1}^{d}\Pi_{p,p}^{(l)}(\mathbf{x}, \mathbf{z})\,\mathrm{tr}\left(\dot{K}_{\mathcal{D}_{p,p}}^{(l)}(\mathbf{x}, \mathbf{z}) \odot \Sigma_{\mathcal{D}_{p,p}}^{(l)}(\mathbf{x}, \mathbf{z})\right).$$

Putting these together we obtain

$$\Theta_{\text{Eq}}^{(L)}(\mathbf{x}, \mathbf{z}) = \mathbb{E}\left[\left\langle \frac{\partial f(\mathbf{x};\theta)}{\partial \theta}, \frac{\partial f(\mathbf{z};\theta)}{\partial \theta}\right\rangle\right]$$

$$= \mathbb{E}\left[\left\langle \frac{\partial f(\mathbf{x};\theta)}{\partial \mathbf{W}^{(\text{Eq})}}, \frac{\partial f(\mathbf{z};\theta)}{\partial \mathbf{W}^{(\text{Eq})}}\right\rangle\right] + \sum_{l=1}^{L}\mathbb{E}\left[\left\langle \frac{\partial f(\mathbf{x};\theta)}{\partial \mathbf{W}^{(l)}}, \frac{\partial f(\mathbf{z};\theta)}{\partial \mathbf{W}^{(l)}}\right\rangle\right] + \mathbb{E}\left[\left\langle \frac{\partial f(\mathbf{x};\theta)}{\partial \mathbf{V}^{(l)}}, \frac{\partial f(\mathbf{z};\theta)}{\partial \mathbf{V}^{(l)}}\right\rangle\right]$$

$$= \mathcal{K}_{\text{Eq}}^{(L)}(\mathbf{x}, \mathbf{z}) + \frac{\alpha^2}{q}\sum_{l=1}^{L}\sum_{p=1}^{d}\Pi_{p,p}^{(l)}(\mathbf{x}, \mathbf{z})\left(\text{tr}\left(\dot{K}_{\mathcal{D}_{p,p}}^{(l)}(\mathbf{x}, \mathbf{z}) \odot \Sigma_{\mathcal{D}_{p,p}}^{(l)}(\mathbf{x}, \mathbf{z}) + K_{\mathcal{D}_{p,p}}^{(l)}(\mathbf{x}, \mathbf{z})\right)\right).$$

Finally, we provide a formula for of $\Pi$ to Lemma B.8, and denoting $P_j = \Pi_{j,j}$ completes the proof. $\qquad\square$

**Lemma B.2.** *Let $\psi$ a real valued function and $\mathbf{X}, \mathbf{A}$ matrices. If $\partial\psi\left(h\left(\mathbf{X}^T\mathbf{A}\right)\right)$ is well defined then $\frac{\partial\psi(h(\mathbf{X}^T\mathbf{A}))}{\partial\mathbf{X}} = \left(\frac{\partial\psi}{\partial(\mathbf{X}^T\mathbf{A})}\right)^T\mathbf{A}^T = \sum_{s,t}\frac{\partial\psi}{\partial h(\mathbf{X}^T\mathbf{A})_{s,t}}\frac{\partial h(\mathbf{X}^T\mathbf{A})_{s,t}}{(\mathbf{X}^T\mathbf{A})}\mathbf{A}^T$.*

*Proof.* First, by the linearity of derivatives we get that

$$\frac{\partial\left(\mathbf{X}^T\mathbf{A}\right)_{ij}}{\partial\mathbf{X}_{nm}} = \sum_{k}\frac{\partial\mathbf{X}_{ki}\mathbf{A}_{kj}}{\partial\mathbf{X}_{nm}} = \delta_{im}\mathbf{A}_{nj}.$$

Using the chain rule we get

$$\left[\frac{\partial\psi}{\partial\mathbf{X}}\right]_{mn} = \sum_{i,j}\frac{\partial\psi}{\partial\left(\mathbf{X}^T\mathbf{A}\right)_{ij}}\frac{\partial\left(\mathbf{X}^T\mathbf{A}\right)_{ij}}{\partial\mathbf{X}_{nm}} = \sum_{j}\frac{\partial\psi}{\partial\left(\mathbf{X}^T\mathbf{A}\right)_{mj}}\mathbf{A}_{nj}$$

$$= \sum_{j}\mathbf{A}_{nj}\left[\frac{\partial\psi}{\partial\left(\mathbf{X}^T\mathbf{A}\right)}\right]_{jm} = \left[\left(\mathbf{A}\frac{\partial\psi}{\partial\left(\mathbf{X}^T\mathbf{A}\right)}\right)^T\right]_{nm}$$

$$\implies \frac{\partial\psi}{\partial\mathbf{X}} = \left(\frac{\partial\psi}{\partial\left(\mathbf{X}^T\mathbf{A}\right)}\right)^T\mathbf{A}^T = \sum_{s,t}\frac{\partial\psi}{\partial h\left(\mathbf{X}^T\mathbf{A}\right)_{s,t}}\frac{\partial h\left(\mathbf{X}^T\mathbf{A}\right)_{s,t}}{\left(\mathbf{X}^T\mathbf{A}\right)}\mathbf{A}^T.$$

$\qquad\square$

**Lemma B.3.** $\forall 2 \le l \le L$,

$$b^{(l-1)}(\mathbf{x}) = \sum_{m=1}^{C_{l-1}}\sum_{n=1}^{d}b_{mn}^{(l)}(\mathbf{x})\left(\frac{\partial f_{mn}^{(l)}(\mathbf{x})}{\partial f^{(l-1)}(\mathbf{x})}\right)^T.$$

*Proof.* By the definition of $b^{(l-1)}(\mathbf{x})$ we have

$$b^{(l-1)}(\mathbf{x}) = \left(\frac{\partial f(\mathbf{x};\theta)}{\partial f^{(l-1)}(\mathbf{x})}\right)^T = \left(\sum_{m=1}^{C_{l-1}}\sum_{n=1}^{d}\frac{\partial f(\mathbf{x};\theta)}{\partial f_{mn}^{(l)}(\mathbf{x})}\frac{\partial f_{mn}^{(l)}(\mathbf{x})}{\partial f^{(l-1)}(\mathbf{x})}\right)^T$$

$$= \sum_{m=1}^{C_{l-1}}\sum_{n=1}^{d}b_{mn}^{(l)}(\mathbf{x})\left(\frac{\partial f_{mn}^{(l)}(\mathbf{x})}{\partial f^{(l-1)}(\mathbf{x})}\right)^T.$$

$\qquad\square$

**Lemma B.4.**

$$\mathbb{E}\left[\left(b^{(L)^T}(\mathbf{x})b^{(L)}(\mathbf{z})\right)_{j,j'}\right] = \mathbf{1}_{j=j'=1}.$$

*Proof.*

$$b_{:,j}^{(L)}(\mathbf{x}) = \frac{1}{\sqrt{C_L}}\left(\frac{\partial}{\partial f_{:,j}^{(L)}(\mathbf{x})}W^{\text{Eq}}f_{:,1}^{(L)}(\mathbf{x})\right)^T = \delta_{j,1}\frac{1}{\sqrt{C_L}}W^{\text{Eq}}.$$

Therefore,

$$\mathbb{E}\left[\left(b^{(L)T}(\mathbf{x})\,b^{(L)}(\mathbf{z})\right)_{j,j'}\right] = \sum_{k=1}^{C_L}\mathbb{E}\left[b_{kj}^{(L)}(\mathbf{x})\,b_{kj'}^{(L)}(\mathbf{z})\right] = \frac{\delta_{j,1}\delta_{j',1}}{C_L d}\sum_{k=1}^{C_L}\mathbb{E}\left[W_k^{\text{Eq}}W_{k'}^{\text{Eq}}\right] = \delta_{j,1}\delta_{j',1}.$$

$\square$

**Lemma B.5.** $\forall 2 \le l \le L$,

$$\frac{\partial f_{mn}^{(l)}(\mathbf{x})}{\partial f^{(l-1)}(\mathbf{x})} = \boldsymbol{J}^{nm} + \frac{\alpha}{q}\sqrt{\frac{c_w c_v}{C_l C_{l-1}}}\sum_{j=1}^{C_l}\sum_{k=1}^{q}\mathbf{V}_{k,j,m}^{(l)}\dot{\sigma}\left(g_{j,n+k-\frac{q+1}{2}}^{(l)}(\mathbf{x})\right)\boldsymbol{J}^{\mathcal{D}_{n+k-\frac{q+1}{2}}}\mathbf{W}_{:,:,j}^{(l)}.$$

*Proof.* by the definition of $f^{(l)}$ we have

$$f^{(l)}(\mathbf{x}) = f^{(l-1)}(\mathbf{x}) + \frac{\alpha}{q}\sqrt{\frac{c_w c_v}{C_l C_{l-1}}}\sum_{j=1}^{C_l}\mathbf{V}_{:,j,:}^{(l)T}\,\varphi\left(\sigma\left(g_j^{(l)}(\mathbf{x})\right)\right).$$

Taking a derivative of $f_{mn}^{(l)}$ w.r.t. $f^{(l-1)}$ we obtain

$$\frac{\partial f_{mn}^{(l)}(\mathbf{x})}{\partial f^{(l-1)}(\mathbf{x})} = \boldsymbol{J}^{nm} + \frac{\alpha}{q}\sqrt{\frac{c_w c_v}{C_l C_{l-1}}}\sum_{j=1}^{C_l}\frac{\partial}{\partial f^{(l-1)}(\mathbf{x})}\underbrace{\left(\left(\mathbf{V}_{:,j,:}^{(l)}\right)^T\varphi\left(\sigma\left(g_j^{(l)}(\mathbf{x})\right)\right)\right)_{mn}}_{\text{denote by }y_j}.$$

To simplify this, notice first that the derivative can be expressed as

$$\frac{\partial}{\partial f^{(l-1)}(\mathbf{x})}y_j = \sum_{k=1}^{q}\mathbf{V}_{k,j,m}^{(l)}\frac{\partial}{\partial f^{(l-1)}(\mathbf{x})}\varphi_{kn}\left(\sigma\left(g_j^{(l)}(\mathbf{x})\right)\right) = \sum_{k=1}^{q}\mathbf{V}_{k,j,m}^{(l)}\frac{\partial}{\partial f^{(l-1)}(\mathbf{x})}\sigma\left(g_{j,n+k-\frac{q+1}{2}}^{(l)}(\mathbf{x})\right).$$

Using the chain rule we can express the derivative of $\sigma\left(g_{j,n+k-\frac{q+1}{2}}^{(l)}(\mathbf{x})\right)$ as follows:

$$\left[\frac{\sigma\left(g_{j,n+k-\frac{q+1}{2}}^{(l)}(\mathbf{x})\right)}{\partial f^{(l-1)}(\mathbf{x})}\right]_{m'n'} = \dot{\sigma}\left(g_{j,n+k-\frac{q+1}{2}}^{(l)}(\mathbf{x})\right)\cdot\frac{\partial\sum_{j'=1}^{C_{l-1}}\left\langle\mathbf{W}_{:,j',j}^{(l)},f_{j',\mathcal{D}_{n+k-\frac{q+1}{2}}}^{(l-1)}(\mathbf{x})\right\rangle}{\partial f_{n'm'}^{(l-1)}(\mathbf{x})}$$

$$= \dot{\sigma}\left(g_{j,n+k-\frac{q+1}{2}}^{(l)}(\mathbf{x})\right)\sum_{j'=1}^{C_{l-1}}\delta_{n'j'}\mathbf{1}_{m'\in\mathcal{D}_{n+k-\frac{q+1}{2}}}\mathbf{W}_{m'-n-k+q+1,j',j}^{(l)}$$

$$= \dot{\sigma}\left(g_{j,n+k-\frac{q+1}{2}}^{(l)}(\mathbf{x})\right)\mathbf{1}_{m'\in\mathcal{D}_{n+k-\frac{q+1}{2}}}\mathbf{W}_{m'-n-k+q+1,j',j}^{(l)}.$$

$$= \frac{\partial g_{j,n+k-\frac{q+1}{2}}^{(l)}(\mathbf{x})}{\partial f^{(l-1)}(\mathbf{x})} = \dot{\sigma}\left(g_{j,n+k-\frac{q+1}{2}}^{(l)}(\mathbf{x})\right)\boldsymbol{J}^{\mathcal{D}_{n+k-\frac{q+1}{2}}}\mathbf{W}_{:,:,j}^{(l)}$$

In summary we obtain

$$\frac{\partial f_{mn}^{(l)}(\mathbf{x})}{\partial f^{(l-1)}(\mathbf{x})} = \boldsymbol{J}^{nm} + \frac{\alpha}{q}\sqrt{\frac{c_w c_v}{C_l C_{l-1}}}\sum_{j=1}^{C_l}\sum_{k=1}^{q}\mathbf{V}_{k,j,m}^{(l)}\dot{\sigma}\left(g_{j,n+k-\frac{q+1}{2}}^{(l)}(\mathbf{x})\right)\boldsymbol{J}^{\mathcal{D}_{n+k-\frac{q+1}{2}}}\mathbf{W}_{:,:,j}^{(l)}.$$

$\square$

**Lemma B.6.** *For any two matrices $M, M' \in \mathbb{R}^{m \times s}$ and $N, N' \in \mathbb{R}^{s \times n}$, if $M^T M'$ is uncorrelated with $N N'^T$, and for every $k \neq k'$ either $\mathbb{E}\left[\left(M^T M'\right)_{kk'}\right] = 0$ or $\mathbb{E}\left[\left(N N'^T\right)_{k'k}\right] = 0$, then*

$$\mathbb{E}\left[\left\langle MN, M'N'\right\rangle\right] = \sum_{p=1}^{s} \mathbb{E}\left[\left(M^T M'\right)_{p,p}\right] \mathbb{E}\left[\left(N^T N'\right)_{p,p}\right] = \mathbb{E}\left[tr\left(M^T M' \odot N^T N'\right)\right].$$

*Proof.* Following the definition of an inner product we have

$$\mathbb{E}\left[\left\langle MN, M'N'\right\rangle\right] = \mathbb{E}\left[\sum_{i=1}^{m}\sum_{j=1}^{n}(MN)_{ij}\left(M'N'\right)_{ij}\right] = \mathbb{E}\left[\sum_{i=1}^{m}\sum_{j=1}^{n}\left(\sum_{p=1}^{s}M_{i,p}N_{p,j}\right)\left(\sum_{p'=1}^{s}M'_{i,p'}N'_{p'j}\right)\right]$$

$$= \mathbb{E}\left[\sum_{i=1}^{m}\sum_{j=1}^{n}\sum_{p=1}^{s}\sum_{p'=1}^{s}M_{i,p}M'_{ip'}N_{p,j}N'_{p'j}\right] \underset{\text{uncorrelated}}{=} \sum_{p=1}^{s}\sum_{p'=1}^{s}\mathbb{E}\left[\left(M^T M'\right)_{p,p'}\right]\mathbb{E}\left[\left(N N'^T\right)_{p'p}\right]$$

$$\underset{0 \text{ when } p \neq p'}{=} \sum_{p=1}^{s}\mathbb{E}\left[\left(M^T M'\right)_{p,p}\right]\mathbb{E}\left[\left(N N'^T\right)_{p,p}\right] = \mathbb{E}\left[tr\left(M^T M' \odot N^T N'\right)\right].$$

$\square$

**Lemma B.7.** *$\forall 1 \leq l \leq L - 1, 1 \leq k, k' \leq C_l, 1 \leq j, j' \leq d$, it holds that*

$$\mathbb{E}\left[b_{kj}^{(l)}(\mathbf{x})b_{k'j'}^{(l)}(\mathbf{z})\right] = \delta_{kk'}\delta_{jj'}\left(\mathbb{E}\left[b_{kj}^{(l+1)}(\mathbf{x})b_{kj}^{(l+1)}(\mathbf{z})\right] + \frac{\alpha^2 c_w}{q^2 C_l}tr\left(\left(\sum_{p=-\frac{q-1}{2}}^{\frac{q-1}{2}}\Pi_{\mathcal{D}_{j+p,j+p}}^{(l+1)}(\mathbf{x},\mathbf{z})\right)\odot \dot{K}_{\mathcal{D}_{j,j}}^{(l+1)}(\mathbf{x},\mathbf{z})\right)\right).$$

*Proof.* By Lemma B.3 we have

$$\mathbb{E}\left[b_{kj}^{(l)}(\mathbf{x})b_{k'j'}^{(l)}(\mathbf{z})\right] = \mathbb{E}\left[\sum_{m=1}^{C_l}\sum_{n=1}^{d}b_{mn}^{(l+1)}(\mathbf{x})\frac{\partial f_{mn}^{(l+1)}(\mathbf{x})}{\partial f_{kj}^{(l)}}, \sum_{m'=1}^{C_l}\sum_{n'=1}^{d}b_{m'n'}^{(l+1)}(\mathbf{z})\frac{\partial f_{m'n'}^{(l+1)}(\mathbf{z})}{\partial f_{k'j'}^{(l)}}\right]$$

$$= \sum_{m=1}^{C_l}\sum_{n=1}^{d}\sum_{m'=1}^{C_l}\sum_{n'=1}^{d}\mathbb{E}\left[b_{mn}^{(l+1)}(\mathbf{x})b_{m'n'}^{(l+1)}(\mathbf{z})\frac{\partial f_{mn}^{(l+1)}(\mathbf{x})}{\partial f_{kj}^{(l)}}\frac{\partial f_{m'n'}^{(l+1)}(\mathbf{z})}{\partial f_{k'j'}^{(l)}}\right].$$

Now as $b^{(l+1)}(\mathbf{x})$ is uncorrelated with $\frac{\partial f_{mn}^{(l+1)}(\mathbf{x})}{\partial f_{kj}^{(l)}}$ we get

$$= \sum_{m=1}^{C_l}\sum_{n=1}^{d}\sum_{m'=1}^{C_l}\sum_{n'=1}^{d}\mathbb{E}\left[b_{mn}^{(l+1)}(\mathbf{x})b_{m'n'}^{(l+1)}(\mathbf{z})\right]\mathbb{E}\left[\frac{\partial f_{mn}^{(l+1)}(\mathbf{x})}{\partial f_{kj}^{(l)}}\frac{\partial f_{m'n'}^{(l+1)}(\mathbf{z})}{\partial f_{k'j'}^{(l)}}\right].$$

By induction (where the base case is Lemma B.4), we can assume that if $m \neq m'$ or $n \neq n'$ then $\mathbb{E}\left[b_{mn}^{(l+1)}(\mathbf{x})b_{m'n'}^{(l+1)}(\mathbf{x})\right] = 0$. We therefore obtain

$$\mathbb{E}\left[b_{kj}^{(l)}(\mathbf{x})b_{k'j'}^{(l)}(\mathbf{x})\right] = \sum_{m=1}^{C_l}\sum_{n=1}^{d}\mathbb{E}\left[b_{mn}^{(l+1)}(\mathbf{x})b_{mn}^{(l+1)}(\mathbf{z})\right]\mathbb{E}\left[\frac{\partial f_{mn}^{(l+1)}(\mathbf{x})}{\partial f_{kj}^{(l)}}\frac{\partial f_{mn}^{(l+1)}(\mathbf{z})}{\partial f_{k'j'}^{(l)}}\right].$$

It remains to calculate $\mathbb{E}\left[\frac{\partial f_{mn}^{(l+1)}(\mathbf{x})}{\partial f_{kj}^{(l)}}\frac{\partial f_{mn}^{(l+1)}(\mathbf{z})}{\partial f_{k'j'}^{(l)}}\right]$. Lemma B.5 states that

$$\frac{\partial f_{mn}^{(l+1)}(\mathbf{x})}{\partial f_{kj}^{(l)}(\mathbf{x})} = \left[\frac{\partial f_{mn}^{(l+1)}(\mathbf{x})}{\partial f^{(l)}(\mathbf{x})}\right]_{jk}$$

$$= \underbrace{\delta_{km}\delta_{jn}}_{\text{Denote by } A_{kj}^{mn}} + \underbrace{\underbrace{\frac{\alpha}{q}\sqrt{\frac{c_w c_v}{C_l C_{l+1}}}}_{\text{Denote by } C}\sum_{s=1}^{C_{l+1}}\sum_{p=1}^{q}\mathbf{V}_{p,s,m}^{(l+1)}\dot{\sigma}\left(g_{s,p+n-\frac{q+1}{2}}^{(l+1)}(\mathbf{x})\right)\mathbf{1}_{j\in\mathcal{D}_{n+p-\frac{q-1}{2}}}\mathbf{W}_{j-n-p+q+1,k,s}^{(l+1)}}_{B_{kj}^{mn}(\mathbf{x})}.$$

Using this notation we have:

$$\mathbb{E}\left[\frac{\partial f_{mn}^{(l+1)}(\mathbf{x})}{\partial f_{kj}^{(l)}}\frac{\partial f_{mn}^{(l+1)}(\mathbf{z})}{\partial f_{k'j'}^{(l)}}\right] = A_{kj}^{mn}A_{k'j'}^{mn} + C\left(A_{kj}^{mn}\mathbb{E}\left[B_{k'j'}^{mn}\right] + A_{k'j'}^{mn}\mathbb{E}\left[B_{kj}^{mn}\right]\right) + C^2\mathbb{E}\left[B_{kj}^{mn}(\mathbf{x})B_{k'j'}^{mn}(\mathbf{z})\right].$$

We will deal with each term separately. First, we consider the first term:

$$A_{kj}^{mn}A_{k'j'}^{mn} = \delta_{km}\delta_{jn}\delta_{k'm}\delta_{j'n} = \delta_{kk'm}\delta_{jj'n},$$

where we use the notation $\delta_{kk'm} = \begin{cases} 1 & k = k' = m \\ 0 & \text{otherwise} \end{cases}$ and likewise for $\delta_{jj'n}$. For the second term, since $\mathbf{V}, \mathbf{W}$ are $0$ meaned i.i.d Gaussians (Equation (8)), and $\dot{\sigma}$ is the indicator function we have $\mathbb{E}\left[B_{kj}^{mn}(\mathbf{x})\right] = \mathbb{E}\left[B_{k'j'}^{mn}(\mathbf{z})\right] = 0$, Therefore,

$$C\left(A_{kj}^{mn}\mathbb{E}\left[B_{k'j'}^{mn}(\mathbf{z})\right] + A_{k'j'}^{mn}\mathbb{E}\left[B_{kj}^{mn}(\mathbf{x})\right]\right) = 0.$$

For the last term, by definition

$$B_{kj}^{mn}(\mathbf{x}) = \sum_{s=1}^{C_{l+1}}\sum_{p=1}^{q}\mathbf{V}_{p,s,m}^{(l+1)}\dot{\sigma}\left(g_{s,n+p-\frac{q+1}{2}}^{(l+1)}(\mathbf{x})\right)\mathbf{1}_{j\in\mathcal{D}_{n+p-\frac{q+1}{2}}}\mathbf{W}_{j-n-p+q+1,k,s}^{(l+1)}$$

and

$$B_{k'j'}^{mn}(\mathbf{z}) = \sum_{s'=1}^{C_{l+1}}\sum_{p'=1}^{q}\mathbf{V}_{p',s',m}^{(l+1)}\dot{\sigma}\left(g_{s',n+p'-\frac{q+1}{2}}^{(l+1)}(\mathbf{z})\right)\mathbf{1}_{j'\in\mathcal{D}_{n+p'-\frac{q+1}{2}}}\mathbf{W}_{j'-n-p'+q+1,k',s'}^{(l+1)}.$$

From (8), $\mathbb{E}\left[\mathbf{V}_{p,s,m}^{(l+1)}\mathbf{V}_{p',s',m}^{(l+1)}\right] = \delta_{p,p'}\delta_{s,s'}$, and since they are uncorrelated with $\mathbf{W}^{(l+1)}$ and $g^{(l+1)}$ then

$$\mathbb{E}\left[B_{kj}^{mn}(\mathbf{x})B_{k'j'}^{mn}(\mathbf{z})\right] =$$

$$= C^2\sum_{s=1}^{C_{l+1}}\sum_{p=1}^{q}\mathbb{E}\left[\dot{\sigma}\left(g_{s,n+p-\frac{q+1}{2}}^{(l+1)}(\mathbf{x})\right)\mathbf{W}_{j-n-p+q+1,k,s}^{(l+1)}\dot{\sigma}\left(g_{s,p+n-\frac{q+1}{2}}^{(l+1)}(\mathbf{z})\right)\mathbf{W}_{j'-n-p+q+1,k',s}^{(l+1)}\right]\mathbf{1}_{j,j'\in\mathcal{D}_{n+p-\frac{q+1}{2}}}$$

$$= \delta_{kk'}\delta_{jj'}C^2\sum_{s=1}^{C_{l+1}}\sum_{p=1}^{q}\mathbb{E}\left[\dot{\sigma}\left(g_{s,n+p-\frac{q+1}{2}}^{(l+1)}(\mathbf{x})\right)\dot{\sigma}\left(g_{s,n+p-\frac{q+1}{2}}^{(l+1)}(\mathbf{z})\right)\right]\mathbf{1}_{j,j'\in\mathcal{D}_{n+p-\frac{q+1}{2}}}$$

$$= \delta_{kk'}\delta_{jj'}\frac{\alpha^2}{q^2C_l}\sum_{p=1}^{q}\dot{K}_{n+p-\frac{q+1}{2},n+p-\frac{q+1}{2}}^{(l+1)}(\mathbf{x},\mathbf{z})\mathbf{1}_{j,j'\in\mathcal{D}_{n+p-\frac{q+1}{2}}}$$

Overall,

$$\mathbb{E}\left[b_{kj}^{(l)}(\mathbf{x})b_{k'j'}^{(l)}(\mathbf{z})\right] =$$

$$= \sum_{m=1}^{C_l}\sum_{n=1}^{d}\mathbb{E}\left[b_{mn}^{(l+1)}(\mathbf{x})b_{mn}^{(l+1)}(\mathbf{z})\right]\left(\delta_{kk'm}\delta_{jj'n} + \delta_{kk'}\delta_{jj'}\frac{\alpha^2}{q^2C_l}\sum_{p=1}^{q}\dot{K}_{n+p-\frac{q+1}{2},n+p-\frac{q+1}{2}}^{(l+1)}(\mathbf{x},\mathbf{z})\mathbf{1}_{j,j'\in\mathcal{D}_{n+p-\frac{q+1}{2}}}\right)$$

$$= \delta_{kk'}\delta_{jj'}\left(\mathbb{E}\left[b_{kj}^{(l+1)}(\mathbf{x})b_{kj}^{(l+1)}(\mathbf{z})\right] + \frac{\alpha^2}{q^2C_l}\sum_{n=1}^{d}\sum_{p=1}^{q}\Pi_{n,n}^{(l+1)}(\mathbf{x},\mathbf{z})\dot{K}_{n+p-\frac{q+1}{2},n+p-\frac{q+1}{2}}^{(l+1)}(\mathbf{x},\mathbf{z})\mathbf{1}_{j,j'\in\mathcal{D}_{n+p-\frac{q+1}{2}}}\right).$$

Now observe that

$$j \in \mathcal{D}_{n+p-\frac{q+1}{2}} \iff n + p - q \le j \le n + p - 1 \iff j - p + 1 \le n \le j - p + q.$$

So we can rewrite the above as

$$\mathbb{E}\left[b_{kj}^{(l)}(\mathbf{x})b_{k'j'}^{(l)}(\mathbf{z})\right] =$$

$$= \delta_{kk'}\delta_{jj'}\left(\mathbb{E}\left[b_{kj}^{(l+1)}\left(\mathbf{x}\right)b_{kj}^{(l+1)}\left(\mathbf{z}\right)\right] + \frac{\alpha^2 c_w}{q^2 C_l}\sum_{p=1}^{q}\sum_{n=j-p+1}^{j-p+q}\Pi_{n,n}^{(l+1)}\left(\mathbf{x},\mathbf{z}\right)\dot{K}_{n+p-\frac{q+1}{2},n+p-\frac{q+1}{2}}^{(l+1)}\left(\mathbf{x},\mathbf{z}\right)\right)$$

$$= \delta_{kk'}\delta_{jj'}\left(\mathbb{E}\left[b_{kj}^{(l+1)}\left(\mathbf{x}\right)b_{kj}^{(l+1)}\left(\mathbf{z}\right)\right] + \frac{\alpha^2 c_w}{q^2 C_l}\sum_{p=1}^{q}\mathrm{tr}\left(\Pi_{\mathcal{D}_{j-p+\frac{q+1}{2},j-p+\frac{q+1}{2}}}^{(l+1)}\left(\mathbf{x},\mathbf{z}\right)\odot\dot{K}_{\mathcal{D}_{j,j}}^{(l+1)}\left(\mathbf{x},\mathbf{z}\right)\right)\right)$$

$$= \delta_{kk'}\delta_{jj'}\left(\mathbb{E}\left[b_{kj}^{(l+1)}\left(\mathbf{x}\right)b_{kj}^{(l+1)}\left(\mathbf{z}\right)\right] + \frac{\alpha^2 c_w}{q^2 C_l}\mathrm{tr}\left(\left(\sum_{p=-\frac{q-1}{2}}^{\frac{q-1}{2}}\Pi_{\mathcal{D}_{j+p,j+p}}^{(l+1)}\left(\mathbf{x},\mathbf{z}\right)\right)\odot\dot{K}_{\mathcal{D}_{j,j}}^{(l+1)}\left(\mathbf{x},\mathbf{z}\right)\right)\right).$$

$\square$

**Lemma B.8.** $\forall 1 \leq j, j' \leq d,$

$$\Pi_{j,j'}^{(L)}\left(\mathbf{x},\mathbf{z}\right) = \frac{1}{dc_w}\mathbf{1}_{j=j'=1},$$

and $\forall 1 \leq l \leq L-1, 1 \leq j, j' \leq d$, it holds that:

$$\Pi_{j,j'}^{(l)}\left(\mathbf{x},\mathbf{z}\right) = \delta_{jj'}\left(\Pi_{j,j}^{(l+1)}\left(\mathbf{x},\mathbf{z}\right) + \frac{\alpha^2}{q^2}tr\left(\left(\sum_{p=\frac{q-1}{2}}^{\frac{q-1}{2}}\Pi_{\mathcal{D}_{j+p,j+p}}^{(l+1)}\left(\mathbf{x},\mathbf{z}\right)\right)\odot\dot{K}_{\mathcal{D}_{j,j}}^{(l+1)}\left(\mathbf{x},\mathbf{z}\right)\right)\right).$$

*Proof.* First, from Lemma B.4 we know that:

$$\frac{1}{c_w}\mathbb{E}\left[\left(b^{(L)^T}\left(\mathbf{x}\right)b^{(L)}\left(\mathbf{z}\right)\right)_{j,j'}\right] = \frac{1}{dc_w}\delta_{j,j'}.$$

Now let $1 \leq l \leq L-1$. Using lemma B.7 we get that

$$\frac{1}{c_w}\mathbb{E}\left[\left(b^{(l)^T}\left(\mathbf{x}\right)b^{(l)}\left(\mathbf{z}\right)\right)_{j,j'}\right] = \frac{1}{c_w}\sum_{k=1}^{C_l}\mathbb{E}\left[b_{kj}^{(l)}\left(\mathbf{x}\right)b_{kj'}^{(l)}\left(\mathbf{z}\right)\right]$$

$$= \sum_{k=1}^{C_l}\delta_{jj'}\frac{1}{c_w}\left(\mathbb{E}\left[b_{kj}^{(l+1)}\left(\mathbf{x}\right)b_{kj}^{(l+1)}\left(\mathbf{z}\right)\right] + \delta_{kk'}\delta_{jj'}\frac{\alpha^2 c_w}{q^2 C_l}\mathrm{tr}\left(\left(\sum_{p=-\frac{q-1}{2}}^{\frac{q-1}{2}}\Pi_{\mathcal{D}_{j+p,j+p}}^{(l+1)}\left(\mathbf{x},\mathbf{z}\right)\right)\odot\dot{K}_{\mathcal{D}_{j,j}}^{(l+1)}\left(\mathbf{x},\mathbf{z}\right)\right)\right)$$

$$= \delta_{jj'}\left(\Pi_{j,j}^{(l+1)}\left(\mathbf{x},\mathbf{z}\right) + \frac{\alpha^2}{q^2}\mathrm{tr}\left(\left(\sum_{p=\frac{q-1}{2}}^{\frac{q-1}{2}}\Pi_{\mathcal{D}_{j+p,j+p}}^{(l+1)}\left(\mathbf{x},\mathbf{z}\right)\right)\odot\dot{K}_{\mathcal{D}_{j,j}}^{(l+1)}\left(\mathbf{x},\mathbf{z}\right)\right)\right).$$

$\square$

**Lemma B.9.** $\forall 1 \leq l \leq L,$

$$\mathbb{E}\left[\left\langle\frac{\partial f\left(\mathbf{x};\theta\right)}{\partial \mathbf{W}^{(l)}}, \frac{\partial f\left(\mathbf{z};\theta\right)}{\partial \mathbf{W}^{(l)}}\right\rangle\right] = \frac{\alpha^2}{q}\sum_{n=1}^{d}\Pi_{nn}^{(l)}\left(\mathbf{x},\mathbf{z}\right)tr\left(\dot{K}_{\mathcal{D}_{n,n}}^{(l)}\left(\mathbf{x},\mathbf{z}\right)\odot\Sigma_{\mathcal{D}_{n,n}}^{(l)}\left(\mathbf{x},\mathbf{z}\right)\right).$$

*Proof.* We first show the case for $2 \leq l \leq L$. $\forall 2 \leq l \leq L, 1 \leq i \leq C_{l-1}, 1 \leq c \leq C_l$, we can express $\frac{\partial f(\mathbf{x};\theta)}{\partial \mathbf{W}_{:,i,c}^{(l)}}$ as

$$\frac{\partial f_{mn}^{(l)}\left(\mathbf{x}\right)}{\partial \mathbf{W}_{:,i,c}^{(l)}} = \frac{\partial}{\partial \mathbf{W}_{:,i,c}^{(l)}}\left[f^{(l-1)}\left(\mathbf{x}\right) + \alpha\sqrt{\frac{c_v}{qC_l}}\sum_{j=1}^{C_l}\left(\mathbf{V}_{:,j,:}^{(l)}\right)^T\varphi\left(\sigma\left(g_j^{(l)}\left(\mathbf{x}\right)\right)\right)\right]_{mn}.$$

Since $g_j^{(l)}\left(\mathbf{x}\right)$ depends on $\mathbf{W}_{:,i,c}^{(l)}$ iff $c = j$, we get

$$\frac{\partial f_{mn}^{(l)}\left(\mathbf{x}\right)}{\partial \mathbf{W}_{:,i,c}^{(l)}} = \alpha\sqrt{\frac{c_v}{qC_l}}\sum_{k=1}^{q}\mathbf{V}_{k,c,m}^{(l)}\frac{\partial}{\partial \mathbf{W}_{:,i,c}^{(l)}}\sigma\left(g_{c,n+k-\frac{q+1}{2}}^{(l)}\left(\mathbf{x}\right)\right)$$

$$= \frac{\alpha}{q} \sqrt{\frac{c_w c_v}{C_l C_{l-1}}} \sum_{k=1}^{q} \mathbf{V}_{k,c,m}^{(l)} \dot{\sigma} \left( g_{c,n+k-\frac{q+1}{2}}^{(l)} (\mathbf{x}) \right) \frac{\partial}{\partial \mathbf{W}_{:,i,c}^{(l)}} \left[ \left( W_{:,i,:}^{(l)} \right)^T \varphi \left( f_i^{(l-1)} (\mathbf{x}) \right) \right]_{c,n+k-\frac{q+1}{2}}$$

$$= \frac{\alpha}{q} \sqrt{\frac{c_w c_v}{C_l C_{l-1}}} \sum_{k=1}^{q} \mathbf{V}_{k,c,m}^{(l)} \dot{\sigma} \left( g_{c,n+k-\frac{q+1}{2}}^{(l)} (\mathbf{x}) \right) \left( \varphi^T \left( f_i^{(l-1)} (\mathbf{x}) \right) \right)_{n+k-\frac{q+1}{2},:}.$$

Now we have

$$\frac{\partial f(\mathbf{x};\theta)}{\partial \mathbf{W}_{:,i,c}^{(l)}} = \sum_{m=1}^{C_l} \sum_{n=1}^{d} \frac{\partial f(\mathbf{x};\theta)}{\partial f_{mn}^{(l)}(\mathbf{x})} \frac{\partial f_{mn}^{(l)}(\mathbf{x})}{\partial \mathbf{W}_{:,i,c}^{(l)}}$$

$$= \frac{\alpha}{q} \sqrt{\frac{c_w c_v}{C_l C_{l-1}}} \sum_{m=1}^{C_l} \sum_{n=1}^{d} \sum_{k=1}^{q} b_{mn}^{(l)}(\mathbf{x}) \mathbf{V}_{k,c,m}^{(l)} \dot{\sigma} \left( g_{c,n+k-\frac{q+1}{2}}^{(l)}(\mathbf{x}) \right) \left( \varphi^T \left( f_i^{(l-1)}(\mathbf{x}) \right) \right)_{n+k-\frac{q+1}{2},:}.$$

Taking the inner product we get the following expression:

$$\mathbb{E} \left[ \frac{\partial f(\mathbf{x};\theta)}{\partial \mathbf{W}_{:,i,c}^{(l)}}, \frac{\partial f(\mathbf{z};\theta)}{\partial \mathbf{W}_{:,i,c}^{(l)}} \right] = \frac{\alpha^2 c_w c_v}{q^2 C_l C_{l-1}} \sum_{m=1}^{C_l} \sum_{m'=1}^{C_l} \sum_{n=1}^{d} \sum_{n'=1}^{d} \sum_{k=1}^{q} \sum_{k'=1}^{q} \mathbb{E} \left[ b_{mn}^{(l)}(\mathbf{x}) b_{mn'}^{(l)}(\mathbf{z}) \right] \cdot \mathbb{E} \left[ \mathbf{V}_{k,c,m}^{(l)} \mathbf{V}_{k',c,m'}^{(l)} \right] \cdot$$

$$\cdot \mathbb{E} \left[ \left\langle \dot{\sigma} \left( g_{c,n+k-\frac{q+1}{2}}^{(l)}(\mathbf{x}) \right) \left( \varphi^T \left( f_i^{(l-1)}(\mathbf{x}) \right) \right)_{n+k-\frac{q+1}{2},:}, \dot{\sigma} \left( g_{c,n'+k'-\frac{q+1}{2}}^{(l)}(\mathbf{z}) \right) \left( \varphi^T \left( f_i^{(l-1)}(\mathbf{z}) \right) \right)_{n'+k'-\frac{q+1}{2},:} \right\rangle \right].$$

Note however that (8) implies that $\mathbf{V}_{k,c,m}^{(l)} \mathbf{V}_{k',c,m'}^{(l)} = \delta_{kk'} \delta_{mm'}$, and Lemma B.7 implies that $\mathbb{E} \left[ b_{mn}^{(l)}(\mathbf{x}) b_{mn'}^{(l)}(\mathbf{z}) \right] = 0$ when $n \neq n'$. Therefore,

$$\mathbb{E} \left[ \frac{\partial f(\mathbf{x};\theta)}{\partial \mathbf{W}_{:,i,c}^{(l)}}, \frac{\partial f(\mathbf{z};\theta)}{\partial \mathbf{W}_{:,i,c}^{(l)}} \right] =$$

$$= \frac{\alpha^2 c_w c_v}{q^2 C_l C_{l-1}} \sum_{n=1}^{d} \sum_{k=1}^{q} \underbrace{\left( \sum_{m=1}^{C_l} \mathbb{E} \left[ b_{mn}^{(l)}(\mathbf{x}) b_{mn}^{(l)}(\mathbf{z}) \right] \right)}_{=c_w \Pi_{nn}^{(l)}(\mathbf{x},\mathbf{z})} \cdot$$

$$\cdot \mathbb{E} \left[ \left\langle \dot{\sigma} \left( g_{c,n+k-\frac{q+1}{2}}^{(l)}(\mathbf{x}) \right) \left( \varphi^T \left( f_i^{(l-1)}(\mathbf{x}) \right) \right)_{n+k-\frac{q+1}{2},:}, \dot{\sigma} \left( g_{c,n+k-\frac{q+1}{2}}^{(l)}(\mathbf{z}) \right) \left( \varphi^T \left( f_i^{(l-1)}(\mathbf{z}) \right) \right)_{n+k-\frac{q+1}{2},:} \right\rangle \right]$$

$$= \frac{\alpha^2 c_w}{q^2 C_l C_{l-1}} \sum_{n=1}^{d} \Pi_{nn}^{(l)}(\mathbf{x},\mathbf{z}) \sum_{k=1}^{q} \left[ \dot{K}_{\mathcal{D}_{n,n}}^{(l)}(\mathbf{x},\mathbf{z}) \right]_{kk} \sum_{k'=-\frac{q-1}{2}}^{\frac{q-1}{2}} \mathbb{E} \left[ f_{i,n+k-\frac{q+1}{2}+k'}^{(l-1)}(\mathbf{x}) f_{i,n+k-\frac{q+1}{2}+k'}^{(l-1)}(\mathbf{z}) \right].$$

As a result, by linearity and the fact that this result does not depend on $c$ we obtain

$$\mathbb{E} \left[ \left\langle \frac{\partial f(\mathbf{x};\theta)}{\partial \mathbf{W}^{(l)}}, \frac{\partial f(\mathbf{z};\theta)}{\partial \mathbf{W}^{(l)}} \right\rangle \right] =$$

$$= \frac{\alpha^2 c_w}{q^2 C_{l-1}} \sum_{n=1}^{d} \Pi_{nn}^{(l)}(\mathbf{x},\mathbf{z}) \sum_{k=1}^{q} \left[ \dot{K}_{\mathcal{D}_{n,n}}^{(l)}(\mathbf{x},\mathbf{z}) \right]_{kk} \sum_{k'=-\frac{q-1}{2}}^{\frac{q-1}{2}} \sum_{i=1}^{C_{l-1}} \mathbb{E} \left[ f_{i,n+k-\frac{q+1}{2}+k'}^{(l-1)}(\mathbf{x}) f_{i,n+k-\frac{q+1}{2}+k'}^{(l-1)}(\mathbf{z}) \right].$$

Using Lemma A.2 the last term simplifies as follows:

$$\sum_{k'=-\frac{q-1}{2}}^{\frac{q-1}{2}} \sum_{i=1}^{C_{l-1}} \mathbb{E} \left[ f_{i,n+k-\frac{q+1}{2}+k'}^{(l-1)}(\mathbf{x}) f_{i,n+k-\frac{q+1}{2}+k'}^{(l-1)}(\mathbf{z}) \right]$$

$$= \frac{q C_{l-1}}{c_w} \mathbb{E} \left[ g_{*,n+k-\frac{q+1}{2}}^{(l)}(\mathbf{x}) g_{*,n+k-\frac{q+1}{2}}^{(l)}(\mathbf{z}) \right] = \frac{q C_{l-1}}{c_w} \Sigma_{n+k-\frac{q+1}{2},n+k-\frac{q+1}{2}}^{(l)}(\mathbf{x},\mathbf{z})$$

. Finally we obtain

$$\mathbb{E} \left[ \left\langle \frac{\partial f(\mathbf{x};\theta)}{\partial \mathbf{W}^{(l)}}, \frac{\partial f(\mathbf{z};\theta)}{\partial \mathbf{W}^{(l)}} \right\rangle \right] = \frac{\alpha^2}{q} \sum_{n=1}^{d} \Pi_{nn}^{(l)}(\mathbf{x},\mathbf{z}) \sum_{k=1}^{q} \left[ \dot{K}_{\mathcal{D}_{n,n}}^{(l)}(\mathbf{x},\mathbf{z}) \right]_{kk} \left[ \Sigma_{\mathcal{D}_{n,n}}^{(l)}(\mathbf{x},\mathbf{z}) \right]_{kk}$$

$$= \frac{\alpha^2}{q} \sum_{n=1}^{d} \Pi_{nn}^{(l)} (\mathbf{x}, \mathbf{z}) \operatorname{tr} \left( \dot{K}_{\mathcal{D}_{n,n}}^{(l)} (\mathbf{x}, \mathbf{z}) \odot \Sigma_{\mathcal{D}_{n,n}}^{(l)} (\mathbf{x}, \mathbf{z}) \right).$$

The case of $l = 1$ is analogous, except that we replace $\varphi^T \left( f_i^{(l-1)} (\mathbf{x}) \right)$ with $\mathbf{x}_i^T$ and similarly for $\mathbf{z}$, (making minor adjustments accordingly). We therefore obtain

$$\mathbb{E} \left[ \left\langle \frac{\partial f(\mathbf{x}; \theta)}{\partial \mathbf{W}^{(1)}}, \frac{\partial f(\mathbf{z}; \theta)}{\partial \mathbf{W}^{(1)}} \right\rangle \right] =$$

$$= \frac{\alpha^2}{qC_0} \sum_{n=1}^{d} \Pi_{nn}^{(l)} (\mathbf{x}, \mathbf{z}) \sum_{k=1}^{q} \left[ \dot{K}_{\mathcal{D}_{n,n}}^{(l)} (\mathbf{x}, \mathbf{z}) \right]_{kk} \sum_{i=1}^{C_0} \mathbb{E} \left[ \mathbf{x}_{i,n+k-\frac{q+1}{2}} \mathbf{z}_{i,n+k-\frac{q+1}{2}} \right]$$

$$= \frac{\alpha^2}{q} \sum_{n=1}^{d} \Pi_{nn}^{(1)} (\mathbf{x}, \mathbf{z}) \operatorname{tr} \left( \dot{K}_{\mathcal{D}_{n,n}}^{(1)} (\mathbf{x}, \mathbf{z}) \odot \Sigma_{\mathcal{D}_{n,n}}^{(1)} (\mathbf{x}, \mathbf{z}) \right).$$

$\square$

## C    Spectral Decomposition

### C.1    Proof of Theorem 5.1 in the main text

*Proof.* The strategy is to bound the Taylor expansion of the kernels. We use qualities of the Ordinary Bell Polynomials for the lower bound, and use previous work on singularity analysis (Flajolet & Sedgewick, 2009; Chen & Xu, 2020; Bietti & Bach, 2020; Belfer et al., 2021) for the upper bound. The details can be found in the lemmas that follow, resulting in that $\mathcal{K}_{\mathrm{Eq}}^{(L)}$ can be written as $\sum_{\mathbf{n} \geq 0} b_{\mathbf{n}} \mathbf{t}^{\mathbf{n}}$ with

$$c_1 \mathbf{n}^{-2.5} \leq b_{\mathbf{n}} \leq c_2 \mathbf{n}^{-\frac{3}{2d} - 1}$$

and $\Theta_{\mathrm{Eq}}^{(L)}$ can similarly be written as $\sum_{\mathbf{n} \geq 0} b_{\mathbf{n}} \mathbf{t}^{\mathbf{n}}$ with

$$c_1 \mathbf{n}^{-2.5} \leq b_{\mathbf{n}} \leq c_2 \mathbf{n}^{-\frac{1}{2d} - 1}$$

Thus, applying Geifman et al. (2022)[Theorems 3.3,3.4] completes the proof. $\square$

**Lemma C.1.** $\mathcal{K}_{Eq}^{(L)}$ can be written as $\mathcal{K}_{Eq}^{(L)} (\mathbf{t}) = \sum_{\mathbf{n} \geq 0} b_{\mathbf{n}} \mathbf{t}^{\mathbf{n}}$ with

$$c_1 \mathbf{n}^{-\nu} \leq b_{\mathbf{n}},$$

where $c_1$ is a constant if the receptive field of $\mathcal{K}_{Eq}^{(L)}$ includes $\mathbf{n}$ and $0$ otherwise that depends on $L$.

*Proof.* Let $\kappa_1(t) = \sum_{n=0}^{\infty} a_n t^n$ be the power series expansion of $\kappa_1(t)$, where $a_n \sim n^{-2.5}$ (Chen & Xu, 2020).

We prove this by induction on $L$, starting with $L = 1$:

$$\mathcal{K}_{\mathrm{Eq}}^{(1)} (\mathbf{t}) = \frac{1}{C_0} \mathbf{t}_1 + \frac{\alpha^2}{q c_w C_0} \sum_{k=-\frac{q-1}{2}}^{\frac{q-1}{2}} \kappa_1 (\mathbf{t}_{1+k}) = \frac{1}{C_0} \mathbf{t}_1 + \frac{\alpha^2}{q c_w C_0} \sum_{k=-\frac{q-1}{2}}^{\frac{q-1}{2}} \sum_{n=0}^{\infty} a_n \mathbf{t}_{1+k}^n.$$

By letting $\mathbf{e}_i$ be the multi-index with 1 in the $i$ index and 0 elsewhere, it is clear that we can write the above as $\sum_{\mathbf{n} \geq 0} b_{\mathbf{n}} \mathbf{t}^{\mathbf{n}}$ where

$$b_{\mathbf{n}} = \begin{cases} \frac{\alpha^2}{c_w C_0} a_0 & \mathbf{n} = 0 \\ \frac{\alpha^2}{q c_w C_0} a_n + \frac{1}{C_0} \delta_{j,1} & \mathbf{n} = n \mathbf{e}_j \text{ for } -\frac{q-1}{2} \leq j \leq \frac{q-1}{2} \text{ and } n \in \mathbb{N} \\ 0 & \text{otherwise.} \end{cases}$$

For $L \geq 2$, by the induction hypothesis $\mathcal{K}_{\mathrm{Eq}}^{(L-1)}(\mathbf{t}) = \sum_{\mathbf{n} \geq 0} \tilde{b}_{\mathbf{n}} \mathbf{t}^{\mathbf{n}}$ s.t. $c_1 \mathbf{n}^{-\nu} \leq \tilde{b}_{\mathbf{n}}$. Let $N_L$ be the value of $\Sigma_{j,j}^{(l)}(\mathbf{x}, \mathbf{x})$ from Lemma A.6 and $\tilde{N}_L = \frac{\alpha^2 c_v c_w}{2q^2} N_L$.

$$\kappa_1 \left( \frac{\mathcal{K}_{\mathrm{Eq}}^{(L)}(\mathbf{s}_j \mathbf{t})}{N_L} \right) = \sum_{n=0}^{\infty} \frac{a_n}{N_L^n} \left( \sum_{\mathbf{n} \geq 0} \tilde{b}_{\mathbf{n}} (\mathbf{s}_j \mathbf{t})^{\mathbf{n}} \right)^n = \sum_{n=0}^{\infty} \frac{a_n}{N_L^n} \left( \sum_{\mathbf{n} \geq 0} \tilde{b}_{\mathbf{s}_{-j} \mathbf{n}} (\mathbf{t})^{\mathbf{s}_{-j} \mathbf{n}} \right)^n.$$

Using the derivations for the multivariate ordinary Bell Polynomials in Withers & Nadarajah (2010); Schumann (2019) we can rewrite the above as:

$$\kappa_1 \left( \frac{\mathcal{K}_{\mathrm{Eq}}^{(L)}(\mathbf{s}_j \mathbf{t})}{N_L} \right) = \sum_{\mathbf{n} \geq 0} \left( \sum_{n=0}^{\infty} \frac{a_n}{N_L^n} \hat{B}_{\mathbf{n},n} \left( \mathbf{s}_{-j} \tilde{\mathbf{b}} \right) \right) \mathbf{t}^{\mathbf{n}},$$

where $\tilde{\mathbf{b}} = \left( \tilde{b}_{\mathbf{n}} \right)_{\mathbf{n} \geq 0}$ and $\hat{B}_{\mathbf{n},n}$ denotes the ordinary Bell Polynomials. Plugging this in to our formula for the ResCGPK from Proposition A.1 we get

$$\mathcal{K}_{\mathrm{Eq}}^{(L)}(\mathbf{t}) = \mathcal{K}_{\mathrm{Eq}}^{(L-1)}(\mathbf{t}) + \tilde{N}_L \sum_{k,k'=-\frac{q-1}{2}}^{\frac{q-1}{2}} \kappa_1 \left( \frac{\mathcal{K}_{\mathrm{Eq}}^{(L-1)}(\mathbf{s}_{k+k'} \mathbf{t})}{N_L} \right)$$

$$= \sum_{\mathbf{n} \geq 0} \left( \tilde{b}_{\mathbf{n}} + \tilde{N}_L \sum_{j=-q+1}^{q-1} |q - j| \sum_{n=0}^{\infty} \frac{a_n}{N_L^n} \hat{B}_{\mathbf{n},n} \left( \mathbf{s}_{-j} \tilde{\mathbf{b}} \right) \right) \mathbf{t}^{\mathbf{n}}.$$

Let this be $\sum_{\mathbf{n} \geq 0} b_{\mathbf{n}} \mathbf{t}^{\mathbf{n}}$. All the terms in $b_{\mathbf{n}}$ are positive so for a lower bound, it suffices to sum up only specific $n$'s and $j$'s. We choose $n = 1, j = 0$, and since Withers & Nadarajah (2010) showed that $\hat{B}_{\mathbf{n},1} \left( \tilde{\mathbf{b}} \right) = \tilde{b}_{\mathbf{n}}$ we get that

$$b_{\mathbf{n}} \geq \tilde{b}_{\mathbf{n}} + q \frac{a_1 \tilde{N}_L}{N_L} \tilde{b}_{\mathbf{n}} = \left( 1 + \frac{a_1 \alpha^2 c_v c_w}{2q} \right) c_1 \mathbf{n}^{-\nu},$$

where the last equality is by the induction hypothesis. $\square$

**Lemma C.2.** *The bound in Lemma (C.1) holds for $\Theta_{Eq}^{(L)}$.*

*Proof.* Denote by $b_{\mathbf{n}}(\mathcal{K})$ the Taylor coefficients for some kernel $\mathcal{K}$. Theorem 4.2 implies that

$$\Theta_{\mathrm{Eq}}^{(L)}(\mathbf{x}, \mathbf{z}) = \mathcal{K}_{\mathrm{Eq}}^{(L)}(\mathbf{x}, \mathbf{z}) + \underbrace{\frac{\alpha^2}{q} \sum_{l=1}^{L} \sum_{p=1}^{d} \Pi_{tt}^{(l)}(\mathbf{x}, \mathbf{z}) \left( \mathrm{tr} \left( \dot{K}_{\mathcal{D}_{p,p}}^{(l)}(\mathbf{x}, \mathbf{z}) \odot \Sigma_{\mathcal{D}_{p,p}}^{(l)}(\mathbf{x}, \mathbf{z}) + K_{\mathcal{D}_{p,p}}^{(l)}(\mathbf{x}, \mathbf{z}) \right) \right)}_{\text{Denote by } \mathcal{K}'}.$$

Since for any positive definite kernel the Taylor coefficients are non negative, we get that:

$$b_{\mathbf{n}} \left( \Theta_{\mathrm{Eq}}^{(L)} \right) = b_{\mathbf{n}} \left( \mathcal{K}^{\mathrm{Eq}} \right) + b_{\mathbf{n}} \left( \mathcal{K}' \right) \geq b_{\mathbf{n}} \left( \mathcal{K}^{\mathrm{Eq}} \right).$$

$\square$

**Definition C.1.** *Let $\mathcal{K}_{FC}^{(L)}$ be the GPK and $\Theta_{FC}^{(L)}$ the NTK of the bias free fully connected ResNet defined in Huang et al. (2020). For $\mathbf{x}, \mathbf{z} \in \mathbb{S}^{C_0 - 1}, u = \mathbf{x}^T \mathbf{z}$, following the derivation in Huang et al. (2020) and Belfer et al. (2021) (in particular, Appendix B.1 of the latter), the normalized version of these kernels will be:*

$$\overline{\mathcal{K}}_{FC}^{(0)}(u) = u$$

$$\overline{\mathcal{K}}_{FC}^{(L)}(u) = \frac{1}{1 + \alpha^2} \left( \overline{\mathcal{K}}_{FC}^{(L-1)}(u) + \alpha^2 \kappa_1 \left( \overline{\mathcal{K}}_{FC}^{(L-1)}(u) \right) \right)$$

$$\overline{\Theta}_{FC}^{(L)} = \frac{1}{2L v_{L-1}} \sum_{l=1}^{L} v_{l-1} \tilde{P}^{(l)}(u) \left( \kappa_1 \left( \overline{\mathcal{K}}_{FC}^{(l-1)}(u) \right) + \overline{\mathcal{K}}_{FC}^{(l-1)}(u) \kappa_0 \left( \overline{\mathcal{K}}_{FC}^{(l-1)}(u) \right) \right),$$

*where $v_l = \left( 1 + \alpha^2 \right)^l$, $\tilde{P}^L = 1$ and $\tilde{P}^l(u) = \tilde{P}^{l+1}(u) \left( 1 + \alpha^2 \kappa_0 \left( \overline{\mathcal{K}}_{FC}^{(l+1)}(u) \right) \right)$.*

**Lemma C.3.** *For $u \in \mathbb{R}^d$, letting $\mathbf{t}_1 = \mathbf{t}_2 = \ldots = u$ we obtain $\overline{\mathcal{K}}_{FC}^{(L)}(u) = \overline{\mathcal{K}}_{Eq}^{(L)}(\mathbf{t})$ and letting $\mathbf{k}^{(L)} = \Theta_{Eq}^{(L)} - \mathcal{K}_{Eq}^{(L)}$ we obtain $\overline{\Theta}_{FC}^{(L)}(u) = \overline{\mathbf{k}}^{(L)}(\mathbf{t})$.*

*Proof.* For the GPK, this is immediate from Corollary A.2. For the ResCNTK, first recall that

$$\Theta_{\text{Eq}}^{(L)}(\mathbf{x}, \mathbf{z}) = \mathcal{K}_{\text{Eq}}^{(L)}(\mathbf{x}, \mathbf{z}) + \frac{\alpha^2}{q} \sum_{l=1}^{L} \sum_{p=1}^{d} P_p^{(l)}(\mathbf{x}, \mathbf{z}) \left( \text{tr}\left( \dot{K}_{\mathcal{D}_{p,p}}^{(l)}(\mathbf{x}, \mathbf{z}) \odot \Sigma_{\mathcal{D}_{p,p}}^{(l)}(\mathbf{x}, \mathbf{z}) + K_{\mathcal{D}_{p,p}}^{(l)}(\mathbf{x}, \mathbf{z}) \right) \right).$$

Observe first that a direct consequence of Theorem A.1 is that $\forall L \geq 2$

$$\Sigma_{1,1}^{(L)}(\mathbf{x}, \mathbf{z}) = \frac{c_w}{q} \sum_{k=-\frac{q-1}{2}}^{\frac{q-1}{2}} \mathcal{K}_{\text{Eq}}^{(L-1)}(\mathbf{s}_k \mathbf{x}, \mathbf{s}_k \mathbf{z}).$$

and $\Sigma_{1,1}^{(1)}(\mathbf{x}, \mathbf{z}) = \frac{1}{C_0} u$.

Therefore, by choice of $\mathbf{t}$, $\Sigma^{(l)}$, $K^{(l)}$ and $\dot{K}^{(l)}$ have constant diagonals, and so the terms in the trace do not depend on the index $p$, and we get

$$\mathbf{k}^{(L)}(\mathbf{t}) = \alpha^2 \sum_{l=1}^{L} N_{l+1} \left( \kappa_0\left( \overline{\mathcal{K}}_{\text{FC}}^{(l-1)}(u) \right) \overline{\mathcal{K}}_{\text{FC}}^{(l-1)}(u) + \kappa_1\left( \overline{\mathcal{K}}_{\text{FC}}^{(l-1)}(u) \right) \right) \sum_{p=1}^{d} P_p^{(l)}(\mathbf{t}).$$

Note that for each $1 \leq l \leq L$, $\sum_{p=1}^{d} P_p^{(l)}(\mathbf{t}) = \tilde{P}^l(u)$. For $l = L$ they are both equal to 1 by definition. Now by induction we assume for $l+1$ and now show for $l$.

$$\sum_{p=1}^{d} P_p^{(l)}(\mathbf{t}) = \sum_{p=1}^{d} \left( P_p^{(l+1)}(\mathbf{t}) + \frac{\alpha^2}{q^2} \text{tr}\left( \left( \sum_{k=\frac{q-1}{2}}^{\frac{q-1}{2}} P_{\mathcal{D}_{p+k}}^{(l+1)}(\mathbf{t}) \right) \odot \dot{K}_{\mathcal{D}_{p,p}}^{(l+1)}(\mathbf{t}) \right) \right)$$

$$= \sum_{p=1}^{d} \left( P_p^{(l+1)}(\mathbf{t}) + \frac{\alpha^2}{q^2} \kappa_0\left( \overline{K}_{FC}^{(l+1)} \right) \text{tr}\left( \sum_{k=\frac{q-1}{2}}^{\frac{q-1}{2}} P_{\mathcal{D}_{p+k}}^{(l+1)}(\mathbf{t}) \right) \right)$$

$$= \left( \sum_{p=1}^{d} P_p^{(l+1)}(\mathbf{t}) \right) + \alpha^2 \kappa_0\left( \overline{K}_{FC}^{(l+1)} \right) \left( \frac{1}{q^2} \sum_{k=\frac{q-1}{2}}^{\frac{q-1}{2}} \left( \sum_{p=1}^{d} P_p^{(l+1)}(\mathbf{t}) \right) \right).$$

Plugging this in the induction hypothesis we prove the induction. So overall:

$$\mathbf{k}^{(L)}(\mathbf{t}) = \alpha^2 \sum_{l=1}^{L} N_{l+1} \tilde{P}^l(u) \left( \kappa_0\left( \overline{\mathcal{K}}_{\text{FC}}^{(l-1)}(u) \right) \overline{\mathcal{K}}_{\text{FC}}^{(l-1)}(u) + \kappa_1\left( \overline{\mathcal{K}}_{\text{FC}}^{(l-1)}(u) \right) \right).$$

Since

$$N_{l+1} = N_2 v_{l-1},\tag{16}$$

normalizing $\mathbf{k}^{(L)}$ completes the proof. ∎

**Lemma C.4** ((Bietti & Bach, 2020) Section 3.2)**.** *For a small $t > 0$,*

$$\kappa_1(1-t) = 1 - t - \frac{2\sqrt{2}}{3\pi} t^{\frac{3}{2}} + \mathcal{O}\left( t^{\frac{5}{2}} \right).$$

**Lemma C.5.** *For all $L \geq 0$, and for a small $t > 0$,*

$$\overline{\mathcal{K}}_{FC}^{(L)}(1-t) = 1 - t + \Theta\left( t^{\frac{3}{2}} \right).$$

**Remark.** *This lemma and its proof are a slightly modified version of Lemma B.4 from Belfer et al. (2021) where we tighten the bound.*

*Proof.* We prove this by induction. For $l = 0, \overline{\mathcal{K}}_{FC}^{(0)}(1-t) = 1-t$, trivially satisfying the lemma. Suppose the lemma holds for $l-1$, then using Lemma C.4,

$$\overline{\mathcal{K}}_{FC}^{(L)}(1-t) = \frac{1}{1+\alpha^2}\left(\overline{\mathcal{K}}_{FC}^{(L-1)}(1-t) + \alpha^2\kappa_1\left(\overline{\mathcal{K}}_{FC}^{(L-1)}(1-t)\right)\right)$$

$$= \frac{1}{1+\alpha^2}\left(1-t+\Theta\left(t^{\frac{3}{2}}\right) + \alpha^2\kappa_1\left(1-t+\Theta\left(t^{\frac{3}{2}}\right)\right)\right)$$

$$= \frac{1}{1+\alpha^2}\left(1-t+\Theta\left(t^{\frac{3}{2}}\right) + \alpha^2\left(1-t+\Theta\left(t^{\frac{3}{2}}\right) + \frac{2\sqrt{2}}{3\pi}\left(t-\Theta\left(t^{\frac{3}{2}}\right)\right)^{\frac{3}{2}} + \mathcal{O}\left(t-\Theta\left(t^{\frac{3}{2}}\right)\right)^{\frac{5}{2}}\right)\right)$$

$$= \frac{1}{1+\alpha^2}\left(1-t+\Theta\left(t^{\frac{3}{2}}\right) + \alpha^2\left(1-t+\Theta\left(t^{\frac{3}{2}}\right)\right)\right)$$

$$= 1-t+\Theta\left(t^{\frac{3}{2}}\right).$$

$\square$

**Lemma C.6.** $\overline{\mathcal{K}}_{FC}^{(L)}(u)$ *and* $\overline{\Theta}_{FC}^{(L)}(u)$ *can be written as* $\sum_{n=0}^{\infty} a_n u^n$ *with* $a_n \sim n^{-2.5}$ *for* $\overline{\mathcal{K}}_{FC}^{(L)}$ *and* $a_n \sim n^{-1.5}$ *for* $\overline{\Theta}_{FC}^{(L)}$.

*Proof.* Using Lemma C.5 we know that for $t \nearrow 1$, it holds that $\overline{\mathcal{K}}_{FC}^{(L)}(t) = t + f(t)$ where $f(t) = \Theta\left((1-t)^{\frac{3}{2}}\right)$. Using (Flajolet & Sedgewick, 2009)[page 392 Thm. VI.1] we get that $f$ admits a Taylor expansion $\sum_{n=0}^{\infty} a_n t^n$ around 0 with $a_n \sim n^{-2.5}$. Therefore, $\overline{\mathcal{K}}_{FC}^{(L)}(u)$ has a Taylor expansion with coefficients that exhibit the same decay.

For $\overline{\Theta}_{FC}^{(L)}(u)$, using (Belfer et al., 2021)[Lemma 4.5] we know that for $t \nearrow 1$, it holds that $\overline{\Theta}_{FC}^{(L)}(t) = 1 + c_1(1-t)^{\frac{1}{2}} + o\left((1-t)^{\frac{1}{2}}\right)$ for some constant $c_1 < 0$. Similarly to the previous case, using (Flajolet & Sedgewick, 2009)[Thm. VI.1, page 392] we get the desired bound. $\square$

**Lemma C.7.** *Both* $\overline{\mathcal{K}}_{Eq}^{(L)}(\mathbf{t})$ *and* $\overline{\Theta}_{Eq}^{(L)}(\mathbf{t})$ *can be written as* $\sum_{\mathbf{n}\geq 0} b_\mathbf{n}\mathbf{t}^\mathbf{n}$ *with*

$$b_\mathbf{n} \leq c_2\mathbf{n}^{-\nu},$$

*where* $\nu = \frac{3}{2d} + 1$ *for* $\overline{\mathcal{K}}_{Eq}^{(L)}$ *and* $\nu = \nu = \frac{1}{2d} + 1$ *for* $\overline{\Theta}_{Eq}^{(L)}$ *and* $c_2$ *depends on L.*

*Proof.* By Lemma C.6, $\overline{\mathcal{K}}_{FC}^{(L)}(u) = \sum_{n=0}^{\infty} a_n u^n$ with $a_n \sim n^{-2.5}$. Moreover, we have that $\overline{\mathcal{K}}_{Eq}^{(L)}(\mathbf{t}) = \sum_{\mathbf{n}\geq 0} b_\mathbf{n}\mathbf{t}^\mathbf{n}$. Together with lemma C.3 we get that

$$\overline{\mathcal{K}}_{Eq}^{(L)}(\mathbf{t}) = \sum_{\mathbf{n}\geq 0} b_\mathbf{n}\mathbf{t}^\mathbf{n} = \sum_{\mathbf{n}\geq 0} b_\mathbf{n}u^{|\mathbf{n}|} = \sum_{n=0}^{\infty} b_\mathbf{n}u^{|\mathbf{n}|} = \sum_{n=0}^{\infty} u^n \sum_{|\mathbf{n}|=n} b_\mathbf{n}.$$

The uniqueness of the power series implies

$$\sum_{|\mathbf{n}|=n} b_\mathbf{n} = a_n = \Theta\left(n^{-2.5}\right).$$

Plugging in Lemma D.8 From (Geifman et al., 2022) we get that $b_\mathbf{n} \leq c_2\mathbf{n}^{-\frac{3}{2d}-1}$ for some constant $c_2 > 0$.

For the bound for ResCNTK, since by Lemma C.6 $\overline{\Theta}_{FC}^{(L)}(u) = \sum_{n=0}^{\infty} \tilde{a}_n u^n$ with $\tilde{a}_n \sim n^{-1.5}$ and $\overline{k}^{(L)}(\mathbf{t}) = \sum_{\mathbf{n}\geq 0} \tilde{b}_\mathbf{n}\mathbf{t}^\mathbf{n}$, we can analogously get that $\tilde{b}_\mathbf{n} \leq c'\mathbf{n}^{-\frac{1}{2d}-1}$ for some constant $c' > 0$. (The difference from the different bound comes from the referenced lemma in (Geifman et al., 2022).)

Since $k^{(L)} = \Theta_{Eq}^{(L)} - \mathcal{K}_{Eq}^{(L)}$, by combining the two results we get that $\overline{\Theta}_{Eq}^{(L)}(\mathbf{t})$ can be written as $\sum_{\mathbf{n}\geq 0} b'_\mathbf{n}\mathbf{t}^\mathbf{n}$ with $b'_\mathbf{n} \leq c_2\mathbf{n}^{-\frac{1}{2d}-1}$. $\square$

# D  POSITIONAL BIAS OF EIGENVALUES

## D.1  PROOF OF THEOREM 5.2 IN THE MAIN TEXT

We define a "stride-$q$" version of the ResCGPK. Let $Q = \{-\frac{q-1}{2} \ldots, \frac{q-1}{2}\}, R_0 = 0^{2L-1}$ (i.e a set that contains only the tuple $\mathbf{0} = (0, \ldots, 0)$), and for $l \geq 1$ let $R_l := Q^{2l-1} \times \{0\}^{2(L-l)}$ (i.e tuples where the first $2l - 1$ elements are in $Q$ and the last $2(L - l)$ elements are 0). We let $[-1, 1]^{R_l}$ be elements of the form $\tilde{\mathbf{t}}_{\mathbf{a}}$ which are $[-1, 1]$ valued parameters indexed by tuples $R_l$. Also, for every $k, k^{'} \in Q$ and $1 \leq l \leq L$ define $\iota^l_{k,k'} : [-1, 1]^{R_L} \to [-1, 1]^{R_L}$ by $\iota^l_{k,k'}(\tilde{\mathbf{t}})_{\mathbf{a}} = \tilde{\mathbf{t}}_{(a_1, \ldots, a_{2l-3}, k, k', a_{2l} \ldots)}$.

We now define the kernel $\boldsymbol{k}^{(1)} : [-1, 1]^{R_1} \to [-1, 1]$ to be

$$\boldsymbol{k}^{(1)}(\tilde{\mathbf{t}}) = \frac{1}{1+\alpha^2}\left( \underbrace{\tilde{t}_{\mathbf{0}}}_{:=\boldsymbol{k}^{(1)}_1(\tilde{\mathbf{t}})} + \underbrace{\alpha^2 \frac{1}{q} \sum_{k=-\frac{q-1}{2}}^{\frac{q-1}{2}} \kappa_1\left(\tilde{t}_{(k,0,\ldots,0)}\right)}_{:=\boldsymbol{k}^{(1)}_2(\tilde{\mathbf{t}})} \right).$$

Also, for all $2 \leq l \leq L$ we let $\boldsymbol{k}^{(l)} : [-1, 1]^{R_l} \to [-1, 1]$ be

$$\boldsymbol{k}^{(l)}(\tilde{\mathbf{t}}) = \frac{1}{1+\alpha^2}\left( \underbrace{\boldsymbol{k}^{(l-1)}(\iota^l_{0,0}(\tilde{\mathbf{t}}))}_{:=\boldsymbol{k}^{(l)}_1(\tilde{\mathbf{t}})} + \underbrace{\alpha^2 \frac{1}{q^2} \sum_{k,k'=-\frac{q-1}{2}}^{\frac{q-1}{2}} \kappa_1\left(\boldsymbol{k}^{(l-1)}\left(\iota^l_{k,k'}(\tilde{\mathbf{t}})\right)\right)}_{:=\boldsymbol{k}^{(l)}_2(\tilde{\mathbf{t}})} \right).$$

We also define the change of variables $S_l : R_l \to [d]$ by $S_l(\mathbf{a}) = |\mathbf{a}| + 1$ (reminder: $|\cdot|$ on a multi-index means sum of all entries).

We are now ready to define a correspondence between the stride-$q$ ResCGPK and the standard one. Namely, for every $\mathbf{t} \in [-1, 1]^d$ we let $\phi(\mathbf{t}) \in [-1, 1]^{R_L}$ be $\phi(\mathbf{t})_{\mathbf{a}} := \mathbf{t}_{S_L(\mathbf{a})}$, I claim that $\boldsymbol{k}^{(L)}(\phi(\mathbf{t})) = \overline{\mathcal{K}}^{(L)}_{\mathrm{Eq}}(\mathbf{t})$

First observe that for $\mathbf{a} \in R^l$ it holds that $\iota^l_{k,k'}(\phi(\mathbf{t}))_{\mathbf{a}} = \mathbf{t}_{|\mathbf{a}|+1+k+k'} = \phi(\mathbf{s}_{k+k'}\mathbf{t})_{\mathbf{a}}$ (because $\mathbf{a} \in R^l$ so $\mathbf{a}_{2l-2}, \mathbf{a}_{2l-1} = 0$). Also notice that $\iota^l_{k,k'}(\phi(\mathbf{t})) \in R^{l+1}$. Thus, we recursively get that for $\mathbf{a} \in R^1, \iota^1_{k_1,k'_1} \circ \ldots \circ \iota^L_{k_L,k'_L}(\phi(\mathbf{t}))_{\mathbf{a}} = \phi(\mathbf{s}_{k_1+k'_1} \circ \ldots \circ \mathbf{s}_{k_L+k'_L}\mathbf{t})_{\mathbf{a}}$.

Now, for $L = 1$ we trivially have that $\boldsymbol{k}^{(1)}(\phi(\mathbf{t})) = \overline{\mathcal{K}}^{(1)}_{\mathrm{Eq}}(\mathbf{t})$. So for $\mathbf{a} \in R^1$,

$$\boldsymbol{k}^{(1)}\left(\iota^1_{k_1,k'_1} \circ \ldots \circ \iota^L_{k_L,k'_L}(\phi(\mathbf{t}))\right)_{\mathbf{a}} = \boldsymbol{k}^{(1)}\left(\phi(\mathbf{s}_{k_1+k'_1} \circ \ldots \circ \mathbf{s}_{k_L+k'_L}\mathbf{t})\right)_{\mathbf{a}} = \overline{\mathcal{K}}^{(1)}_{\mathrm{Eq}}(\mathbf{s}_{k_1+k'_1} \circ \ldots \circ \mathbf{s}_{k_L+k'_L}\mathbf{t})$$

As such, we get that

$$\boldsymbol{k}^{(2)}\left(\iota^2_{k_1,k'_1} \circ \ldots \circ \iota^L_{k_L,k'_L}(\phi(\mathbf{t}))\right)_{\mathbf{a}} = \overline{\mathcal{K}}^{(2)}_{\mathrm{Eq}}(\mathbf{s}_{k_2+k'_2} \circ \ldots \circ \mathbf{s}_{k_L+k'_L}\mathbf{t})$$

And continuing by induction we eventually get $\boldsymbol{k}^{(L-1)}\left(\iota^L_{k,k'}(\phi(\mathbf{t}))\right) = \overline{\mathcal{K}}^{(L-1)}_{\mathrm{Eq}}(\mathbf{s}_{k+k'}\mathbf{t})$ which implies that $\boldsymbol{k}^{(L)}(\phi(\mathbf{t})) = \overline{\mathcal{K}}^{(L)}_{\mathrm{Eq}}(\mathbf{t})$.

We now move towards better understanding their Taylor expansions. For a function $f(\mathbf{t})$ that can be approximated by a Taylor series $\sum_{\mathbf{n}=0}^{\infty} a_{\mathbf{n}} t^{\mathbf{n}}$ let $[\mathbf{t}^{\mathbf{n}}] f$ denote the coefficient of $\mathbf{t}^{\mathbf{n}}$ in its Taylor series (meaning $a_{\mathbf{n}}$). Let $M_l(\mathbf{n}) = \left\{\tilde{\mathbf{n}} \in \mathbb{Z}^{R_L}_{\geq 0} \mid \mathrm{supp}\tilde{\mathbf{n}} \subseteq R_l \text{ and } \forall i \in [d], \sum_{j \in S^{-1}_L(i)} \tilde{\mathbf{n}}_j = \mathbf{n}_i\right\}$ be the set of multi-indices which are indexed by tuples in $R_L$ with support in $R_l$ that correspond to $\mathbf{n}$.

So if $\tilde{\mathbf{n}} \in M_l(\mathbf{n})$ we get that for every $\mathbf{t} \in [-1,1]^d$, $\phi(\mathbf{t})^{\tilde{\mathbf{n}}} = \mathbf{t}^{\mathbf{n}}$. We get a correspondence between the Taylor expansions of the kernel as follows:

$$\sum_{\mathbf{n} \geq 0} b_{\mathbf{n}} \mathbf{t}^{\mathbf{n}} = \overline{\mathcal{K}}_{\mathrm{Eq}}^{(L)}(\mathbf{t}) = \mathbf{k}^{(l)}(\tilde{\mathbf{t}}) = \sum_{\tilde{\mathbf{n}} \geq 0} \tilde{b}_{\tilde{\mathbf{n}}} \tilde{\mathbf{t}}^{\tilde{\mathbf{n}}} = \sum_{\mathbf{n} \geq 0} \left( \sum_{\tilde{\mathbf{n}} \in M_L(\mathbf{n})} \tilde{b}_{\tilde{\mathbf{n}}} \right) \mathbf{t}^{\mathbf{n}}.$$

By the uniqueness of the power series,

$$b_{\mathbf{n}} = \sum_{\tilde{\mathbf{n}} \in M_L(\mathbf{n})} \tilde{b}_{\tilde{\mathbf{n}}} = \frac{1}{1 + \alpha^2} \left( \sum_{\tilde{\mathbf{n}} \in M_{L-1}(\mathbf{n})} [\tilde{\mathbf{t}}^{\tilde{\mathbf{n}}}] \mathbf{k}_1^{(L)} + \alpha^2 \sum_{\tilde{\mathbf{n}} \in M_L(\mathbf{n})} [\tilde{\mathbf{t}}^{\tilde{\mathbf{n}}}] \mathbf{k}_2^{(L)} \right).$$

Now $\mathbf{k}_1^{(L)}(\tilde{\mathbf{t}}) = \frac{1}{1+\alpha^2} \left( \mathbf{k}_1^{(L-1)}(\tilde{\mathbf{t}}) + \alpha^2 \mathbf{k}_2^{(L-1)}(\tilde{\mathbf{t}}) \right)$ so we can continue to apply this recursively and eventually get:

$$= \frac{1}{(1+\alpha^2)^L} \sum_{\tilde{\mathbf{n}} \in M_0(\mathbf{n})} [\tilde{\mathbf{t}}^{\tilde{\mathbf{n}}}] \mathbf{k}_1^{(1)} + \sum_{l=1}^{L} \left( \frac{\alpha^2}{1+\alpha^2} \right)^{L-l+1} \sum_{\tilde{\mathbf{n}} \in M_l(\mathbf{n})} [\tilde{\mathbf{t}}^{\tilde{\mathbf{n}}}] \mathbf{k}_2^{(l)}$$

$$= \frac{1}{(1+\alpha^2)^L} \mathbf{1}_{\mathrm{supp}\mathbf{n} \subseteq R_0} + \sum_{l=1}^{L} \left( \frac{\alpha^2}{1+\alpha^2} \right)^{L-l+1} \sum_{\tilde{\mathbf{n}} \in M_l(\mathbf{n})} [\tilde{\mathbf{t}}^{\tilde{\mathbf{n}}}] \mathbf{k}_2^{(l)}.$$

Now let $\nu = 2.5$, via the proof in Lemma C.1 we know that for every $1 \leq l \leq L$ there is some $\tilde{c}_l > 0$ s.t.

$$\left( \frac{\alpha^2}{1+\alpha^2} \right)^{L-l+1} \sum_{\tilde{\mathbf{n}} \in M_l(\mathbf{n})} [\tilde{\mathbf{t}}^{\tilde{\mathbf{n}}}] \mathbf{k}_2^{(l)}(\tilde{\mathbf{t}}) \geq \tilde{c}_l \sum_{\tilde{\mathbf{n}} \in M_l(\mathbf{n})} \tilde{\mathbf{n}}^{-\nu}.$$

Let $p_i(l) = \left| S_l^{-1}(i) \right|$, the number of paths from an input pixel $i$ to the output of an $l$ layer CGPK. Using Geifman et al. (2022)[Lemma C.4, C.5] we get that for $A > 1$, some $c_l$ constants, and $c_{\mathbf{n},l} = c_l \prod_{i=1}^{d} A^{\min(p_i^{(l)}, \mathbf{n}_i)}$ it holds that $c_{\mathbf{n},l} \mathbf{n}^{-\nu} \leq \left( \frac{\alpha^2}{1+\alpha^2} \right)^{L-l+1} \sum_{\tilde{\mathbf{n}} \in M_l(\mathbf{n})} [\tilde{\mathbf{t}}^{\tilde{\mathbf{n}}}] \mathbf{k}^{(l)}(\tilde{\mathbf{t}})$.

Overall, we obtain that

$$b_{\mathbf{n}} \geq \sum_{l=0}^{L} c_{\mathbf{n},l} \mathbf{n}^{-\nu}.$$

Now consider the kernel $\hat{\mathbf{k}}^{(l)}(\mathbf{t}) = \sum_{\mathbf{n} \geq 0} c_{\mathbf{n},l} \mathbf{n}^{-\nu}$ then by Geifman et al. (2022)[Lemma C.7], the eigenvalues of this kernel satisfy

$$\lambda_{\mathbf{k}}(\hat{\mathbf{k}}^{(l)}) \geq c_{\mathbf{k},l} \prod_{\substack{i=1 \\ n_i > 0}}^{d} k_i^{-C_0 - 2},$$

where $c_{\mathbf{k},l} = \tilde{c}_l \prod_{i=1}^{d} A^{\min(p_i^{(L)}, k_i)}$ for some constant $\tilde{c}_l$. As for every $l$ the eigenvectors that correspond to $\lambda_{\mathbf{k}}(\hat{k}^{(l)})$ are the same (given by the spherical harmonics), we get by linearity that

$$\lambda_{\mathbf{k}} \left( \overline{\mathcal{K}}_{\mathrm{Eq}}^{(L)} \right) \geq \sum_{l=1}^{L} c_{\mathbf{k},l} \prod_{\substack{i=1 \\ n_i > 0}}^{d} k_i^{-C_0 - 2}.$$

As in Lemma C.2, this also gives a bound on the eigenvalues of $\Theta_{\mathrm{Eq}}^{(L)}$.

# E  INFINITE DEPTH LIMIT

## E.1  PROOF OF THEOREM 6.1 IN THE MAIN TEXT

**Lemma E.1.** *Suppose that* $\alpha = L^{-\gamma}$ *for* $\gamma \in (0.5, 1]$ *then for any* $\mathbf{t} \in [-1,1]^d$, $\left| \overline{\Theta}_{Eq}^{(L)}(\mathbf{t}) - \overline{\Sigma}_{1,1}^{(1)}(\mathbf{t}) \right| \leq \mathcal{O}\left( L^{1-2\gamma} \right)$.

*Proof.* First recall that

$$\Theta_{\text{Eq}}^{(L)}(\mathbf{t}) = \mathcal{K}_{\text{Eq}}^{(L)}(\mathbf{t}) + \frac{\alpha^2}{q} \sum_{l=1}^{L} \sum_{p=1}^{d} P_p^{(l)}(\mathbf{t}) \left( \text{tr} \left( \dot{K}_{\mathcal{D}_{p,p}}^{(l)}(\mathbf{t}) \odot \Sigma_{\mathcal{D}_{p,p}}^{(l)}(\mathbf{t}) + K_{\mathcal{D}_{p,p}}^{(l)}(\mathbf{t}) \right) \right).$$

Let $\boldsymbol{k}^{(L)}(\mathbf{t}) = \frac{1}{\alpha^2} \left( \Theta_{\text{Eq}}^{(L)}(\mathbf{t}) - \mathcal{K}_{\text{Eq}}^{(L)}(\mathbf{t}) \right)$ so that $\Theta_{\text{Eq}}^{(L)}(\mathbf{t}) = \mathcal{K}_{\text{Eq}}^{(L)}(\mathbf{t}) + \alpha^2 k^{(L)}(\mathbf{t})$. Using our calculations in (16) we know that $\boldsymbol{k}(\vec{\mathbf{1}}) = \frac{2}{C_0} L(1 + \alpha^2)^{L-1}$. Therefore,

$$\alpha^2 \boldsymbol{k}(\vec{\mathbf{1}}) = \frac{2}{C_0}(\alpha^2 L)(1 + \alpha^2)^{L-1} \leq \frac{2}{C_0}(\alpha^2 L)(1 + \alpha^2)^L = \frac{2}{C_0} L^{1-2\gamma} \left( 1 + \frac{1}{L^{2\gamma}} \right)^L$$

$$\leq \frac{2}{C_0} L^{1-2\gamma} \left( \left( 1 + \frac{1}{L^{2\gamma}} \right)^{L^{2\gamma}} \right)^{L^{-2\gamma+1}} = \frac{2}{C_0} L^{1-2\gamma} e^{L^{1-2\gamma}} = \mathcal{O}\left( L^{1-2\gamma} \right).$$

Consequently, $\left| \Theta_{\text{Eq}}^{(L)}(\mathbf{t}) - \mathcal{K}_{\text{Eq}}^{(L)}(\mathbf{t}) \right| \leq \alpha^2 \boldsymbol{k}(\vec{\mathbf{1}}) \leq \mathcal{O}\left( L^{1-2\gamma} \right)$. It therefore suffices to prove that $\left| \mathcal{K}_{\text{Eq}}^{(L)}(\mathbf{t}) - \overline{\Sigma}_{1,1}^{(1)}(\mathbf{t}) \right| \leq \mathcal{O}\left( L^{1-2\gamma} \right)$.

To avoid confusion, we denote by $\overline{\mathcal{K}}_{\text{Eq}}^{(L,\alpha)}(\mathbf{t})$ the ResCGPK with a specific $\alpha$ (that may not necessarily be $L^{-\gamma}$). For all $2 \leq l \leq L$ as a result of Corollary A.2 we get that

$$\left| \overline{\mathcal{K}}_{\text{Eq}}^{(l,L^{-\gamma})}(\mathbf{t}) - \overline{\mathcal{K}}_{\text{Eq}}^{(l-1,L^{-\gamma})}(\mathbf{t}) \right|$$

$$\leq \left| \frac{1}{1+\alpha^2} \left( \overline{\mathcal{K}}_{\text{Eq}}^{(l-1,L^{-\gamma})}(\mathbf{t}) + \frac{\alpha^2}{q^2} \sum_{k,k'=-\frac{q-1}{2}}^{\frac{q-1}{2}} \kappa_1 \left( \overline{\mathcal{K}}_{\text{Eq}}^{(l-1,L^{-\gamma})}(\mathbf{s}_{k+k'}\mathbf{t}) \right) \right) - \overline{\mathcal{K}}_{\text{Eq}}^{(l-1,L^{-\gamma})}(\mathbf{t}) \right|$$

$$= \frac{\alpha^2}{1+\alpha^2} \left| \frac{1}{q^2} \sum_{k,k'=-\frac{q-1}{2}}^{\frac{q-1}{2}} \kappa_1 \left( \overline{\mathcal{K}}_{\text{Eq}}^{(l-1,L^{-\gamma})}(\mathbf{s}_{k+k'}\mathbf{t}) \right) - \overline{K}_{\text{Eq}}^{(l-1,L^{-\gamma})}(\mathbf{t}) \right| \leq \frac{\alpha^2}{1+\alpha^2},$$

Where the last inequality follows because $\overline{K}_{\text{Eq}}^{(l-1,L^{-\gamma})} \in [-1,1]$ and so is $\kappa_1$, This implies that

$$\left| \overline{\mathcal{K}}_{\text{Eq}}^{(L,L^{-\gamma})}(\mathbf{t}) - \overline{\Sigma}_{1,1}^{(1)}(\mathbf{t}) \right| = \left| \overline{\mathcal{K}}_{\text{Eq}}^{(L,L^{-\gamma})}(\mathbf{t}) - \overline{\mathcal{K}}_{\text{Eq}}^{(1,L^{-\gamma})}(\mathbf{t}) \right| \leq L \cdot \frac{\alpha^2}{1+\alpha^2} \leq \mathcal{O}\left( L^{1-2\gamma} \right),$$

which completes the proof. $\qquad\square$

### E.2 PROOF OF THEOREM 6.2 IN THE MAIN TEXT

Our goal of this subsections is to prove the following:

**Theorem E.1.** *Let $\bar{K}_{ResCGPK}^{(L)}$ and $\bar{K}_{CGPK}^{(L)}$ respectively denote kernel matrices for the normalized trace kernels ResCGPK and CGPK of depth L. Let $\boldsymbol{B}(K)$ be a double-constant matrix defined for a matrix K as in Lemma 6.1. Then,*

1. $\left\| \bar{K}_{ResCGPK}^{(L)} - \boldsymbol{B}\left( \bar{K}_{ResCGPK}^{(L)} \right) \right\|_1 \underset{L\to\infty}{\longrightarrow} 0$ *and* $\left\| \bar{K}_{CGPK}^{(L)} - \boldsymbol{B}\left( \bar{K}_{CGPK}^{(L)} \right) \right\|_1 \underset{L\to\infty}{\longrightarrow} 0.$

2. $\rho\left( \boldsymbol{B}\left( \bar{K}_{ResCGPK}^{(L)} \right) \right) \underset{L\to\infty}{\longrightarrow} \infty$ *and* $\rho\left( \boldsymbol{B}\left( \bar{K}_{CGPK}^{(L)} \right) \right) \underset{L\to\infty}{\longrightarrow} \infty.$

3. $\exists L_0 \in \mathbb{N}$ *s.t.* $\forall L \geq L_0$, $\rho\left( \boldsymbol{B}\left( \bar{K}_{ResCGPK}^{(L)} \right) \right) < \rho\left( \boldsymbol{B}\left( \bar{K}_{CGPK}^{(L)} \right) \right).$

*Proof.* We give here the main ideas and leave the dirty work to the lemmas.

By Lemmas E.3 and E.2 both $\bar{K}_{\text{ResCGPK}}^{(L)}$ and $\bar{K}_{\text{CGPK}}^{(L)}$ tend towards $\boldsymbol{B}_{1,1}$ as $L \to \infty$. As such, $\boldsymbol{B}\left( \bar{K}_{\text{ResCGPK}}^{(L)} \right) \underset{L\to\infty}{\longrightarrow} \boldsymbol{B}_{1,1}$ and so we get (1).

For (2), observe that $\lambda_{\min}(\boldsymbol{B}(\bar{K}_{\mathrm{CGPK}}^{(L)})) = 1 - \frac{1}{n(n-1)} \sum_{i \neq j} \bar{K}_{\mathrm{CGPK}}^{(L)} \xrightarrow[L \to \infty]{} 0$.

For (3), let $L_0 \in \mathbb{N}$ be the minimal such that the entries of $\boldsymbol{B}\left(\bar{K}_{\mathrm{ResCGPK}}^{(L)}\right)$ and $\boldsymbol{B}\left(\bar{K}_{\mathrm{CGPK}}^{(L)}\right)$ are non negative and let $L \geq L_0$. By lemma $E.3$ we get that $\frac{1}{n(n-1)} \sum_{i \neq j} \bar{K}_{\mathrm{CGPK}}^{(L)} > \frac{1}{n(n-1)} \sum_{i \neq j} \bar{K}_{\mathrm{ResCGPK}}^{(L)}$.

Since $\rho(\boldsymbol{B}_{1,b}) = 1 + n \frac{b}{1-b}$ we get that $\rho\left(\boldsymbol{B}\left(\bar{K}_{\mathrm{ResCGPK}}^{(L)}\right)\right) < \rho\left(\boldsymbol{B}\left(\bar{K}_{\mathrm{CGPK}}^{(L)}\right)\right)$ as desired. $\qquad\square$

**Lemma E.2.** *Suppose that $\alpha$ is a constant that does not depend on $L$, then for any $\mathbf{t} \in [-1,1]^d, \overline{\mathcal{K}}_{Eq}^{(L)}(\mathbf{t}) \to 1$ as $L \to \infty$.*

*Proof.* Denote by $\overline{\mathcal{K}}^{(L)}(\mathbf{t})$ the vector that is $\overline{\mathcal{K}}_{\mathrm{Eq}}^{(L)}(\mathbf{s}_{j-1}\mathbf{t})$ in the $j'th$ coordinate. Using Corollary A.2, let $\mathbb{E}\left[\overline{\mathcal{K}}^{(L)}(\mathbf{t})\right]$ denote the mean of the vector $\overline{\mathcal{K}}^{(L)}(\mathbf{t})$, then by linearity we get:

$$\mathbb{E}\left[\overline{\mathcal{K}}^{(L)}(\mathbf{t})\right] = \frac{1}{1+\alpha^2}\left(\mathbb{E}\left[\overline{\mathcal{K}}^{(L-1)}(\mathbf{t})\right] + \frac{\alpha^2}{q^2}\sum_{k,k'=-\frac{q-1}{2}}^{\frac{q-1}{2}}\mathbb{E}\left[\kappa_1\left(\overline{\mathcal{K}}^{(L-1)}(\mathbf{s}_{k+k'}\mathbf{t})\right)\right]\right),$$

where we let $\kappa_1$ act point-wise on vectors. Since we can permute $\mathbf{t}$ without chaning the mean (i.e., for any $j$, $\mathbb{E}\left[\overline{\mathcal{K}}^{(L)}(\mathbf{t})\right] = \mathbb{E}\left[\overline{\mathcal{K}}^{(L)}(\mathbf{s}_j\mathbf{t})\right]$) we get:

$$\mathbb{E}\left[\overline{\mathcal{K}}^{(L)}(\mathbf{t})\right] = \frac{1}{1+\alpha^2}\left(\mathbb{E}\left[\overline{\mathcal{K}}^{(L-1)}(\mathbf{t})\right] + \alpha^2\mathbb{E}\left[\kappa_1\left(\overline{\mathcal{K}}^{(L-1)}(\mathbf{t})\right)\right]\right) \tag{17}$$

$$= \mathbb{E}\left[\overline{\mathcal{K}}^{(L-1)}(\mathbf{t})\right] + \frac{\alpha^2}{1+\alpha^2}\left(\mathbb{E}\left[\kappa_1\left(\overline{\mathcal{K}}^{(L-1)}(\mathbf{t})\right)\right] - \mathbb{E}\left[\overline{\mathcal{K}}^{(L-1)}(\mathbf{t})\right]\right)$$

$$\geq \mathbb{E}\left[\overline{\mathcal{K}}^{(L-1)}(\mathbf{t})\right] + \frac{\alpha^2}{1+\alpha^2}\left(\kappa_1\left(\mathbb{E}\left[\overline{\mathcal{K}}^{(L-1)}(\mathbf{t})\right]\right) - \mathbb{E}\left[\overline{\mathcal{K}}^{(L-1)}(\mathbf{t})\right]\right), \tag{18}$$

where the last inequality is Jensen's inequality (since $\kappa_1$ is convex (Daniely et al., 2016)). THerefore, let $a_L = \mathbb{E}\left[\overline{\mathcal{K}}^{(L)}(\mathbf{t})\right]$, we can rewrite (18) as:

$$a_L \geq a_{L-1} + \frac{\alpha^2}{1+\alpha^2}\left(\kappa_1\left(a_{L-1}\right) - a_{L-1}\right).$$

We therefore need to show that $a_L \to 1$ as $L \to \infty$. Since $a_L$ is monotonically increasing ($\kappa_1(u) > u$ for all $u \in [-1,1]$ (Daniely et al., 2016)) and bounded in $[-1,1]$ it suffices to show that for all $\epsilon > 0$ there exists some $L \in \mathbb{N}$ s.t. $a_L \geq 1 - \epsilon$. Suppose not, then let $\epsilon > 0$ s.t. for all $L$, $a_L < 1 - \epsilon$. As $\frac{d}{du}\kappa_1(u) = \kappa_0(u)$ and $\kappa_0(u) \in [0,1]$ (Daniely et al., 2016) then $h(u) := \kappa_1(u) - u$ satisfies $\frac{d}{du}h(u) = \kappa_0(u) - 1 \leq 0$ with equality iff $u = 1$. Therefore, for any $u \in [-1, 1-\epsilon], h(u) \geq h(1-\epsilon) > 0$ (The $> 0$ is because $\kappa_1(u) > u$ for $u \in [-1, 1-\epsilon]$). Since we assumed by contradiction that for every $L \in \mathbb{N}, a_L < 1 - \epsilon$, we get that $h(a_L) \geq h(1-\epsilon)$ and thus

$$a_L \geq a_{L-1} + \frac{\alpha^2}{1+\alpha^2}h(a_{L-1}) \geq a_{L-1} + \frac{\alpha^2}{1+\alpha^2}h(1-\epsilon) \geq a_0 + L\frac{\alpha^2}{1+\alpha^2}h(1-\epsilon) \xrightarrow[L \to \infty]{} \infty.$$

However, since $a_L \in [-1,1]$ this leads to a contradiction. $\qquad\square$

**Lemma E.3.** *Let $\mu^{(i)}(\mathbf{t}) = \frac{1}{d}\sum_{j=1}^{d}\underbrace{\kappa_1 \circ \ldots \circ \kappa_1}_{L \text{ times}}(t_j)$, (an average of the entries of $\mathbf{t}$ after $i$ compositions of $\kappa_1$), where $\kappa_1(t) = \frac{1}{\pi}\left(\sqrt{1-t^2} + (\pi - \arccos(t))t\right)$. Let $\overline{\mathcal{K}}_{CGPK\text{-}Tr}^{(L)}(\mathbf{t})$ be the corresponding CGPK-Tr without skip connections. Then*

$$\overline{\mathcal{K}}_{CGPK\text{-}Tr}^{(L)}(\mathbf{t}) - \overline{\mathcal{K}}_{Tr}^{(L)}(\mathbf{t}) \geq \sum_{l=1}^{L}\frac{\mu^{(l)}(\mathbf{t}) - \mu^{(l-1)}(\mathbf{t})}{(1+\alpha^2)^{L-l+1}},$$

*where if $\mathbf{t} \neq \vec{\mathbf{1}}$ this quantity is strictly positive.*

*Proof.* Let $k_{\mathrm{Eq}}^{(L)}(\mathbf{t})$ be the normalized CGPK-EqNet and $k^{(L)}(\mathbf{t})$ be the matrix that is $k_{\mathrm{Eq}}^{(L)}(\mathbf{s}_{1+j}\mathbf{t})$ in the $j$ index. Similarly define the matrix $\overline{\mathcal{K}}^{(L)}(\mathbf{t})$ using $\overline{\mathcal{K}}_{\mathrm{Eq}}^{(L)}(\mathbf{t})$. For convenience let $\overline{\mathcal{K}}^{(0)}(\mathbf{t}) = k^{(0)}(\mathbf{t}) = \mathbf{t}$. Note that $\mathbb{E}\left[k^{(L)}(\mathbf{t})\right] = \overline{\mathcal{K}}_{\mathrm{CGPK\text{-}Tr}}^{(L)}(\mathbf{t})$ and $\mathbb{E}\left[\overline{\mathcal{K}}^{(L)}(\mathbf{t})\right] = \overline{\mathcal{K}}_{\mathrm{Tr}}^{(L)}(\mathbf{t})$. By equation 17 we have that:

$$\mathbb{E}\left[\overline{\mathcal{K}}^{(L)}(\mathbf{t})\right] = \frac{1}{1+\alpha^2}\left(\mathbb{E}\left[\overline{\mathcal{K}}^{(L-1)}(\mathbf{t})\right] + \alpha^2 \mathbb{E}\left[\kappa_1\left(\overline{\mathcal{K}}^{(L-1)}(\mathbf{t})\right)\right]\right),$$

and similarly it can be readily verified that

$$\mathbb{E}\left[k^{(L)}(\mathbf{t})\right] = \mathbb{E}\left[\kappa_1\left(k^{(L-1)}(\mathbf{t})\right)\right].$$

(Note that the CGPK is naturally normalized so we can omit the bar.) We prove this by induction. For $L = 1$ we have:

$$\mathbb{E}\left[k^{(1)}(\mathbf{t})\right] - \mathbb{E}\left[\overline{\mathcal{K}}^{(1)}(\mathbf{t})\right] = \frac{1}{1+\alpha^2}\left(\mu^{(1)} - \mu^{(0)}\right).$$

Now assume for $L - 1 \in \mathbb{N}$, then

$$\mathbb{E}\left[k^{(L)}(\mathbf{t})\right] - \mathbb{E}\left[\overline{\mathcal{K}}^{(L)}(\mathbf{t})\right] = \mathbb{E}\left[\kappa_1\left(k^{(L-1)}(\mathbf{t})\right)\right] - \frac{1}{1+\alpha^2}\left(\mathbb{E}\left[\overline{\mathcal{K}}^{(L-1)}(\mathbf{t})\right] + \alpha^2 \mathbb{E}\left[\kappa_1\left(\overline{\mathcal{K}}^{(L-1)}(\mathbf{t})\right)\right]\right).$$

Since $\kappa_1$ is increasing, using the induction hypothesis we know that $-\alpha^2\mathbb{E}\left[\kappa_1\left(\overline{\mathcal{K}}^{(L-1)}(\mathbf{t})\right)\right] \geq -\alpha^2\mathbb{E}\left[\kappa_1\left(k^{(L-1)}(\mathbf{t})\right)\right]$, and therefore

$$\mathbb{E}\left[k^{(L)}(\mathbf{t})\right] - \mathbb{E}\left[\overline{\mathcal{K}}^{(L)}(\mathbf{t})\right] \geq$$

$$\geq \mathbb{E}\left[\kappa_1\left(k^{(L-1)}(\mathbf{t})\right)\right] - \frac{1}{1+\alpha^2}\left(\mathbb{E}\left[\overline{\mathcal{K}}^{(L-1)}(\mathbf{t})\right] + \alpha^2\mathbb{E}\left[\kappa_1\left(k^{(L-1)}(\mathbf{t})\right)\right]\right)$$

$$= \frac{1}{1+\alpha^2}\left(\mathbb{E}\left[\kappa_1\left(k^{(L-1)}(\mathbf{t})\right)\right] - \mathbb{E}\left[\overline{\mathcal{K}}^{(L-1)}(\mathbf{t})\right]\right)$$

$$= \frac{1}{1+\alpha^2}\left(\left(\mu^{(L)} - \mu^{(L-1)}\right) + \left(\mathbb{E}\left[k^{(L-1)}(\mathbf{t})\right] - \mathbb{E}\left[\overline{\mathcal{K}}^{(L-1)}(\mathbf{t})\right]\right)\right).$$

Applying the induction hypothesis recursively provides the desired result. $\square$

**Lemma E.4** (Lemma 6.1 in the main text). *Let $A \in \mathbb{R}^{n \times n}$ ($n \geq 2$) be a normalized kernel matrix with $\sum_{i \neq j} A_{ij} \geq 0$. Let $B(A) = B_{1,b}$ with $b = \frac{1}{n(n-1)}\sum_{i \neq j} A_{ij}$ and $\epsilon = \sup_i \sum_{j \neq i} |A_{ij} - B(A)_{ij}|$. Then,*

*1. $\rho(B(A)) \leq \rho(A)$.*

*2. If $\epsilon < \lambda_{\min}(B(A))$ then $\rho(A) \leq \frac{\lambda_{\max}(B(A))+\epsilon}{\lambda_{\min}(B(A))-\epsilon}$,*

*where $\lambda_{\max}$ and $\lambda_{\min}$ denote the maximal and minimal eigenvalues of $B(A)$.*

*Proof.* For (1), using Marsli (2015)[Theorem 4.4] we have

$$\rho(A) \geq \frac{\gamma_2(A)}{\gamma_1(A)},$$

where

$$\gamma_1(A) = \min\{\frac{1}{n}\sum_{i=1}^{n}\sum_{j=1}^{n} A_{ij} \; , \; \frac{1}{n}\sum_{i=1}^{n} A_{i,i} - \frac{1}{n(n-1)}\sum_{i \neq j} A_{ij}\}$$

$$\gamma_2(A) = \max\{\frac{1}{n}\sum_{i=1}^{n}\sum_{j=1}^{n} A_{ij} \; , \; \frac{1}{n}\sum_{i=1}^{n} A_{ii} - \frac{1}{n(n-1)}\sum_{i \neq j} A_{ij}\}.$$

By the assumptions that the diagonal entries of $\boldsymbol{A}$ are $1$ and that $\sum_{i \neq j} \boldsymbol{A}_{ij} \geq 0$, we get that $\gamma_1(\boldsymbol{A}) = 1 - \frac{1}{n(n-1)} \sum_{i \neq j} \boldsymbol{A}_{ij} = \lambda_{\min}(\boldsymbol{B}(\boldsymbol{A}))$ and $\gamma_2(\boldsymbol{A}) = \frac{1}{n} \sum_{i=1}^{n} \sum_{j=1}^{n} \boldsymbol{A}_{ij} = \lambda_{\max}(\boldsymbol{B}(\boldsymbol{A}))$. Therefore,

$$\rho(\boldsymbol{A}) \geq \frac{\gamma_2(\boldsymbol{A})}{\gamma_1(\boldsymbol{A})} = \frac{\lambda_{\max}(\boldsymbol{B}(\boldsymbol{A}))}{\lambda_{\min}(\boldsymbol{B}(\boldsymbol{A}))} = \rho(\boldsymbol{B}(\boldsymbol{A})).$$

For (2), by the Gershgorin circle theorem, since $\boldsymbol{A} - \boldsymbol{B}(\boldsymbol{A})$ is a matrix with diagonal zero, every eigenvalue $\lambda$ of $\boldsymbol{A} - \boldsymbol{B}(\boldsymbol{A})$ must satisfy $|\lambda| \leq \epsilon$. Since $\boldsymbol{A}$ and $\boldsymbol{A} - \boldsymbol{B}(\boldsymbol{A})$ are symmetric, it holds that $\lambda_{\max}(\boldsymbol{A}) \leq \lambda_{\max}(\boldsymbol{B}(\boldsymbol{A})) + \lambda_{\max}(\boldsymbol{A} - \boldsymbol{B}(\boldsymbol{A}))$ and $\lambda_{\min}(\boldsymbol{A}) \geq \lambda_{\min}(\boldsymbol{B}(\boldsymbol{A})) + \lambda_{\min}(\boldsymbol{A} - \boldsymbol{B}(\boldsymbol{A}))$ from which the lemma follows. □

## F  INFINITE DEPTH DISCUSSION

Lemma C.3 states that for $u \in \mathbb{R}^d$, letting $\mathbf{t}_1 = \mathbf{t}_2 = \ldots = u$, we obtain $\overline{\mathcal{K}}_{\mathrm{FC}}^{(L)}(u) = \overline{\mathcal{K}}_{\mathrm{Eq}}^{(L)}(\mathbf{t})$, and letting $\boldsymbol{k}^{(L)} = \Theta_{\mathrm{Eq}}^{(L)} - \mathcal{K}_{\mathrm{Eq}}^{(L)}$ we obtain $\overline{\Theta}_{\mathrm{FC}}^{(L)}(u) = \overline{\boldsymbol{k}}^{(L)}(\mathbf{t})$, where the fully connected ResNTK and ResGPK are defined in Huang et al. (2020).

One may ask why does $\overline{\Theta}_{\mathrm{FC}}^{(L)}(u) = \overline{\boldsymbol{k}}^{(L)}(\mathbf{t})$ and not $\overline{\Theta}_{\mathrm{FC}}^{(L)}(u) = \overline{\Theta}_{\mathrm{Eq}}^{(L)}(\mathbf{t})$? This is in fact a consequence of Huang et al. (2020) not training the last layer (denoted by $\mathbf{v}$ in their paper.) If they were to train the parameters $\mathbf{v}$, the term $\mathbb{E}\left[\left\langle \frac{\partial f(\mathbf{x};\theta)}{\partial \mathbf{v}} \frac{\partial f(\mathbf{z};\theta)}{\partial \mathbf{v}} \right\rangle\right]$ would be added to their ResNTK expression. But this term is exactly equal to the ResGPK. Therefore, training the last layer adds the ResCGPK to the ResNTK expression. This is indeed confirmed in (Tirer et al., 2022), who derived ResNTK when the last layer is trained.

So if the last layer is trained, we would have $\overline{\Theta}_{\mathrm{FC}}^{(L)}(u) = \overline{\Theta}_{\mathrm{Eq}}^{(L)}(\mathbf{t})$, and thus Theorem 6.1 would imply that $\overline{\Theta}_{\mathrm{FC}}^{(L)}(u) \underset{L \to \infty}{\longrightarrow} u$. Intuitively, this happens because the term $u$ exists in the ResGPK, and is the only term that is not multiplies by $\alpha$. So if $\alpha$ decays quickly enough, $u$ becomes the dominant term.

Instead, by eliminating the ResGPK from the ResNTK expression, all the terms are multiplies by $\alpha$. So after normalizing, the two layer ResNTK becomes equivalent to the two layer FC-NTK (Belfer et al., 2021).

If we were to not train the last layer, we would have a similar result, where the resulting kernel would correspond to a 2 layer CNTK. We give here a sketch proof (the details are analogous to Belfer et al. (2021)). Theorem A.1 states that

$$\Sigma_{j,j'}^{(2)}(\mathbf{t}) = \frac{c_w}{q} \mathrm{tr}\left(\Sigma_{\mathcal{D}_{j,j'}}^{(1)}(\mathbf{t})\right) + \frac{\alpha^2}{q^2} \sum_{k=-\frac{q-1}{2}}^{\frac{q-1}{2}} \mathrm{tr}\left(K_{\mathcal{D}_{j+k,j'+k}}^{(1)}(\mathbf{t})\right).$$

For $3 \leq l \leq L$,

$$\Sigma_{j,j'}^{(l)}(\mathbf{t}) = \Sigma_{j,j'}^{(l-1)}(\mathbf{t}) + \frac{\alpha^2}{q^2} \sum_{k=-\frac{q-1}{2}}^{\frac{q-1}{2}} \mathrm{tr}\left(K_{\mathcal{D}_{j+k,j'+k}}^{(l-1)}(\mathbf{t})\right).$$

So for $\gamma = L^{-\gamma}$ we have that for all $l \geq 3$,

$$\left|\Sigma_{j,j'}^{(l)}(\mathbf{t}) - \Sigma_{j,j'}^{(l-1)}(\mathbf{t})\right| \leq \alpha^2 \left|\sum_{k=-\frac{q-1}{2}}^{\frac{q-1}{2}} \mathrm{tr}\left(K_{\mathcal{D}_{j+k,j'+k}}^{(l-1)}(\mathbf{t})\right)\right| \leq \alpha^2.$$

So $\left|\Sigma_{j,j'}^{(l)}(\mathbf{t}) - \Sigma_{j,j'}^{(2)}(\mathbf{t})\right| \leq L \cdot \alpha^2 = L^{1-2\gamma}$.

In turn we also have $\left|K_{j,j'}^{(l)}(\mathbf{t}) - K_{j,j'}^{(2)}(\mathbf{t})\right| = L^{1-2\gamma}$ and $\left|\dot{K}_{j,j'}^{(l)}(\mathbf{t}) - \dot{K}_{j,j'}^{(2)}(\mathbf{t})\right| = L^{1-2\gamma}$.

Furthermore, by Theorem 4.2 we have:

$$\left| P_j^{(l+1)}(\mathbf{t}) - P_j^{(l)}(\mathbf{t}) \right| \leq \left| \frac{\alpha^2}{q^2} \mathrm{tr} \left( \left( \sum_{p=\frac{q-1}{2}}^{\frac{q-1}{2}} P_{\mathcal{D}_{j+p}}^{(l+1)}(\mathbf{t}) \right) \odot \dot{K}_{\mathcal{D}_{j,j}}^{(l+1)}(\mathbf{t}) \right) \right|$$

$$\leq \alpha^2 \left| \frac{1}{q^2} \sum_{p=\frac{q-1}{2}}^{\frac{q-1}{2}} P_{\mathcal{D}_{j+p}}^{(l+1)}(\mathbf{t}) \right| \leq \alpha^2 \left| P_1^{(l+1)}\left(\vec{\mathbf{1}}\right) \right| \leq \alpha^2 (1+\alpha^2)^{L-l}.$$

Now because it holds that:

$$\alpha^2(1+\alpha^2)^{L-l} = (1+\frac{1}{L^{2\gamma}})^{L-l} \leq (1+\frac{1}{L^{2\gamma}})^{L^{2\gamma} \cdot L^{1-2\gamma}} \leq e^{L^{1-2\gamma}},$$

Since $P_j^{(L)}(\mathbf{t}) = \vec{\mathbf{1}}_{j=1}$ we analogously get $\left| \vec{\mathbf{1}}_{j=1} - P_j^{(2)}(\mathbf{t}) \right| \leq L^{1-2\gamma} \cdot e^{L^{1-2\gamma}} \mathcal{O}(L^{1-2\gamma}).$

Now recall that

$$\boldsymbol{k}^{(L)}(\mathbf{t}) = \frac{\alpha^2}{q} \sum_{l=1}^{L} \underbrace{\sum_{p=1}^{d} P_p^{(l)}(\mathbf{t}) \left( \mathrm{tr} \left( \dot{K}_{\mathcal{D}_{p,p}}^{(l)}(\mathbf{t}) \odot \Sigma_{\mathcal{D}_{p,p}}^{(l)}(\mathbf{t}) + K_{\mathcal{D}_{p,p}}^{(l)}(\mathbf{t}) \right) \right)}_{\text{Denote by } \boldsymbol{k}^{(L,l)}(\mathbf{t})}$$

where by (16) we know that $\boldsymbol{k}^{(L)}(\vec{\mathbf{1}}) = \alpha^2 \frac{2}{C_0} L(1+\alpha^2)^{L-1}.$

Normalizing the kernel implies

$$\overline{\boldsymbol{k}}^{(L)}(\mathbf{t}) = \frac{\boldsymbol{k}^{(L)}(\mathbf{t})}{\boldsymbol{k}^{(L)}(\vec{\mathbf{1}})} \approx \boldsymbol{k}^{(L,2)}(\mathbf{t}) \approx \frac{1}{C} \mathrm{tr} \left( \dot{K}_{\mathcal{D}_{1,1}}^{(2)}(\mathbf{t}) \odot \Sigma_{\mathcal{D}_{1,1}}^{(2)}(\mathbf{t}) + K_{\mathcal{D}_{1,1}}^{(2)}(\mathbf{t}) \right),$$

where $C$ is some normalizing constant (Note that after normalizing, $\boldsymbol{k}^{(L,1)}(\mathbf{t})$ becomes negligible.)

For such $\alpha$, $\overline{\Sigma}_{j,j'}^{(2)}(\mathbf{t}) \xrightarrow[L\to\infty]{} \frac{1}{q}\mathrm{tr}\left(\Sigma_{\mathcal{D}_{1,1}}(\mathbf{t})\right)$. As such, after normalizing, in the infinite depth limit, the expression $\mathrm{tr}\left( \dot{K}_{\mathcal{D}_{1,1}}^{(2)}(\mathbf{t}) \odot \Sigma_{\mathcal{D}_{1,1}}^{(2)}(\mathbf{t}) + K_{\mathcal{D}_{1,1}}^{(2)}(\mathbf{t}) \right)$ becomes the two layer CGPK (aka one hidden layer, denoted by $L=1$) with inputs $\hat{\mathbf{t}}$ where $\hat{t}_i = \frac{1}{q} \sum_{k=-\frac{q-1}{2}}^{\frac{q-1}{2}} t_{i+k}$.

## G  EIGENVALUE DECAY EXPERIMENT

We use (Geifman et al., 2022)[Lemma A.6] to numerically compute the eigenvalues. Namely, for each frequency in Figure 1 we compute the Gegenbaur polynomials and the kernel, and numerically integrate. Note that as this integration requires evaluating the kernel many times, we are limited to $d = 4$ and $L = 3$. To prevent the receptive field from being much larger than $d$, and in order to better match the CGPK expression from (Geifman et al., 2022), we slightly modify the ResCGPK to include one convolution in every layer instead of two, where the layer ends after the ReLU. Thus the kernel computed is with $d = 4, q = 2, \alpha = 1$ is:

$$\boldsymbol{k}_0(\mathbf{t}) = \frac{1}{1+\beta}(\beta t_i + \kappa_1(t_i))$$

$$\boldsymbol{k}_i(\mathbf{t}) = \frac{1}{1+\beta}\left( \beta t_i + \kappa_1\left( \frac{1}{2}(t_i + t_{i+1}) \right) \right),$$

where $\beta = 0$ is the CGPK from (Geifman et al., 2022) and $\beta = 1$ is the modified ResCGPK.

