# OpenReview forum: "A Kernel Perspective of Skip Connections in Convolutional Networks"
_ICLR.cc/2023/Conference — ICLR 2023 notable top 5%_

### Official Review · Reviewer_zQoE · 2022-10-26

**Confidence:** 2
**Correctness:** 3
**Technical Novelty And Significance:** 3
**Empirical Novelty And Significance:** 2
**Recommendation:** 5

**Clarity, Quality, Novelty And Reproducibility:**

The paper is mathematically clear.

In terms of quality, the paper contains significant theoretical results.

The topic that the paper addresses is a bit narrow. They only focus on the skip connections, which I feel somehow limits the novelty.

Reproducibility is OK (There's a minimum experiment).

**Strength And Weaknesses:**

Strengths
- The effect of skip connection is rigorously analyzed in the language of kernels.
- The eigenvalues of the kernels are evaluated.

Weaknesses
- There's no clear outcome for practitioners. It's not straightforward to utilize the theoretical results for new methods/algorithms/architectures.


**Summary Of The Paper:**

This paper studies how skip connections affect CNNs in the sense of learning theory. The authors investigate this by approximating CNN with kernels based on NTK and the gaussian process. They derive the analytic forms of the kernel and analyze their eigenvalues.

**Summary Of The Review:**

The paper provides a solid analysis of the role of skip connections in CNNs. The impact and the novelty, however, are limited.

---

> ### Author Response · Authors · 2022-11-18
> **Response to Reviewer zQoE**
>
> We thank the reviewer for their effort.
>
>
>
> The topic is a bit narrow: Our paper deals with kernels associated with residual, convolutional networks. These networks achieve state-of-the-art accuracies on ImageNet. Additionally, residual network architectures are part of network models for many other image-processing applications. The success of residual networks compared to standard CNNs is still not well understood. We feel that our network makes a significant contribution to this important and long-standing question and can lead in the future to developing techniques for network architecture design and constructing other convolutional kernels.
>
>
>
> Outcome for practitioners: We first wish to emphasize that our paper is theoretical and its main objective is to explore the inductive bias of residual, convolutional networks. Still, we believe that our results can motivate future practical methods. Previous spectral analyses of the neural tangent kernel for fully connected networks have motivated the use of positional encoding [6]. [1] used estimates of the condition number of the NTK matrix to guide a neural architecture search. The authors showed a strong correlation between the condition number of the NTK matrices and the trainability of the corresponding neural networks. NTK was further incorporated in matrix completion [2], image inpainting [2], image synthesis [7], and classification and regression [3], [4], [5]. Our paper provides an explicit derivation of ResCNTK, which can be directly plugged into applications, and an analysis that can potentially guide the design of architectures. In addition, it can be used in designing novel convolutional and residual kernels.
>
> [1] Chen, Wuyang, Xinyu Gong, and Zhangyang Wang. "Neural architecture search on imagenet in four gpu hours: A theoretically inspired perspective." arXiv preprint arXiv:2102.11535 (2021).
>
>
>
> [2] Radhakrishnan, Adityanarayanan, George Stefanakis, Mikhail Belkin, and Caroline Uhler. "Simple, fast, and flexible framework for matrix completion with infinite width neural networks." arXiv preprint arXiv:2108.00131 (2021).
>
>
> [3] Arora, Sanjeev, Simon S. Du, Zhiyuan Li, Ruslan Salakhutdinov, Ruosong Wang, and Dingli Yu. "Harnessing the power of infinitely wide deep nets on small-data tasks." arXiv preprint arXiv:1910.01663 (2019).
>
>
> [4] Li, Zhiyuan, Ruosong Wang, Dingli Yu, Simon S. Du, Wei Hu, Ruslan Salakhutdinov, and Sanjeev Arora. "Enhanced convolutional neural tangent kernels." arXiv preprint arXiv:1911.00809 (2019).
>
>
> [5] Shankar, Vaishaal, Alex Fang, Wenshuo Guo, Sara Fridovich-Keil, Jonathan Ragan-Kelley, Ludwig Schmidt, and Benjamin Recht. "Neural kernels without tangents." In International Conference on Machine Learning, pp. 8614-8623. PMLR, 2020.
>
>
> [6] Tancik, Matthew, Pratul Srinivasan, Ben Mildenhall, Sara Fridovich-Keil, Nithin Raghavan, Utkarsh Singhal, Ravi Ramamoorthi, Jonathan Barron, and Ren Ng. "Fourier features let networks learn high frequency functions in low dimensional domains." Advances in Neural Information Processing Systems 33 (2020): 7537-7547.
>
>
> [7] Zhang, Yu-Rong, Sheng Yen Chou, and Shan-Hung Wu. "Generative Adversarial Method Based on Neural Tangent Kernels." arXiv preprint arXiv:2204.04090 (2022).

---

### Official Review · Reviewer_TKdt · 2022-10-26

**Confidence:** 4
**Correctness:** 4
**Technical Novelty And Significance:** 4
**Empirical Novelty And Significance:** Not applicable
**Recommendation:** 8

**Clarity, Quality, Novelty And Reproducibility:**

The paper is well-written. Many of the proof techniques are similar to Geifman et al (2022), but all the findings about residual connections seem novel.

**Strength And Weaknesses:**

Understanding skip connections in CNNs is an important question in the foundations of deep learning, and the present paper provides a comprehensive picture of what role they may play in the context of kernels. This is thus a significant contribution, and I support acceptance.

There are nonetheless a few questions which could deserve more discussion:

* Some of the bounds derived on eigenvalues are not tight, e.g., only lower bounds in Thm 5.2. While the experiments seem to confirm the findings, and suggest the bounds are relevant, is there a way to show upper bounds that also include the $c_k$ quantities?

* In practice, residual networks often involve pooling/downsampling operations in intermediate layers as well, which seem important for good performance (this is also true in convolutional kernels, e.g. in [these](https://arxiv.org/abs/2003.02237) [papers](https://arxiv.org/abs/2102.10032)). Do you have a sense of how these would affect the locality bias in the present paper? Should I interpret your analysis as focusing on a fixed intermediate residual block at a fixed resolution?

* Regarding condition numbers in Section 6.2, this seems to suggest that eigenvalues decay more slowly in the residual case, at least non-asymptotically. Do you have a sense of what could be causing this, and which eigenfunctions might be dominating the spectrum in the residual vs non-residual case? These questions seem important to discuss, since they would give insight on the kinds of target functions for which we can hope that "smaller condition number" => "faster learning" holds.


**Summary Of The Paper:**

The paper studies kernels (GP and NTK) arising from convolutional networks with skip connections, and provides a comparison with those without skip connections. Based on spectral decompositions on products of spheres, the authors make a few findings: (i) the spectral decays are the same as without skip connections, (ii) the skip connections promote a bias towards more localized functions, since the earlier layers play a more prominent role, (iii) when the depth grows, certain parameterizations should be preferred ($\alpha=1$), and the condition number of kernel matrices may be better for residual networks. The results are accompanied by numerical illustrations.

**Summary Of The Review:**

Good paper, interesting and significant contribution.

---

> ### Author Response · Authors · 2022-11-18
> **Response to Reviewer TKdt**
>
> We thank the reviewer for the encouraging comments.
>
>
>
> Upper bound: Unfortunately, the techniques for deriving the $c_k$ for the lower bound are inapplicable for the upper bound. We hope to prove this in the future.
>
>
>
> Pooling/downsampling: Our analysis can handle stride (downsampling), which simply leads to changing $p_i$, the number of paths from input variables to the output. It however does not handle pooling, since generally a kernel with pooling is not multi-dot product.
>
>
>
> Condition numbers: We believe the reason for the difference in the condition number is as follows. Both CGPK and ResCGPK converge to a degenerate, constant kernel as the depth $L$ tends to infinity. However, CGPK converges faster. Therefore, indeed, its lowest eigenvalues, which represent functions that have high frequencies or are non-local, approach zero faster than those of ResCGPK.

---

### Official Review · Reviewer_RWs1 · 2022-10-27

**Confidence:** 3
**Correctness:** 4
**Technical Novelty And Significance:** 4
**Empirical Novelty And Significance:** 2
**Recommendation:** 8

**Clarity, Quality, Novelty And Reproducibility:**

The overall quality and clarity of the paper is good. The author proposed several spectrum bounds for the ResCGPK and ResCNTK and made a comprehensive comparison between them and the kernels of plain CNN. The technical writing for the theorems and proofs is good and well structured to follow.

The paper devotes to justify some phenomenons regarding ResNet that are well known empirically but lack of theoretical explanations. This work makes effort to fill this gap using the NTK method, and the obtained some significant results based on spectrum analysis.

**Strength And Weaknesses:**

Strength:
- An asymptotic bound on the eigenvalues of the kernels of ResNets is given for the first time, which generalized the existing similar bound for that of convolutional networks.
- A new lower bound for the eigenvalues of the kernels of ResNets is given, which implies that the receptive field of ResCNTK is smaller than that of CNTK (without skip connections), thus the ResNet is supposed to be more local biased.
- The author justifies why ResNets converges faster than the normal CNN by utilizing the NTK technology.
- The overall structure and logistics of the paper are good. The formulation and writing of the theorem statements and proofs are well structured and organized.
- The results of experiments match the conclusions derived from theoretical analysis exactly.

Weakness or Questions:
- In Theorem 5.1, the author states that the bound matches the counterpart of the case without skip connections up to constants, and the experiments show that the eigenvalues of ResCGPK are quite close to that of CGPK for the same $k$. The given comparison is between CGPK and ResCGPK, I wonder whether there is any comparison between the eigenvalues of CNTK and ResCNTK ?  Also, this result is verified for some specific function, will the phenomenon that the ratio constants are close to 1 holds true for general functions?

- Theorem 5.2 provides a lower bound of the eigenvalues of ResCNTK. But it seems that there is no strict argument for $\lambda_k$ of ResCNTK being larger than that of CNTK because there is no upper bound for $\lambda_k$ of CNTK. I wonder that which factors would affect the order relationship between them. The author only posted one figure of experiments to qualitatively demonstrate this fact instead of comprehensive quantitive comparison either in numerical or theoretical way.

**Summary Of The Paper:**

This paper derives explicit formulas for kernels of ResNets' Gaussian Process and Neural Tangent kernels, and provide bounds on their implied condition numbers.

The main results include
1) with ReLU activation, the eigenvalues of these residual kernels decay polynomially at a similar rate compared to the same kernel when skip connections are not used.
2) residual kernels are more locally biased.
3) the matrices obtained by the residual kernels have better condition numbers than the counterpart of without the skip connections, enabling therefore faster convergence of training.
4) A theoretical justification for the result that over-parameterized ResNets act like a weighted ensemble of CNNs of various depths.

**Summary Of The Review:**

The theorems proposed in this paper offer several new bounds for the spectrum of ResNets that is helpful for us to understand the practical behavior of ResNets. It is technically innovative to analyze the ResNet with NTK technology and derive some properties of ResNets like frequency-related behavior and locality bias.

---

> ### Author Response · Authors · 2022-11-18
> **Response to Reviewer RWs1**
>
> We thank the reviewer for the encouraging comments.
>
>
>
> Theorem 5.1, the eigenvalues of CNTK vs. ResCNTK: To address this comment, we have empirically compared these eigenvalues.  We note that directly integrating ResCNTK is numerically prohibitive. Instead, we sampled 10,000 points on the multishpere ($d=5$, 3 channels) with a uniform distribution and constructed the respective CNTK and ResCNTK matrices. The plot in https://github.com/anonymous-git446/submission6089/blob/main/eigs_mat_CNTK_L3_eq_n10000_k1000_b2.png shows the first 1000 eigenvalues of these matrices (note that the eigenvalues of kernel matrices include multiplicities, hence the staircase curves). It can be seen that, as with CGPK and ResCGPK, the obtained eigenvalues of CNTK and ResCNTK are nearly the same.
>
>
>
> We finally note that the eigenvalues are independent of the target function. For any learned target function, the rate of convergence of kernel gradient descent (and hence of training an over-parameterized network) is determined by its decomposition onto the eigenfunctions (see, e.g., Arora, ICML 2019).
>
>
>
> Theorem 5.2, the eigenvalues of CNTK vs. ResCNTK: Our analysis does not imply that generically the eigenvalues of ResCNTK are larger than the corresponding eigenvalues of CNTK. Theorem 5.2 indicates that ResCNTK is more biased toward the center of the receptive field than CNTK (for the equivariant architecture) and more locally biased (for the trace and GAP architectures). This suggests that the eigenvalues of ResCNTK that involve center pixels should be larger than the corresponding eigenvalues of CNTK, whereas ones that involve peripheral pixels should be lower. This in fact is seen in Figure 1 in the paper. The eigenvalues of ResCNTK for the exponent pattern $(k,0,0,0)$, which captures the center pixel, are larger than the corresponding eigenvalues of CGPK, whereas the eigenvalues of ResCGPK for $(k,k,k,k)$, which involve peripheral pixels, are smaller than those of CGPK.

---

### Official Review · Reviewer_ciS9 · 2022-11-03

**Confidence:** 3
**Correctness:** 4
**Technical Novelty And Significance:** 3
**Empirical Novelty And Significance:** 3
**Recommendation:** 8

**Clarity, Quality, Novelty And Reproducibility:**

The exposition is quite clear overall. In some minor cases, the reader could use some more or repeated definitions of stated variables.
The theoretical support of the empirically observed phenomenon that over-parameterized ResNets act like a weighted ensemble of CNNs of various depths seems to be novel and also does not seem to follow trivially from past derivations of residual fully-connected or vanilla convolutional kernels.
However, the authors should make the technical challenges of their derivations more precise. Currently, it sounds like a minor technical contribution to combine ideas for convolutional and residual fully-connected kernels.

**Details Of Ethics Concerns:**

I have no ethical concerns.

**Strength And Weaknesses:**

Strengths:
+ Explicit formulas or residual convolutional Gaussian Process Kernels (GPK) and Neural Tangent Kernels (NTKs) are provided.
+ The main insight is provided by Thm. 5.2 and the comparison with the respective convolutional kernels. With their lower bound on the eigenvalues, the authors explain an empirical observation by Veit
et al. (2016): over-parameterized ResNets act like a weighted ensemble of CNNs of various depths.
+ The experiments support the theory and provide some quantitative insights.
+ The importance of the scaling parameter $\alpha$ for large depth and some corrections to typical assumptions in the literature are discussed. (The authors make a case for $\alpha=1$ for convolutional kernels.)
+ Bounds on condition numbers: It is shown that the lower bound for ResCGPK matrices is lower than that of CGPK matrices.
+ The authors claim that they are the first to establish a relationship between skip connections and the condition number of the kernel matrix.

Weaknesses and open questions:
- The authors only study the NTK regime and thus no feature learning. This limits the practical relevance of their insights. However, this is a common challenge with the analytic approach, which still can provide an intuition for some observed phenomena in practice.
- What is the additional challenge of considering convolutional residual kernels in comparison with the derivation of fully-connected ones?
- What is the exact additional challenge in the derivation of the bounds on the eigenvalues in comparison with convolutional kernels?
-> It sounds like a minor technical contribution to simply combine derivations for fully-connected ResNet kernels and convolutional kernels.
- What is the rough dependence of $c_1$ and $c_2$ on the depth in Thm. 5.1? This seems to be one of the most relevant questions. If this is not easily derived theoretically, also empirical results could shed light on that issue. (This question is only partially addressed by Thm. 5.2.)
- How sharp is the bound in Thm. 5.2?
- The experiments could be extended in support of Thm 5.2. An analysis of the dependence on the depth $L$ is missing.

Points of minor critique and open questions:
- The need/advantage of considering three different heads could be better motivated on page 4.
- How is the set $\mathcal{R}$ defined in Theorem 5.1?
- How is $C_0$ defined on page 6? Is it also just an existence statement?
- How is the number of paths $p_i$ scaled in the infinite width limit? Is this quantity not diverging?
- It could strengthen the last result on the condition number to discuss briefly why a slower diverging condition number could help in practice.

**Summary Of The Paper:**

The authors provide explicit formulas or residual convolutional Gaussian Process Kernels (GPK) and Neural Tangent Kernels (NTKs), bounds on their eigenvalues, and their condition numbers.
As a highlight, they support an empirical observation by Veit et al. (2016) (i.e., that over-parameterized ResNets act like a weighted ensemble of CNNs of various depths). Furthermore, they claim to be the first to establish a relationship between skip connections and the condition number of the kernel matrix. Both derivations suggest possible advantages of residual architectures over vanilla convolutional layers without skip connections.

**Summary Of The Review:**

I find the theoretical insights interesting, as they explain empirically observed phenomena.
They are of relevance for the ICLR community that tries to understand possible advantages of residual structures (in comparison with architectures without skip connections).

---

> ### Author Response · Authors · 2022-11-18
> **Response to Reviewer ciS9**
>
> We thank the reviewer for the constructive comments.
>
> Challenge compared to residual FC + convolutional: Kernels associated with fully connected (FC) networks are dot product kernels, for which there is extensive literature. Kernels associated with convolutional networks are multi-dot product kernels. These are significantly more complicated and were analyzed only in very few papers. A particular complication of convolutional kernels is due to their hierarchical structure; the recursive definition of these kernels is affected by the receptive field of nodes in the corresponding network, which varies with the layer depth, and the spectral structure of these kernels varies with either the spatial location of pixels or their mutual distances. For example, while GPK for FC network is formed by a straightforward repeated composition of the $\kappa_1$ kernel ($L$ times for $L$ layers), with CGPK each composition changes the receptive field, and with ResCGPK each new layer sums the composed (residual) and uncomposed (skip) components, which have different receptive fields. As a consequence, proof techniques applicable to FC kernels are generally inapplicable to convolutional ones.
>
> Moreover, proving the bounds on the eigenvalues of both ResCGPK and ResCNTK relies on an analysis of their Taylor expansion. This, in turn, relies on the Taylor expansion of their respective residual, FC kernels, which was not available in the literature. Here ResNKT pose a particular challenge as it is complicated by the bi-directional recursion, rendering the proofs of both the eigenvalue decay rate and leading coefficient significantly more difficult than those for CNTK.
>
> Dependence on depth in Thm 5.1: To address this question, we show in https://github.com/anonymous-git446/submission6089/blob/main/eigs_mat_CGPK_L14_tr_n10000-0_k1000_b%5B1%5D.png a plot of the eigenvalues of the ResCGPK kernel matrix for different depths (10,000 points, $d=5, q=3, C_0=3$). Asymptotically, the global coefficient appears to decay exponentially fast with $L$. This is consistent with Figure 3 in the paper, which shows that the condition number of ResCGPK grows exponentially fast with depth (but still, significantly more slowly than the condition number of CGPK).
> Sharpness of bounds: Figure 1 in the paper indicates that the eigenvalues are close to the lower bound for frequencies concentrated in one pixel -- with $(k,0,0,0)$ the exponent in the lower bound according to Thms. 5.1 and 5.2 is -5, while the measured slope in Figure 1 is –5.3 (and should converge to –5 asymptotically). The exponent in the upper bound for $(k,k,k,k)$ is -11, and the slope in Figure 1 is –9.41. We conclude that the lower bound is sharp and the upper bound is somewhat looser.
>
> Additional experiments in support of Thm 5.2: To address this comment, we have produced a figure (see https://github.com/anonymous-git446/submission6089/blob/main/ResCGPK_2d_eigens_rebuttal.jpg) that compares the eigenvalues of the equivariant kernel for spherical harmonic products with frequencies $(k_1,k_2,0,0)$ (in red) and $(k_1,0,k_2,0)$ (in blue), where the second coordinate aligns with the receptive field center ($L=3$, $q=2$). Consistent with the theorem, the eigenvalues in red, which represent the more central pixels, are larger than those in blue. Also, the polynomial decay with the frequency for each pixel is evident.
>
> The need for three heads: The three heads allow us to analyze kernels corresponding to (1) shift equivariant networks (e.g., image segmentation networks), (2) a ResNet followed by a fully connected head, akin to AlexNet (but with skip connections), and (3) a ResNet followed by global average pooling, as in He 2016. We will include this motivating statement in the text.
>
> $R$ in Thm. 5.1: The set $R$ denotes the receptive field of the kernel, defined as set of input pixels that affect the output.
>
> Definition of $C_0$ in page 6: $C_0$ is the number of input channels. To avoid confusion with multiplicative constants we will rename this instead to $m_0$.
>
> $p_i$ in the infinite width limit: $p_i$ denotes the number of paths in the CNTK kernel, which is equivalent to the number of paths in the respective CNN in which there is only one channel in each node. As such, it is independent of the number of channels. We will clarify this definition.
>
> Why a slower diverging condition number could help in practice: Thank you for this comment. Prior works showed that the convergence rate of kernel gradient descent depends inversely on the condition number [Basri, Neurips 2019; Lee, Neurips, 2019; Xiao, ICML 2020]. Since CNTK is the limit of ConvNets with infinitely many channels, it follows that training overparameterized networks should converge more slowly when the condition number is large. Indeed, the condition number of the kernel matrix has been used as a measure of trainability of finite-size networks in both theoretical [Xiao, ICML 2020] and empirical [Chen, arXiv 2102.11535 2021] works.

---

> > ### Comment · Reviewer_ciS9 · 2022-12-06
> > **Acknowledgement of response**
> >
> > I thank the authors for their detailed response and have increased my score accordingly.

---

### Decision · Program_Chairs · 2023-01-20

**Decision:**

Accept: notable-top-5%

**Justification For Why Not Higher Score:**

N/A

**Justification For Why Not Lower Score:**

The paper has a high quality and no major concern. It is a significant contribution to the area of bridging GPs and neural networks.


**Metareview: Summary, Strengths And Weaknesses:**

The authors provide explicit formulas or residual convolutional Gaussian Process Kernels (GPK) and Neural Tangent Kernels (NTKs), bounds on their eigenvalues, and their condition numbers. As a highlight, they support an empirical observation that over-parameterized ResNets act like a weighted ensemble of CNNs of various depths. Furthermore, they are the first to establish a relationship between skip connections and the condition number of the kernel matrix. Both derivations suggest possible advantages of residual architectures over vanilla convolutional layers without skip connections.

The overall structure and logistics of the paper are good. The formulation and writing of the theorem statements and proofs are well structured and organized. The experiments support the theory and provide good quantitative insights. The paper has a high quality and no major concern. It is a significant contribution to the area of bridging GPs and neural networks.


**Note From Pc:**

if the above contains the word "oral" or "spotlight" please see: "oral" presentation means -> notable-top-5% and "spotlight" means -> notable-top-25%. As stated in our emails, we are disassociating presentation type from AC recommendations